# Leptin receptor+ cells promote bone marrow innervation and regeneration by synthesizing nerve growth factor

Xiang Gao[1,2,3], Malea M. Murphy [4,9], James G. Peyer[4,10], Yuehan Ni[1,5], Min Yang[1,5], Yixuan Zhang[1], Jiaming Guo[1,3], Nergis Kara[4,11], Claire Embree[4], Alpaslan Tasdogan[4,12], Jessalyn M. Ubellacker [4,13], Genevieve M. Crane [6], Shentong Fang [7], Zhiyu Zhao [4], Bo Shen [1,2,4]✉ & Sean J. Morrison [4,8]✉

The bone marrow contains peripheral nerves that promote haematopoietic regeneration after irradiation or chemotherapy (myeloablation), but little is known about how this is regulated. Here we found that nerve growth factor (NGF) produced by leptin receptor-expressing (LepR+) stromal cells is required to maintain nerve fibres in adult bone marrow. In nerveless bone marrow, steady-state haematopoiesis was normal but haematopoietic and vascular regeneration were impaired after myeloablation. LepR+ cells, and the adipocytes they gave rise to, increased NGF production after myeloablation, promoting nerve sprouting in the bone marrow and haematopoietic and vascular regeneration. Nerves promoted regeneration by activating β2 and β3 adrenergic receptor signalling in LepR+ cells, and potentially in adipocytes, increasing their production of multiple haematopoietic and vascular regeneration growth factors. Peripheral nerves and LepR+ cells thus promote bone marrow regeneration through a reciprocal relationship in which LepR+ cells sustain nerves by synthesizing NGF and nerves increase regeneration by promoting the production of growth factors by LepR+ cells.

Peripheral nerves promote the regeneration of diverse tissues, but in most cases little is known about the mechanisms by which they promote regeneration[1–5]. The bone marrow contains peripheral nerves, including sympathetic[6], parasympathetic[7] and sensory[8,9] nerve fibres. Lumbar sympathetic nerve transection depletes sympathetic nerve fibres and Schwann cells in the bone marrow, leading to haematopoietic stem cell (HSC) depletion[5]. Sympathetic denervation with systemic 6-hydroxydopamine does not affect HSC frequency or function under steady-state conditions[4], but systemic ablation of both sympathetic and sensory nerves depletes bone marrow HSCs[8]. Nerve fibres regulate the

[1]National Institute of Biological Sciences, Beijing, China. [2]Tsinghua Institute of Multidisciplinary Biomedical Research, Tsinghua University, Beijing, China. [3]Academy for Advanced Interdisciplinary Studies, Peking University, Beijing, China. [4]Children's Research Institute and the Department of Pediatrics, University of Texas Southwestern Medical Center, Dallas, TX, USA. [5]College of Life Sciences, Beijing Normal University, Beijing, China. [6]Robert J. Tomsich Pathology and Laboratory Medicine Institute, Cleveland Clinic, Cleveland, OH, USA. [7]School of Biopharmacy, China Pharmaceutical University, Nanjing, China. [8]Howard Hughes Medical Institute, UT Southwestern Medical Center, Dallas, TX, USA. [9]Present address: Integrated Microscopy and Imaging Laboratory, Texas A&M Health Science Center, Texas A&M University, College Station, TX, USA. [10]Present address: Cambrian Bio, Inc., New York, NY, USA. [11]Present address: Ensoma, Inc., Boston, MA, USA. [12]Present address: Department of Dermatology, University Hospital Essen and German Cancer Consortium, Essen, Germany. [13]Present address: Department of Molecular Metabolism, Harvard T.H. Chan School of Public Health, Boston, MA, USA. ✉e-mail: ShenBo@nibs.ac.cn; Sean.Morrison@UTSouthwestern.edu

circadian mobilization of haematopoietic stem/progenitor cells into the blood[6,8,10,11] and the regeneration of haematopoiesis after myeloablation by irradiation or chemotherapy[4,7,12,13]. Nerve fibres promote haematopoietic regeneration and changes in haematopoiesis during ageing by activating β adrenergic receptors[4,14,15], though the mechanism by which β adrenergic receptors promote haematopoietic regeneration, and the cells in which they act, are unknown.

Leptin receptor-expressing (LepR+) stromal cells in adult mouse bone marrow synthesize growth factors that promote the maintenance of haematopoietic stem and progenitor cells[16–21] as well as osteogenesis[22,23] and vascular regeneration[24,25]. LepR+ cells promote the maintenance of HSCs and early restricted progenitors by synthesizing stem cell factor (SCF)[16,18,21], CXCL12 (refs. [17,26]), IL7 (ref. [19]), pleiotrophin[20] and Csf1 (refs. [27,28]). Analysis of SCF[16] and CXCL12 (refs. [17,26]) reporter genes, as well as single-cell RNA sequencing[29–32], has shown that LepR+ cells are the major source of these factors in adult bone marrow. LepR+ cells also promote vascular regeneration by producing Angiopoietin-1 (ref. [25]) and VEGF-C[24].

LepR+ cells also include skeletal stem and progenitor cells that form the adipocytes and osteoblasts that arise in adult bone marrow[33–35]. The osteoblasts formed by LepR+ cells contribute to the maintenance and repair of the adult skeleton[33,34,36] and secrete factors that promote osteogenesis[22,23]. The adipocytes that arise from LepR+ cells in adult bone marrow promote the regeneration of HSCs and haematopoiesis after myeloablation by synthesizing SCF[35]. LepR+ cells and adipocytes also promote HSC maintenance and quiescence by secreting adiponectin, which suppresses inflammation[37]. In this article, bone marrow nerve fibers were found to be maintained by NGF synthesized by LepR+ cells and, in turn, nerves promote haematopoietic and vascular regeneration by secreting adrenergic neurotransmitters that activate β2/β3 adrenergic receptor signaling in LepR+ cells.

## Results

### Nerve growth factor is mainly synthesized by LepR+ cells

Peripheral nerves require neurotrophic factors for their maintenance[38], but the source of such factors in the bone marrow is unknown. Analysis of published microarray data[16] (National Center for Biotechnology Information (NCBI) accession number GSE33158) suggested that nerve growth factor (Ngf) was the only neurotrophic factor detected in adult bone marrow (Fig. 1a). Ngf expression was detected in Scf–GFP+CD45−Ter119−CD31− stromal cells, nearly all of which are LepR+(ref. [33]), but little or no Ngf was detected in osteoblasts, endothelial cells or unfractionated whole bone marrow (WBM) cells (Fig. 1a). Similar results were obtained by RNA sequencing[22] (NCBI accession number PRJNA914703), which detected Ngf in PDGFRα+CD45−Ter119−CD31− stromal cells, nearly all of which are LepR+(ref. [33]), but not in endothelial cells or WBM cells (Fig. 1b).

Single-cell RNA sequencing of enzymatically dissociated cells from the femurs and tibias of 8-week-old mice showed that most Ngf-expressing cells in adult bone marrow were LepR+ cells (Fig. 1c,d; NCBI accession number PRJNA835050)[39]. Ngf was also expressed by a much smaller number of SMA+NG2+ smooth muscle cells, and by rare

osteoblasts (OLC-2 cells), Schwann cells and fibroblasts (Fig. 1c–e). Little or no Ngf was detected in endothelial cells, chondrocytes or other stromal cells (Fig. 1c–e). Quantitative reverse-transcription polymerase chain reaction (qRT-PCR) confirmed that Ngf was highly expressed by LepR+CD45−Ter119−CD31− stromal cells and SMA+NG2+ smooth muscle cells, with approximately 100-fold lower expression by Col2.3−GFP+CD45−Ter119−CD31− osteoblasts and no expression by endothelial cells or WBM cells (Fig. 1f). LepR+ cells and smooth muscle cells were thus the main sources of NGF in the bone marrow.

To identify the location of Ngf-expressing cells in adult bone marrow, we generated an Ngf–mScarlet (Ngf^mScarlet) knock-in reporter allele (Extended Data Fig. 1a–c). Confocal imaging[40] of cleared femurs from adult Ngf^mScarlet/+ mice showed that Ngf–mScarlet was expressed by stromal cells surrounding Endomucin^low arterioles as well as Endomucin^high sinusoids (Fig. 1g). While the peri-arteriolar staining appeared more prominent, the abundance of sinusoids throughout the bone marrow meant that most of the Ngf–mScarlet staining was peri-sinusoidal. The peri-arteriolar staining probably reflected Ngf–mScarlet expression by both peri-arteriolar LepR+Osteolectin+ cells[36] as well as SMA+NG2+ smooth muscle cells (Fig. 1e).

Flow cytometric analysis of enzymatically dissociated bone marrow cells showed that $0.087 \pm 0.029\%$ of bone marrow cells were Ngf–mScarlet+ (Fig. 1h). Consistent with the single-cell RNA sequencing (Fig. 1d), $89 \pm 5.3\%$ of bone marrow Ngf–mScarlet+ cells were LepR+ (Fig. 1h) and $82 \pm 10\%$ of all bone marrow LepR+ cells were Ngf–mScarlet+ (Fig. 1i). The remaining ~10% of Ngf–mScarlet+ cells that were negative for LepR within the bone marrow were mainly SMA+NG2+ smooth muscle cells (Extended Data Fig. 1d,f). There were also rare osteoblasts (Extended Data Fig. 1e,g), Schwann cells (Extended Data Fig. 1h) and macrophages (Extended Data Fig. 1i,j) that were Ngf–mScarlet+. The flow cytometry gates used to sort each cell population characterized in this study are shown in Extended Data Fig. 2.

### NGF from LepR+ cells is required for bone marrow innervation

To test if NGF is required for bone marrow innervation we generated mice with a floxed Ngf allele (Extended Data Fig. 3a–c). We conditionally deleted Ngf in LepR+ cells using Lepr^cre, in smooth muscle cells using NG2–creER, in osteoblasts using Col1a1–creER, and in Schwann cells using GFAP–cre. Deletion from smooth muscle cells, osteoblasts or Schwann cells had no significant effect on bone marrow NGF levels (Fig. 2a) or the number of nerve fibres in adult bone marrow (Fig. 2b and Extended Data Fig. 3d–g). Therefore, smooth muscle cells, osteoblasts and Schwann cells were not significant sources of NGF for nerve maintenance in bone marrow.

Adult Lepr^cre/+; Ngf^fl/Δ mice were born in expected numbers and did not differ from littermate controls in terms of gross appearance (Extended Data Fig. 4a), body length (Extended Data Fig. 4b) or body mass (Extended Data Fig. 4c). LepR+ cells from Lepr^cre/+; Ngf^fl/Δ mice had Ngf transcript levels that were approximately 30% of control levels at 2 months of age and less than 10% of control levels at 6 months of age (Fig. 2e). Deletion of Ngf from LepR+ cells profoundly depleted NGF from the bone marrow by 6 months of age (Fig. 2a).

---

**Fig. 1 | Ngf is mainly expressed in the bone marrow by LepR+ stromal cells.**
**a,b,** The expression of neurotrophic factors by microarray analysis[16] (**a**) and RNA sequencing[22] (**b**) in bone marrow stromal cells (isolated on the basis of expression of Scf–GFP (**a**) or PDGFRα (**b**) staining, both of which are nearly completely overlapping with LepR expression[33]), VE-cadherin+ bone marrow endothelial cells, Col2.3–GFP+CD45−Ter119−CD31− osteoblasts, and WBM cells (three mice in **a** and two mice in **b**, from three or two independent experiments, respectively). **c,** Uniform manifold approximation and projection (UMAP) plot showing clustering of single-cell RNA sequencing analysis of 4,209 non-haematopoietic cells from enzymatically dissociated bones/bone marrow in 8-week-old mice[39]. **d,** Ngf is mainly expressed by Lepr+ stromal cells (cell cluster 11 in **c**) and smooth muscle cells (cell cluster 12). **e,** Ngf expression by all cell clusters shown in **c**

(cells were obtained from four mice and analysed in three independent experiments). **f,** Ngf expression by qRT–PCR in LepR+CD45−Ter119−CD31− stromal cells, NG2–DsRed+ smooth muscle cells, Col1a1–GFP+ osteoblasts, VE-cadherin+ endothelial cells and unfractionated cells from the bone marrow of 2-month-old mice (three mice from three independent experiments). **g,** Deep imaging of femur bone marrow from adult Ngf^mScarlet/+ mouse: the Ngf–mScarlet+ cells were found around endomucin^high sinusoids (arrowhead) as well as around endomucin^low arterioles (arrow; the images are representative of five mice). **h,i,** Flow cytometric analysis of enzymatically dissociated bone marrow from Ngf^mScarlet/+ mice: 89% of Ngf–mScarlet+ cells were LepR+, and most LepR+ cells were Ngf–mScarlet+ (four mice from four independent experiments). SSC-A, side scatter area. All data represent mean ± standard deviation.

No bone marrow innervation defect was apparent in *Lepr*$^{cre/+}$;*Ngf*$^{fl/\Delta}$ mice during development as the number of nerve fibres in the bone marrow was normal in 2-month-old *Lepr*$^{cre/+}$;*Ngf*$^{fl/\Delta}$ mice (Fig. 2d). However, by 6 months of age, when recombination in LepR$^+$ cells was nearly complete, we observed virtually no nerve fibres in the bone marrow of *Lepr*$^{cre/+}$;*Ngf*$^{fl/\Delta}$ mice (Fig. 2b–d). Peripheral nerves appeared to be present in normal numbers in the quadriceps of 6-month-old *Lepr*$^{cre/+}$;

*Ngf*$^{fl/\Delta}$ mice (Extended Data Fig. 4d–f). Nerve fibres thus grew into the bone marrow normally during development in *Lepr*$^{cre/+}$;*Ngf*$^{fl/\Delta}$ mice but became depleted within the bone marrow, but not outside of the bone marrow, by 6 months of age, when NGF was depleted to less than 10% of control levels in the bone marrow.

Consistent with prior studies[4,6,11], loss of nerve fibres from the bone marrow did not have any gross effect on steady-state

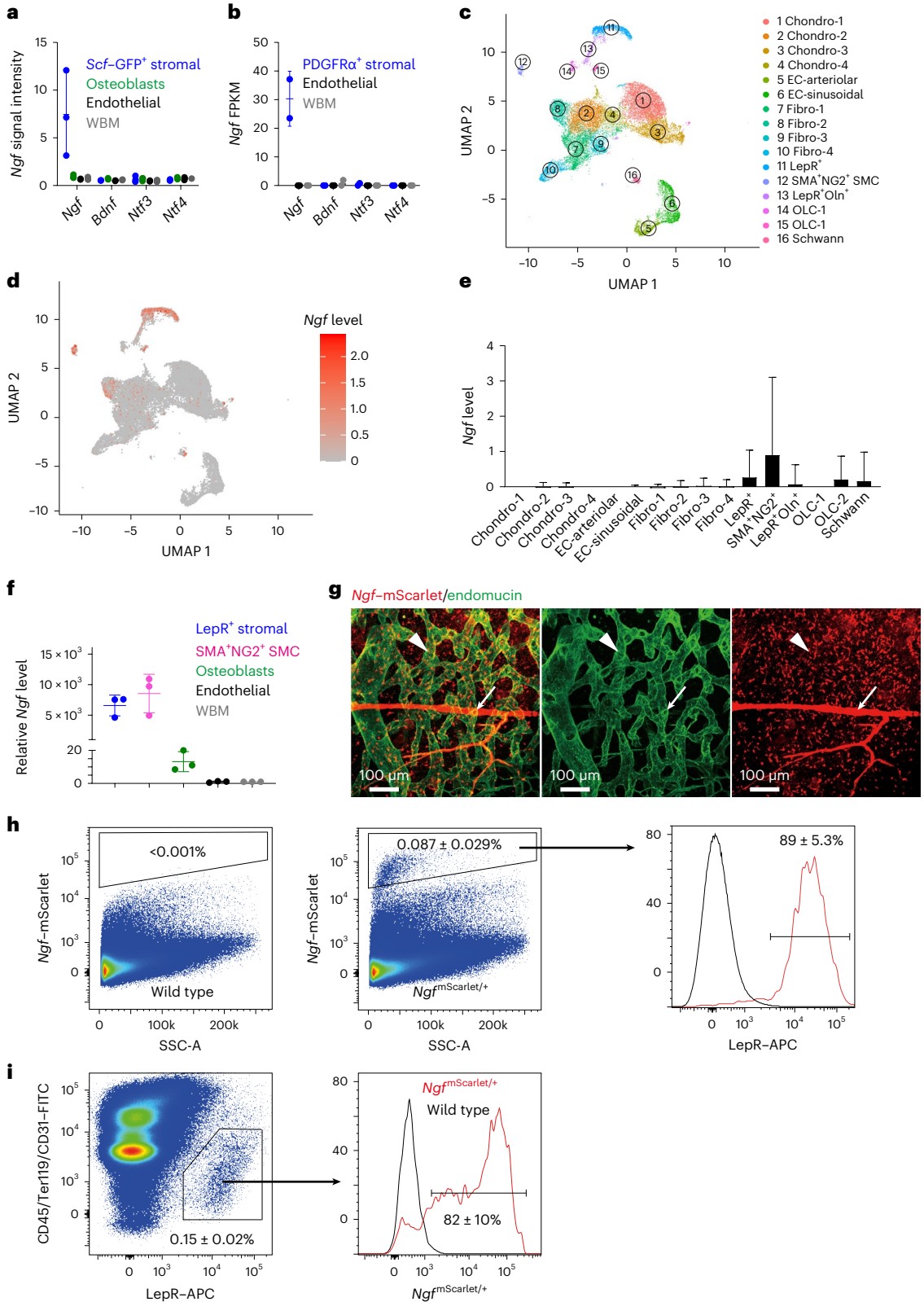

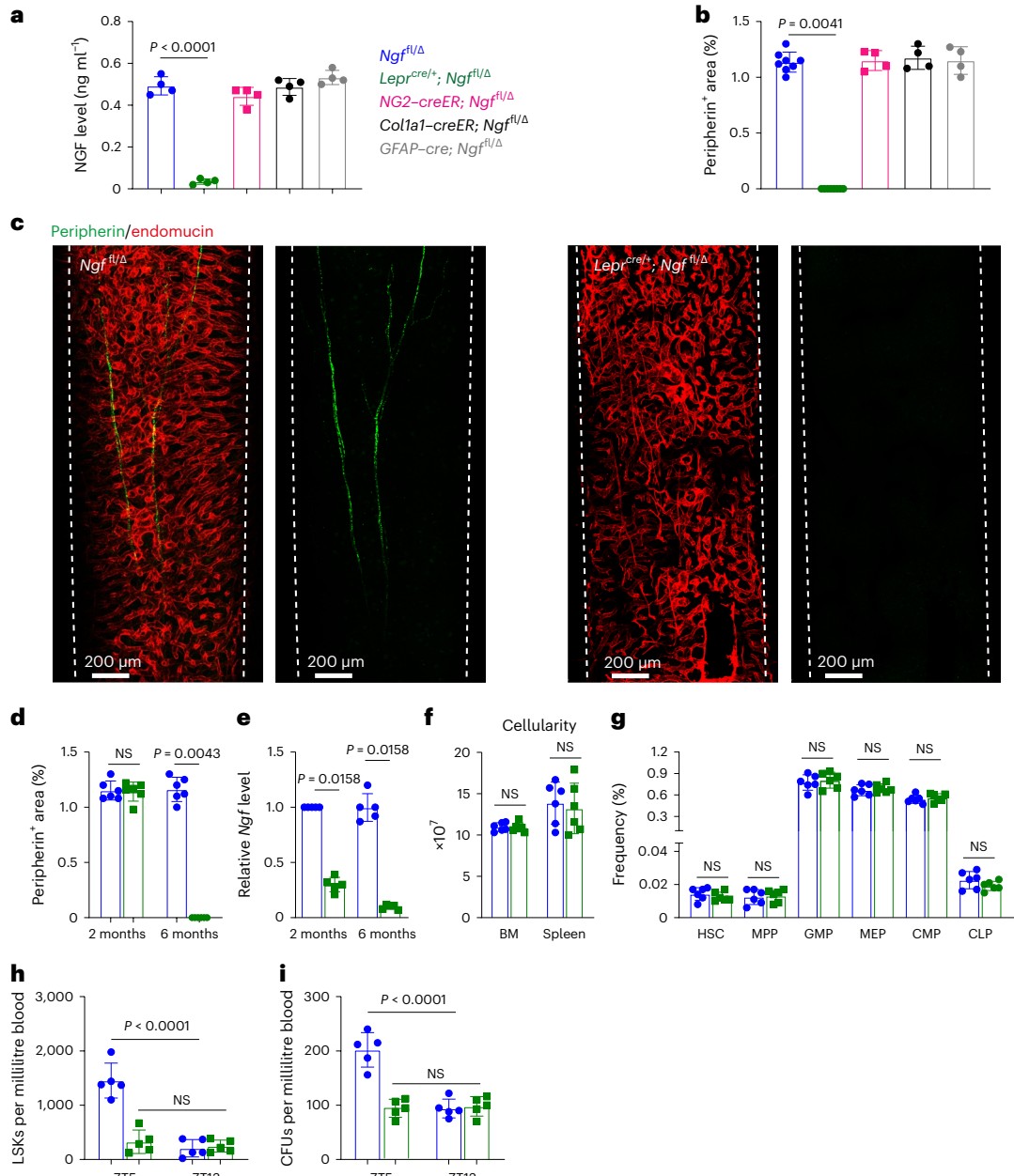

**Fig. 2 | NGF from LepR⁺ cells is necessary to maintain nerve fibres in the bone marrow. a,b,** NGF protein levels in bone marrow serum (**a**) and the area occupied by peripheral nerves in bone marrow (**b**) from 6–8-month-old *Ngf*^fl/Δ^ control (*n* = 8), *Lepr*^cre/+^;*Ngf*^fl/Δ^ (*n* = 8), *NG2*–creER;*Ngf*^fl/Δ^ (*n* = 4), *Col1a1*–creER; *Ngf*^fl/Δ^ (*n* = 4), and *GFAP*–cre;*Ngf*^fl/Δ^ mice (*n* = 4) (four to eight mice per genotype from four independent experiments). **c**, Deep imaging of femur bone marrow from 6–8-month-old *Lepr*^cre/+^;*Ngf*^fl/Δ^ and *Ngf*^fl/Δ^ littermate control mice (images are representative of three experiments with one mouse per genotype per experiment). **d,e,** Peripheral nerves were present in normal numbers in the bone marrow of 2-month-old *Lepr*^cre/+^;*Ngf*^fl/Δ^ mice but were absent from the bone marrow of 6-month-old *Lepr*^cre/+^;*Ngf*^fl/Δ^ mice (*n* = 6) (**d**) when the efficiency of *Ngf* deletion was more than 90% (*n* = 5) (**e**) (five to six mice per genotype per age from five to six independent experiments). **f,g,** Bone marrow and spleen cellularity (**f**) and haematopoietic stem and progenitor cell frequencies in the bone marrow (**g**)

of 6–8-month-old *Lepr*^cre/+^;*Ngf*^fl/Δ^ and *Ngf*^fl/Δ^ littermate control mice (six mice per genotype from six independent experiments). **h,i,** Defective circadian regulation of haematopoietic stem/progenitor cell mobilization into the blood of *Lepr*^cre/+^; *Ngf*^fl/Δ^ mice based on numbers of LSK cells (**h**) and colony-forming progenitors (**i**) per millilitre of blood at different Zeitgeber times (ZT5, late morning; ZT13, just after nightfall; five mice per genotype from five independent experiments). All data represent mean ± standard deviation. The statistical significance of differences among treatments was assessed using a one-way ANOVA (**a**) followed by the Dunnett's multiple comparisons adjustment, a Kruskal–Wallis test (**b**) followed by the Dunn's multiple comparisons adjustment, Mann–Whitney tests (**d** and **e**) or Student's *t*-tests (**f** and **g**) followed by the Holm–Šidák's multiple comparisons adjustment, or matched samples two-way ANOVAs (**h** and **i**) followed by the Šidák's multiple comparisons adjustment. All the statistical tests were two-sided. Not significant (NS): *P* > 0.05.

haematopoiesis. Six-month-old *Lepr*^cre/+^; *Ngf*^fl/Δ^ mice did not differ from littermate controls in terms of bone marrow or spleen cellularity (Fig. 2f), or the frequencies of HSCs, multipotent haematopoietic progenitors (MPPs), granulocyte–macrophage progenitors (GMPs),

megakaryocyte–erythroid progenitors (MEPs), common myeloid progenitors (CMPs) or common lymphoid progenitors (CLPs) in the bone marrow (Fig. 2g). There were also no differences in blood cell counts (Extended Data Fig. 4g–i) or in the frequencies of B220⁺

B cells, CD3$^+$ T cells, Gr1$^+$Mac1$^+$ myeloid cells, CD41$^+$ megakaryocyte lineage cells or CD71$^+$/Ter119$^+$ erythroid lineage cells in the bone marrow or spleen (Extended Data Fig. 4j–n). Finally, WBM cells from 6-month-old $Lepr^{cre/+}$; $Ngf^{fl/\Delta}$ mice and littermate controls did not differ in their capacity to reconstitute myeloid, B or T cells upon competitive transplantation into irradiated mice (Extended Data Fig. 4o–r). Bone marrow nerve fibres thus appear to be dispensable for normal adult haematopoiesis.

In agreement with earlier studies[6,11], we did observe a defect in the circadian mobilization of Lineage⁻Sca1⁺c-kit⁺ (LSK) haematopoietic stem/progenitor cells (Fig. 2h) and colony-forming progenitors (Fig. 2i) into the blood during midmorning (Zeitgeber Time 5) in 6-month-old $Lepr^{cre/+}$; $Ngf^{fl/\Delta}$ as compared with littermate control mice.

## Bone marrow innervation promotes haematopoietic regeneration

To test for haematopoietic regeneration defects, we lethally irradiated (1,080 rads) and transplanted a radioprotective dose of 1,000,000 WBM cells into 6-month-old $Lepr^{Cre/+}$; $Ngf^{fl/\Delta}$ mice and littermate controls. While all control mice survived, 41% (9 of 22) of $Lepr^{Cre/+}$; $Ngf^{fl/\Delta}$ mice died between 10 and 18 days after irradiation, consistent with haematopoietic failure (Fig. 3a). The $Lepr^{Cre/+}$; $Ngf^{fl/\Delta}$ mice exhibited significantly lower white blood cell (WBC, Fig. 3b), red blood cell (RBC, Fig. 3c) and platelet (PLT) counts (Fig. 3d) as well as bone marrow cellularity (Fig. 3e) and LSK cell numbers (Fig. 3f) at 14 and 28 days after irradiation. At 28 days after irradiation, HSC numbers were much lower in the bone marrow of $Lepr^{Cre/+}$; $Ngf^{fl/\Delta}$ as compared with littermate control mice (Fig. 3g). We thus observed broad reductions in blood and bone marrow cell counts as well as the numbers of haematopoietic stem and progenitor cells in $Lepr^{Cre/+}$; $Ngf^{fl/\Delta}$ mice at 14–28 days after irradiation.

We also assessed vascular and stromal cell regeneration in $Lepr^{Cre/+}$; $Ngf^{fl/\Delta}$ and littermate control mice. At 10 days after lethal irradiation and transplantation, we observed significantly reduced numbers of bone marrow cells (Fig. 3h), LSK cells (Fig. 3i), LepR$^+$ stromal cells (Fig. 3j) and endothelial cells (Fig. 3k), as well as increased vascular leakage (Fig. 3l) in the bone marrow of $Lepr^{Cre/+}$; $Ngf^{fl/\Delta}$ mice as compared with littermate controls. At 28 days after irradiation, blood vessels were patent in control mice but remained leaky (Fig. 3m) and morphologically abnormal (Fig. 3n) in $Lepr^{Cre/+}$; $Ngf^{fl/\Delta}$ mice. $Lepr^{Cre/+}$; $Ngf^{fl/\Delta}$ mice had significantly less proliferation by LepR$^+$ cells at 14 days after irradiation (Fig. 3o) and by endothelial cells at 14 and 28 days after irradiation (Fig. 3p). In this experiment we observed trends towards reduced numbers of LepR$^+$ cells and endothelial cells at 14 days after irradiation and significant reductions in the numbers of these stromal cells at 28 days after irradiation (Fig. 3q,r). The loss of nerve fibres from the bone marrow in $Lepr^{Cre/+}$; $Ngf^{fl/\Delta}$ mice was thus associated with broad defects in the regeneration of haematopoietic, stromal and vascular cells at 10–28 days after irradiation.

To test if defects in haematopoietic regeneration were also evident after sublethal irradiation, we administered 650 rads to 6-month-old $Lepr^{Cre/+}$; $Ngf^{fl/\Delta}$ and littermate control mice. The $Lepr^{Cre/+}$; $Ngf^{fl/\Delta}$ mice exhibited significantly reduced survival from 12 to 15 days after irradiation (Extended Data Fig. 5a) as well as reduced bone marrow cellularity (Extended Data Fig. 5b), HSC numbers (Extended Data Fig. 5c) and LSK cell numbers (Extended Data Fig. 5d) as compared with littermate controls at 28 days after irradiation.

To test if defects in haematopoietic, vascular and stromal cell regeneration were evident after chemotherapy, we treated $Lepr^{Cre/+}$; $Ngf^{fl/\Delta}$ and littermate control mice with 5-fluorouracil (5-FU). The $Lepr^{Cre/+}$; $Ngf^{fl/\Delta}$ mice exhibited significantly reduced survival from 12 to 15 days after irradiation (Fig. 3s) as well as reduced numbers of bone marrow cells (Fig. 3t), HSCs (Fig. 3u) and LSK cells (Fig. 3v) as compared with littermate controls at 12 days after 5-FU treatment. Impaired haematopoietic regeneration was thus observed in $Lepr^{Cre/+}$;

$Ngf^{fl/\Delta}$ mice irrespective of whether myeloablation was induced by 5-FU treatment, sublethal irradiation or lethal radiation.

## NGF acts locally to promote nerve maintenance in bone marrow

Neurotrophic factors act locally to promote the survival of innervating neurons[41,42]. To test if NGF acts locally within the bone marrow to promote nerve fibre maintenance, we deleted $Ngf$ using $Prx1$–cre. $Prx1$–cre recombines in limb mesenchymal cells, including in LepR$^+$ cells that form in the bone marrow of limb bones, but not within the axial skeleton[33,39,43]. Two-month-old $Prx1$–cre; $Ngf^{fl/fl}$ mice exhibited a lack of nerve fibres in femur bone marrow but normal innervation of vertebral bone marrow (Fig. 4a–c). This demonstrates that NGF acts locally within the bone marrow to promote nerve fibre maintenance.

Consistent with the phenotype observed in $Lepr^{Cre/+}$; $Ngf^{fl/\Delta}$ mice, $Prx1$–cre; $Ngf^{fl/fl}$ mice exhibited normal bone marrow haematopoiesis under steady-state conditions but impaired haematopoietic and vascular regeneration in limb bones. Compared with littermate controls, $Prx1$–cre; $Ngf^{fl/fl}$ mice exhibited normal femur bone marrow, vertebral bone marrow and spleen cellularity (Fig. 4d), and normal frequencies of HSCs and restricted haematopoietic progenitors (Fig. 4e) in femur bone marrow. WBM cells from the femurs of $Prx1$–cre; $Ngf^{fl/fl}$ mice and littermate controls did not differ in their capacity to reconstitute myeloid, B or T cells upon competitive transplantation into irradiated mice (Fig. 4f and Extended Data Fig. 5e–g).

To assess the regeneration of haematopoiesis after irradiation, we lethally irradiated (1,080 rads) and transplanted a radioprotective dose of 1,000,000 WBM cells into 2-month-old $Prx1$–cre; $Ngf^{fl/fl}$ mice and littermate controls. At 28 days after irradiation, the regeneration of bone marrow cellularity (Fig. 4g), HSCs (Fig. 4h), LSK cells (Fig. 4i), LepR$^+$ cells (Fig. 4j) and endothelial cells (Fig. 4k) were all significantly impaired in femur, but not vertebral, bone marrow in $Prx1$–cre; $Ngf^{fl/fl}$ mice. Consistent with this, femur, but not vertebral, bone marrow blood vessels in $Prx1$–cre; $Ngf^{fl/fl}$ mice were leaky at 28 days after irradiation (Fig. 4l). Blood cell counts did not significantly differ between $Prx1$–cre; $Ngf^{fl/fl}$ and littermate control mice before or after irradiation (Extended Data Fig. 5h–j), consistent with the observation that haematopoietic regeneration was impaired only in limb bones. NGF thus acts locally within the bone marrow to promote haematopoietic, stromal and vascular cell regeneration.

## Nerve sprouting increases regeneration factor expression

When compared with non-irradiated bone marrow, we observed a significant increase in NGF levels at 14 days after irradiation in 6-month-old control mice but not in 6-month-old $Lepr^{Cre/+}$; $Ngf^{fl/\Delta}$ mice (Fig. 5a). In the bone marrow, $Ngf$–mScarlet was mainly expressed by adipocytes (Fig. 5b) and LepR$^+$ cells (Fig. 5c and Extended Data Fig. 6a) at 14 days after irradiation. Little or no $Ngf$–mScarlet expression was observed among haematopoietic/endothelial cells or LepR negative stromal cells in the bone marrow (Fig. 5c). By qRT–PCR, $Ngf$ levels were similar in adipocytes and in LepR$^+$ cells (Fig. 5d).

We observed significantly increased nerve fibre density in control bone marrow after irradiation but not in the bone marrow of 6-month-old $Lepr^{Cre/+}$; $Ngf^{fl/\Delta}$ mice (Fig. 5e). We irradiated $Wnt1$–cre; $Rosa26$–tdTomato mice, which express Tomato in neural crest-derived cells, including nerve fibres and Schwann cells[44]. Nerve fibres were much more abundant in the bone marrow at 14 days after irradiation as compared with non-irradiated controls (Fig. 5f,g). Nerve fibres were closely associated with arterioles under steady-state conditions and after irradiation (Extended Data Fig. 6b). By 28 days after irradiation, when NGF levels in the bone marrow returned nearly to normal (Fig. 5a), the density of nerve fibres also returned nearly to normal (Fig. 5g).

To test if the increase in NGF levels in the bone marrow after irradiation caused nerve fibre sprouting, we examined 4–5-month-old

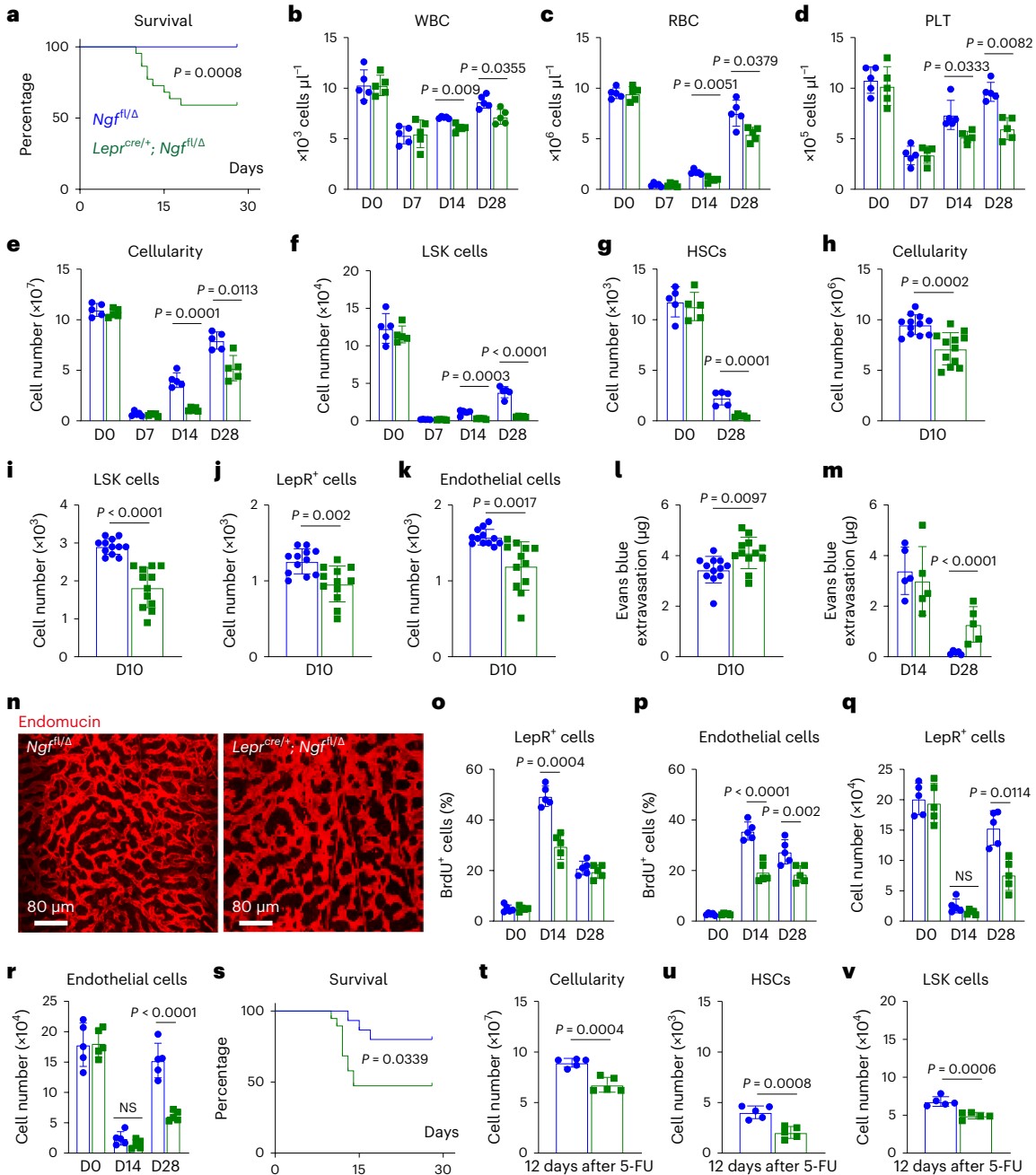

**Fig. 3 | $Lepr^{cre/+}$; $Ngf^{fl/\Delta}$ mice exhibit defects in haematopoietic and vascular regeneration after irradiation. a**, Survival of $Lepr^{cre/+}$; $Ngf^{fl/\Delta}$ and $Ngf^{fl/\Delta}$ littermate control mice after irradiation and transplantation of wild-type bone marrow cells (22 mice per genotype from 3 independent experiments). **b–d**, WBC (**b**), RBC (**c**) and PLT (**d**) counts from $Lepr^{cre/+}$; $Ngf^{fl/\Delta}$ and littermate control mice before (D0) and 7, 14 and 28 days after irradiation. **e,f**, Bone marrow cellularity (**e**) and LSK cell numbers (**f**) from 6-month-old $Lepr^{cre/+}$; $Ngf^{fl/\Delta}$ and littermate control mice on day (D)0, D7, D14 and D28 after irradiation. **g**, Numbers of HSCs in bone marrow (always one tibia and one femur) from $Lepr^{cre/+}$; $Ngf^{fl/\Delta}$ and littermate control mice before and 28 days after irradiation. **h–k**, Cellularity (**h**) and numbers of LSK cells, (**i**) LepR$^+$ cells (**j**) and endothelial cells (**k**) in the bone marrow of $Lepr^{cre/+}$; $Ngf^{fl/\Delta}$ and littermate control mice on D10 after irradiation (a total of 12 mice per genotype from 3 independent experiments). **l,m**, Leakage of intravenously injected Evans blue dye into femur bone marrow at D10 (**l**), D14 and D28 (**m**) after irradiation of $Lepr^{cre/+}$; $Ngf^{fl/\Delta}$ and littermate control mice. **n**, Endomucin staining of the vasculature in the bone marrow of $Lepr^{cre/+}$; $Ngf^{fl/\Delta}$ and littermate control mice 28 days after irradiation (representative of three experiments).

**o,p**, The percentages of LepR$^+$ cells (**o**) and endothelial cells (**p**) from $Lepr^{cre/+}$; $Ngf^{fl/\Delta}$ and littermate control mice that incorporated a 48-h pulse of bromodeoxyuridine (BrdU) 28 days after irradiation. **q,r**, Numbers of LepR$^+$ cells (**q**) and endothelial cells (**r**) in the bone marrow 28 days after irradiation. **s**, Survival of 6-month-old $Lepr^{cre/+}$; $Ngf^{fl/\Delta}$ and $Ngf^{fl/\Delta}$ littermate control mice after 5-FU treatment (19 $Lepr^{cre/+}$; $Ngf^{fl/\Delta}$ mice and 15 $Ngf^{fl/\Delta}$ mice in 3 independent experiments). **t–v**, Cellularity (**t**), numbers of HSCs (**u**) and LSK cells (**v**) in bone marrow from 6-month-old $Lepr^{cre/+}$; $Ngf^{fl/\Delta}$ and littermate control mice 12 days after 5-FU treatment. Unless otherwise specified, each panel shows five mice per genotype from five independent experiments per timepoint. All data represent mean ± standard deviation. Statistical significance was assessed using log-rank tests (**a** and **s**), two-way ANOVAs (**m** and **p**) or matched samples two-way ANOVAs (**b** and **d**) followed by the Šidák's multiple comparisons adjustment, Student's $t$-tests (**h**, **l** and **t**) or Student's $t$-tests (**c**, **e–g**, **o**, **q**, **r**, **u** and **v**) or Welch's $t$-tests (**i–k**) followed by the Holm–Šidák's multiple comparisons adjustment. All statistical tests were two-sided. Not significant (NS) means $P > 0.05$.

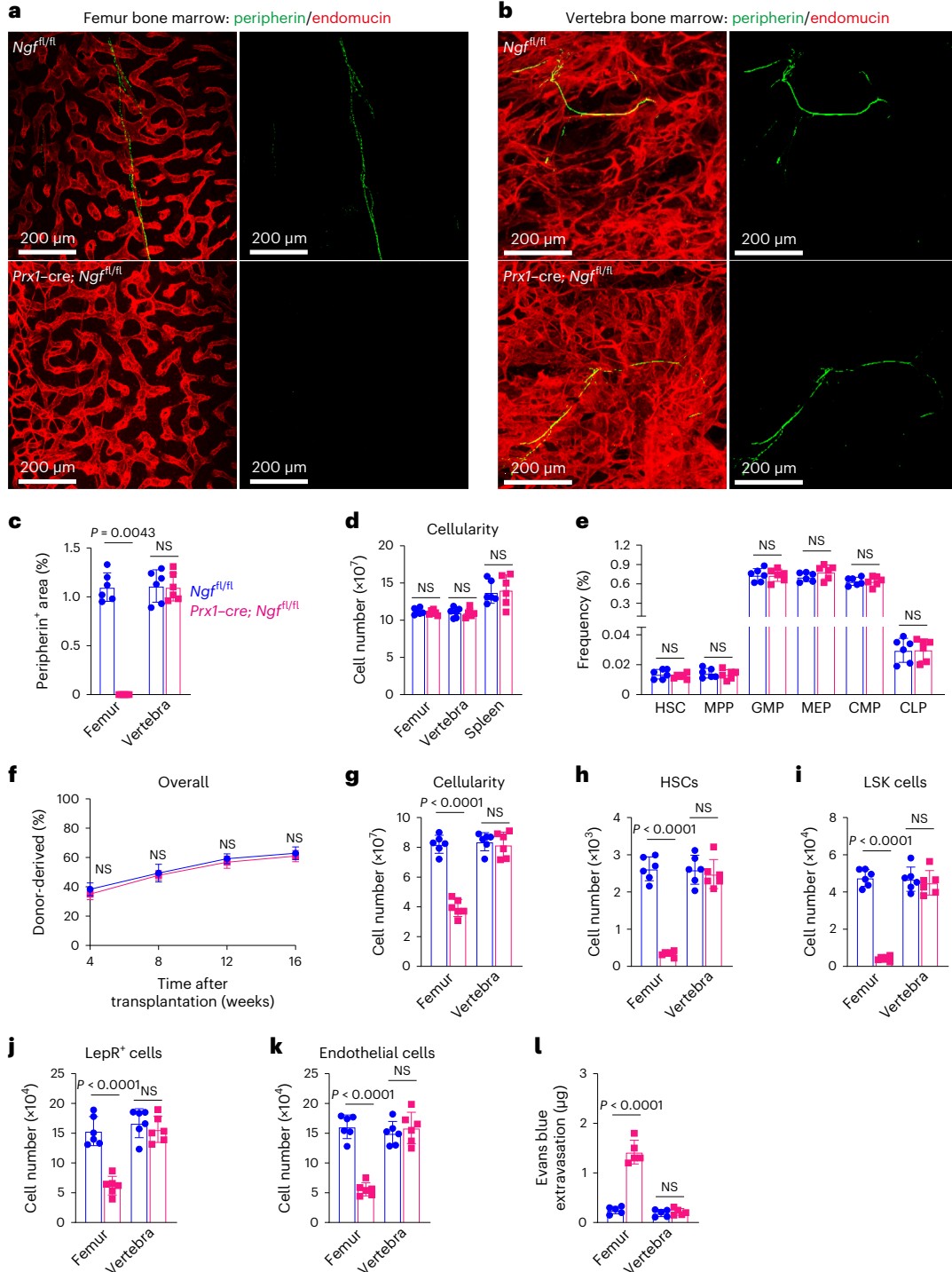

**Fig. 4 | _Prx1–cre;Ngf_<sup>fl/fl</sup> mice exhibit a loss of nerve fibres as well as defects in haematopoietic and vascular regeneration in long bones but not in vertebrae. a,b,** Nerve fibres (green) were visible in femur (**a**) and vertebra (**b**) bone marrow from _Ngf_<sup>fl/fl</sup> control mice and in vertebra bone marrow (**b**) from 2-month-old _Prx1–cre;Ngf_<sup>fl/fl</sup> mice but not in femur bone marrow (**a**) from _Prx1–cre;Ngf_<sup>fl/fl</sup> mice (representative of three independent experiments). **c,** The area occupied by peripherin⁺ nerve fibres in bone marrow sections from 2-month-old _Prx1–cre;Ngf_<sup>fl/fl</sup> and littermate control mice. **d–f,** Under steady-state conditions, 2-month-old _Prx1–cre;Ngf_<sup>fl/fl</sup> mice did not significantly differ from littermate control mice in terms of spleen, femur bone marrow or vertebral bone marrow cellularity (**d**), the frequencies of haematopoietic stem and progenitor cell populations in femur bone marrow (**e**) (six mice per genotype from six independent experiments in **c–e**), or the levels of donor cell reconstitution upon competitive transplantation into irradiated mice (**f,** femur bone marrow cells

from five donor mice were transplanted into a total of five recipients per donor per genotype in five independent experiments). **g–k,** At 28 days after irradiation, _Prx1–cre;Ngf_<sup>fl/fl</sup> and littermate control mice did not significantly differ in terms of bone marrow cellularity (**g**) and the numbers of HSCs (**h**), LSK cells (**i**), LepR⁺ cells (**j**) or endothelial cells (**k**) in the vertebrae, but all of these parameters were significantly lower in femur bone marrow (six mice from six independent experiments). **l,** Leakage of intravenously injected Evans blue dye into femur and vertebra bone marrow 28 days after irradiation (five mice from five independent experiments). All data represent mean ± standard deviation. The statistical significance of differences among treatments was assessed using Mann–Whitney tests (**c**), Student's _t_-tests (**d**, **e** and **l**) or Welch's _t_-tests (**i**) followed by the Holm–Šidák's multiple comparisons adjustment, or matched samples two-way ANOVAs (**f–h**, **j** and **k**) followed by the Šidák's multiple comparisons adjustment. All the statistical tests were two-sided. Not significant (NS): _P_ > 0.05.

$Lepr^{Cre/+}$; $Ngf^{fl/\Delta}$ mice. These mice had lower levels of NGF in the bone marrow as compared with littermate controls (Extended Data Fig. 6c). They had a normal density of nerve fibres in the bone marrow before irradiation but, unlike control mice, did not exhibit an increase in nerve fibres 14 days after irradiation (Extended Data Fig. 6d). These 4–5-month-old $Lepr^{Cre/+}$; $Ngf^{fl/\Delta}$ mice exhibited delayed regeneration of bone marrow cellularity (Extended Data Fig. 6e), HSCs (Extended Data Fig. 6f) and LSK cells (Extended Data Fig. 6g) as compared with littermate controls. Therefore, the sprouting of nerve fibres in the bone marrow after irradiation occurs in response to increased NGF production by LepR$^+$ cells, and the adipocytes they give rise to, and this accelerates haematopoietic regeneration.

We hypothesized that bone marrow nerve fibres increased the production of growth factors that promote haematopoietic and vascular regeneration. We found by enzyme-linked immunosorbent assay analysis that SCF (Fig. 5h), VEGF (Fig. 5i) and Ang2 (Fig. 5j) levels increased significantly in the bone marrow of control mice at 14 days after irradiation but to a significantly lesser extent in the bone marrow of 6-month-old $Lepr^{Cre/+}$; $Ngf^{fl/\Delta}$ mice. By 28 days after irradiation, when NGF levels and nerve fibre density had returned to normal in control mice (Fig. 5a,e), SCF, VEGF and Ang2 levels had also returned to normal (Fig. 5h–j). Each of these factors is necessary for normal haematopoietic[35] or vascular regeneration[45,46]. In contrast to what we observed in the bone marrow, the levels of NGF, SCF, VEGF and Ang2 in the blood did not significantly differ before versus after irradiation, or between $Lepr^{Cre/+}$; $Ngf^{fl/\Delta}$ and littermate control mice (Extended Data Fig. 6h–k).

To test if stromal cells regenerated immediately adjacent to nerve fibres or throughout the bone marrow, we assessed the distances of LepR$^+$ cells, $Scf$–GFP$^+$ stromal cells and $Scf$–GFP$^+$ adipocytes to nerve fibres in non-irradiated mice and mice 14 days after irradiation. In both cases, most LepR$^+$ cells, $Scf$–GFP$^+$ stromal cells and $Scf$–GFP$^+$ adipocytes were distant from nerve fibres and the percentages of cells in each cell population that were at least 20 μm from nerve fibres did not significantly change between non-irradiated and irradiated mice (Extended Data Fig. 6l–n). This suggested that nerve fibres promoted regeneration throughout the bone marrow, not just immediately adjacent to nerve fibres. On the other hand, the percentages of LepR$^+$ cells and $Scf$–GFP$^+$ stromal cells that were within 10 μm of nerve fibres were significantly higher in irradiated as compared with non-irradiated mice. Thus, regeneration may have been somewhat enhanced immediately adjacent to nerve fibres.

We also used $Adiponectin$–creER to delete $Ngf$ from LepR$^+$ cells and the adipocytes they gave rise to after irradiation. Nearly all LepR$^+$ cells and adipocytes express $Adiponectin$ and recombine with $Adiponectin$–creER, including the skeletal stem cells in adult bone marrow[34]. Tamoxifen was administered to $Adiponectin$–creER; $Ngf^{fl/\Delta}$ and littermate control mice at 2–3 months of age, then 2 weeks later the mice were irradiated and transplanted with a radioprotective dose of 1,000,000 WBM cells. At this early timepoint, these mice still had normal numbers of nerve fibres in the bone marrow but they did not exhibit the increase in nerve fibres after irradiation that was observed in control bone marrow (Extended Data Fig. 7a). The $Adiponectin$–creER; $Ngf^{fl/\Delta}$ mice also exhibited impaired regeneration of haematopoietic cells (Extended Data Fig. 7b–d), increased vascular leakiness (Extended Data Fig. 7e), reduced numbers of LepR$^+$ cells and endothelial cells (Extended Data Fig. 7f,g) and lower levels of bone marrow SCF, VEGF and Ang2 (Extended Data Fig. 7h–j) after irradiation. Thus, 2–3-month-old $Adiponectin$–creER; $Ngf^{fl/\Delta}$ mice phenocopied 4–5-month-old $Lepr^{Cre/+}$; $Ngf^{fl/\Delta}$ mice with reduced nerve fibre sprouting as well as impaired haematopoietic and vascular regeneration as compared with control mice.

## Nerves activate β adrenergic receptors in LepR$^+$ cells

Sympathetic nerves in the bone marrow release adrenergic neurotransmitters that promote haematopoietic regeneration by activating β2 and β3 adrenergic receptors[4]. These receptors also modulate myelopoiesis and megakaryopoiesis in ageing bone marrow[14,15]. Consistent with these results, when we administered salbutamol, a β2 agonist, to $Lepr^{Cre/+}$; $Ngf^{fl/\Delta}$ mice that lacked bone marrow nerve fibres, it rescued the regeneration of bone marrow cellularity (Fig. 5k), HSCs (Fig. 5l), LSK cells (Fig. 5m), vasculature (Fig. 5n), LepR$^+$ cells (Extended Data Fig. 7k) and endothelial cells (Extended Data Fig. 7l).

β adrenergic receptors signal through protein kinase A (PKA) to increase the expression of VEGF by cancer cells[47]. Consistent with this, LepR$^+$ cells from $Lepr^{Cre/+}$; $Ngf^{fl/\Delta}$ mice had lower levels of phosphorylated PKA (Fig. 5o) and reduced levels of SCF, VEGF and Ang2 as compared with LepR$^+$ cells from control mice at 14 days after irradiation (Extended Data Fig. 7m–o). Treatment of irradiated mice with salbutamol rescued PKA phosphorylation in LepR$^+$ cells from $Lepr^{Cre/+}$; $Ngf^{fl/\Delta}$ mice (Fig. 5o) as well as SCF, VEGF and Ang2 levels in the bone marrow (Extended Data Fig. 7m–o). These data suggest that β adrenergic receptors in LepR$^+$ cells increase the expression of growth factors by promoting PKA signalling.

Single-cell RNA sequencing[29] showed that $Adrb1$ was not expressed by bone marrow stromal cells (Extended Data Fig. 8a). $Adrb2$ and $Adrb3$ were mainly expressed by LepR$^+$ cells in the bone marrow (Extended Data Fig. 8b,c). By qRT–PCR, we did not detect $Adrb2$ and $Adrb3$ expression in unfractionated bone marrow cells but found that $Adrb2$ and $Adrb3$ were expressed at similar levels in LepR$^+$ cells and adipocytes (Extended Data Fig. 8d,e). Deficiency for either of these receptors did not significantly impair haematopoietic regeneration (Fig. 6a–c); however, deficiency for $Adrb2$ and $Adrb3$ did significantly impair haematopoietic regeneration (Fig. 6a–c). Therefore, β2 and β3 adrenergic receptors both promote haematopoietic regeneration while β1 adrenergic receptor is dispensable.

---

**Fig. 5 | NGF from LepR$^+$ cells and adipocytes promotes nerve sprouting after irradiation, increasing the expression of regeneration factors. a**, NGF in bone marrow serum from 6–8-month-old $Lepr^{cre/+}$; $Ngf^{fl/\Delta}$ and $Ngf^{fl/\Delta}$ littermate controls before ($n = 3$ mice per genotype), or 14 ($n = 4$) or 28 ($n = 4$) days after irradiation and transplantation of radioprotective wild-type bone marrow cells (three to four independent experiments per timepoint). D, day. **b**, Perilipin$^+$ adipocytes in a 30-μm-thick section from $Ngf^{mScarlet/+}$ femur bone marrow were positive for $Ngf$–mScarlet (representative of three experiments). There are also $Ngf$-expressing LepR$^+$ cells in this image (Extended Data Fig. 6a). **c**, Flow cytometric analysis of enzymatically dissociated bone marrow from $Ngf^{mScarlet/+}$ mice 14 days after irradiation (four mice from four independent experiments). **d**, $Ngf$ expression by qRT–PCR (three mice (WBM or LepR$^+$ cells) or five mice (adipocytes) from three independent experiments). **e**, The area occupied by peripherin$^+$ nerve fibres in bone marrow sections from 6–8-month-old $Lepr^{cre/+}$; $Ngf^{fl/\Delta}$ and littermate control mice (five mice per genotype per timepoint from five independent experiments). **f,g**, Representative images (**f**) and quantification (**g**) showing that nerve fibres (red) in femur bone marrow sections from 6–8-month-old $Wnt1$–$Cre$; $Rosa26^{tdTomato}$ mice before or at 14 or 28 days after irradiation (five mice per timepoint from five independent experiments). **h–j**, SCF (**h**), VEGF (**i**) and Ang2 (**j**) in bone marrow serum from 6–8-month-old $Lepr^{cre/+}$; $Ngf^{fl/\Delta}$ and littermate control mice (eight mice per genotype per timepoint from eight independent experiments). **k–n**, The β2 agonist salbutamol (Salb.) rescued the regeneration of bone marrow cellularity ($n = 6$) (**k**) and the numbers of HSCs ($n = 6$ mice per treatment) (**l**) and LSK cells ($n = 6$) (**m**) as well as the patency of the vasculature ($n = 5$) (**n**) in 6–8-month-old $Lepr^{cre/+}$; $Ngf^{fl/\Delta}$ mice at 28 days after irradiation (five to six independent experiments). **o**, Western blot of protein from LepR$^+$ cells isolated from $Lepr^{cre/+}$; $Ngf^{fl/\Delta}$ and littermate control mice at 14 days after irradiation and transplantation (representative of three independent experiments). All data represent mean ± standard deviation. Statistical significance was assessed using two-way ANOVAs followed by Tukey's (**a** and **h**) or Šidák's (**k–n**) multiple comparisons adjustments, Mann–Whitney (**e**) or Student's $t$-tests (**i** and **j**) followed by Holm–Šidák's multiple comparisons adjustments for comparisons between mutants and controls, or one-way ANOVAs (**e**, **g**, **i** and **j**) followed by Šidák's multiple comparisons adjustments for comparisons between timepoints. All statistical tests were two-sided. Not significant (NS): $P > 0.05$.

To identify the cells in which β2/β3 adrenergic receptors signal to promote haematopoietic regeneration, we made floxed alleles of *Adrb2* and *Adrb3* (Extended Data Fig. 8i–l) and conditionally deleted *Adrb2* and *Adrb3* from LepR⁺ cells using *Lepr^Cre^*. Two-month-old *Lepr^Cre/+^*; *Adrb2^fl/fl^*; *Adrb3^fl/fl^* mice had normal vasculature and haematopoiesis under steady-state conditions, including normal blood cell counts (Fig. 6d–f),

and frequencies of HSCs and restricted progenitors (Fig. 6g) in the bone marrow. However, when these mice were lethally irradiated and transplanted with radioprotective wild-type bone marrow cells, they were less likely to survive (Fig. 6h), and exhibited impaired regeneration of bone marrow cellularity (Fig. 6i), HSCs (Fig. 6j), LSK cells (Fig. 6k), vasculature (Fig. 6l), LepR⁺ cells (Fig. 6m) and endothelial cells (Fig.

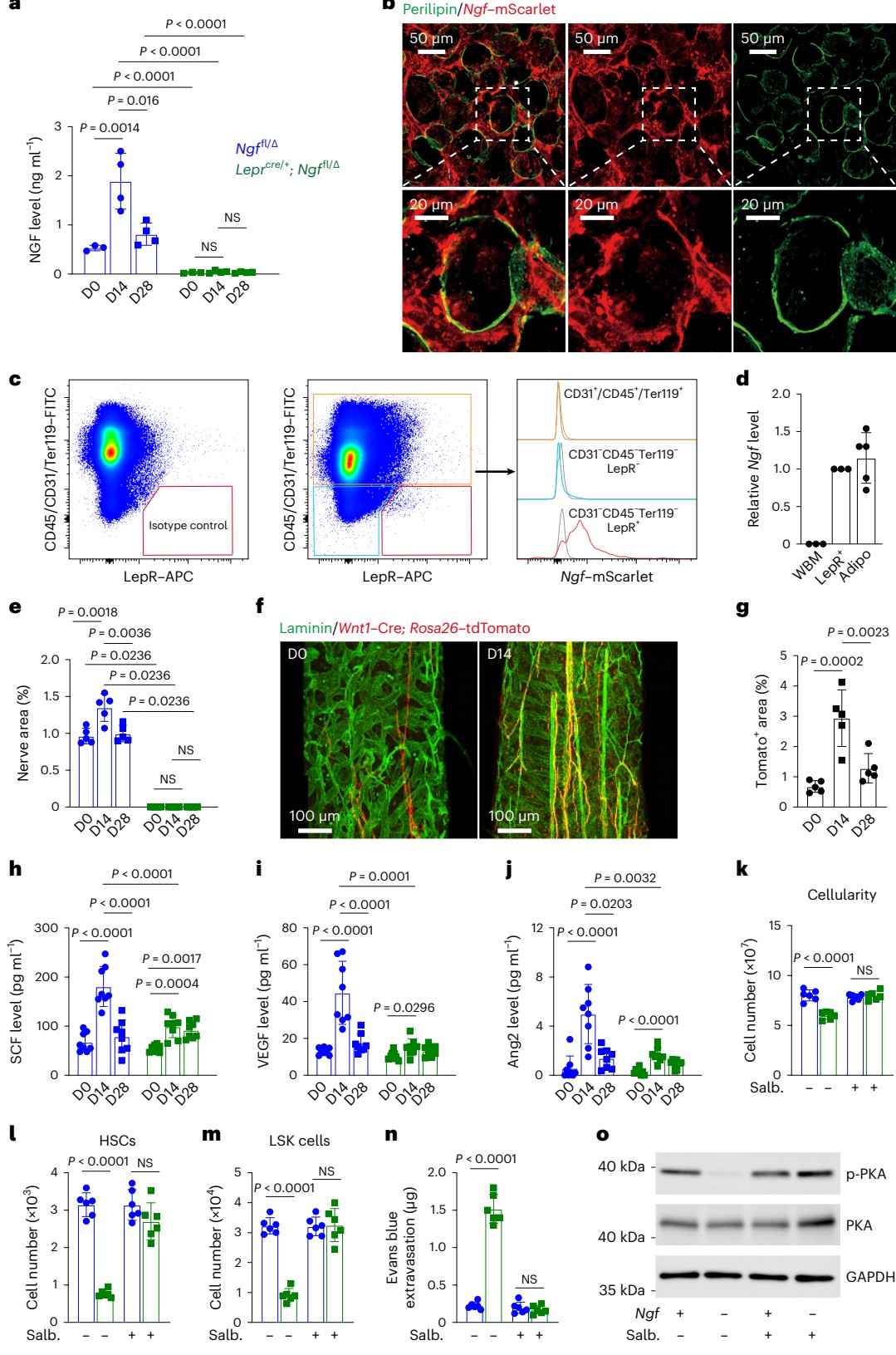

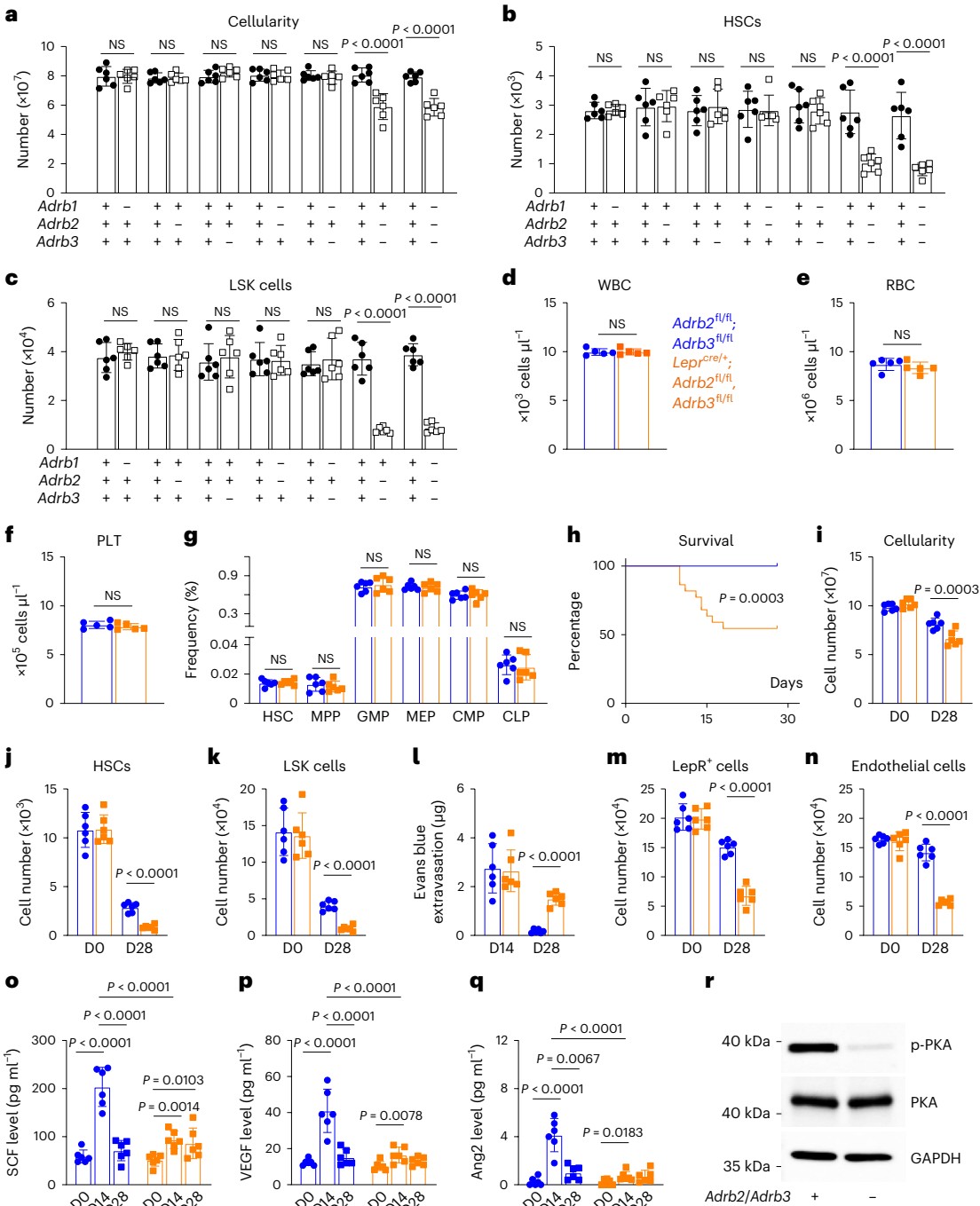

**Fig. 6 | Nerve sprouting after irradiation increases the expression of regeneration factors by activating β adrenergic receptors in LepR⁺ cells and their progeny. a–c**, Two- to 4-month-old mice, with *adrb1*, *adrb2* and/ or *adrb3* deficiency, were irradiated and transplanted with radioprotective wild-type bone marrow cells; then the cellularity (**a**) and numbers of HSCs (**b**) and LSK cells (**c**) in the bone marrow (always one tibia and one femur) were analysed 28 days later. **d–f**, White blood cell (**d**), red blood cell (**e**) and platelet (**f**) counts from non-irradiated 2-month-old *Lepr^cre/+^;Adrb2^fl/fl^;Adrb3^fl/fl^* mice and *Adrb2^fl/fl^;Adrb3^fl/fl^* littermate controls (a total of five mice per genotype from five independent experiments). **g**, Haematopoietic stem and progenitor cell frequencies in the bone marrow of non-irradiated 2-month-old *Lepr^cre/+^;Adrb2^fl/fl^; Adrb3^fl/fl^* and littermate controls. **h**, Survival of 2-month-old *Lepr^cre/+^;Adrb2^fl/fl^; Adrb3^fl/fl^* and littermate controls after irradiation and transplantation (22 mice per genotype in 3 independent experiments). **i–l**, Cellularity (**i**) and numbers of HSCs (**j**) and LSK cells (**k**) in the bone marrow and leakage of intravenously injected Evans blue dye into femur bone marrow (**l**) of 2-month-old *Lepr^cre/+^; Adrb2^fl/fl^, Adrb3^fl/fl^* and littermate controls at 14 or 28 days after irradiation. D, day.

**m,n**, Numbers of LepR⁺ cells (**m**) and endothelial cells (**n**) in the bone marrow 28 days after irradiation. **o–q**, SCF (**o**), VEGF (**p**) and Ang2 (**q**) in bone marrow serum from 2-month-old *Lepr^cre/+^;Adrb2^fl/fl^, Adrb3^fl/fl^* mice and littermate controls before (D0) or 14 or 28 days after irradiation (a total of six mice per genotype per timepoint from six independent experiments in **a–c**, **g** and **i–q**). All data represent mean ± standard deviation. **r**, Western blot of protein from LepR⁺ cells isolated from *Lepr^cre/+^;Adrb2^fl/fl^, Adrb3^fl/fl^* and littermate control mice 14 days after irradiation (representative of three independent experiments). All data represent mean ± standard deviation. The statistical significance of differences among treatments was assessed using two-way ANOVAs (**a–c** and **i–n**) or matched samples two-way ANOVAs (**d–f**) followed by Šidák's multiple comparisons adjustment, Student's *t*-tests followed by Holm–Šidák's multiple comparisons adjustments for comparisons among genotypes (**g** and **q**), a log-rank test (**h**), two-way ANOVAs (**o** and **p**) followed by Tukey's multiple comparisons adjustment, or a one-way ANOVA followed by Sidak's multiple comparisons adjustment for comparisons among timepoints (**q**). All statistical tests were two-sided. Not significant (NS): *P* > 0.05.

6n) as compared with littermate controls. *Lepr*<sup>Cre/+</sup>; *Adrb2*<sup>fl/fl</sup>; *Adrb3*<sup>fl/fl</sup> mice also exhibited significantly reduced levels of SCF (Fig. 6o), VEGF (Fig. 6p) and Ang2 (Fig. 6q) in the bone marrow as compared to littermate controls, as well as reduced PKA phosphorylation in LepR[+] cells (Fig. 6r) at 14 days after irradiation. Adipocytes in the bone marrow 14 days after irradiation and transplantation expressed levels of *Scf*, *Vegf* and *Ang2* that were comparable to LepR[+] cells (Extended Data Fig. 8f–h). Bone marrow nerve fibres thus promote regeneration by activating β2/β3 adrenergic receptors in LepR[+] cells, and potentially their adipocyte progeny, increasing the production of growth factors by these cells.

## Discussion

Nerve fibres in the bone marrow are known to promote haematopoietic regeneration after myeloablation[4], but little is known about the mechanism by which nerve fibres are maintained in the bone marrow or how they promote regeneration. Our results reveal a reciprocal relationship between LepR[+] stromal cells and nerve fibres in which nerve fibres are maintained by NGF produced by LepR[+] cells and, in turn, promote haematopoietic and vascular regeneration by secreting adrenergic neurotransmitters that activate β2/β3 adrenergic receptors in LepR[+] cells (see model in Extended Data Fig. 9). Adrenergic receptor activation in LepR[+] cells increases the production of multiple growth factors by LepR[+] cells, and the adipocytes they give rise to, that promote haematopoietic and vascular regeneration. LepR[+] cells, and the adipocytes they give rise to after myeloablation, are the major sources of SCF and VEGF for haematopoietic and vascular regeneration in the bone marrow[24,35].

Some prior studies of the effects of peripheral nerves on haematopoiesis depended on systemic ablation of sympathetic nerve fibres, such as with 6-hydroxydopamine, raising the question of whether the observed effects reflected local loss of nerve fibres within the bone marrow or more systemic effects. These studies showed that sympathetic nerves regulate circadian variation in HSC mobilization into the blood, as well as the effects of G-CSF on mobilization, by influencing the expression of CXCL12 by stromal cells[6,11]. Nociceptive nerve fibres promote HSC mobilization by releasing calcitonin gene-related peptide, which activates receptors expressed by HSCs[8]. In agreement with these studies[6,11], we observed that circadian mobilization of haematopoietic progenitors is dependent on bone marrow innervation (Fig. 2h,i), suggesting that this reflects a local effect within the bone marrow.

Nerve fibres localize exclusively around arterioles in both irradiated and non-irradiated mice (Extended Data Fig. 6b) but appear to promote the regeneration of LepR[+] cells throughout the bone marrow. The ability of nerve fibres to promote regeneration throughout the bone marrow may be enhanced by the sprouting of nerve fibres after myeloablation. Peripheral nerves also release adrenergic neurotransmitters through non-synaptic volume transmission in which neurotransmitters can diffuse considerable distances away from nerve fibres[48,49]. LepR[+] cells have long processes that allow them to interact with cells that are not adjacent to the LepR[+] cell body. Thus, nerve sprouting, volume transmission, long LepR[+] cell processes and perhaps other mechanisms that propagate signals among LepR[+] cells may enable nerve fibres around arterioles to promote regeneration throughout the bone marrow.

Peripheral nerve fibres are depleted by chemotherapy[50] and in diabetes mellitus[51]. Our results raise the question of whether peripheral neuropathy undermines engraftment in people who receive bone marrow, or other forms of HSC, transplants. Moreover, many people take drugs that block the signalling of β2/β3 adrenergic receptors, such as for heart conditions. Our results raise the question of whether β blockers delay haematopoietic regeneration after transplantation. Another interesting question for future studies is whether nerve fibres also promote the regeneration of non-haematopoietic tissues by sprouting after injury and by promoting the β adrenergic receptor-mediated expression of regeneration factors.

## Online content

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

## Methods

### Mice

All mouse experiments complied with all relevant ethical regulations and were performed according to protocols approved by the Institutional Animal Care and Use Committee at UT Southwestern Medical Center (protocol 2017-101896) and the National Institute of Biological Sciences, Beijing (NIBS2022M0024). All mice were maintained on a C57BL/6 background, including *Lepr^cre* (ref. [52]), *Adiponectin*–CreER (ref. [53]), *NG2*–DsRed (ref. [54]), *NG2*–CreER (ref. [55]), *Col1a1*–CreER (ref. [56]), *GFAP*–Cre (ref. [57]), *Rosa26–CAG–loxp–stop–loxp–tdTomato* (Ai14; ref. [58]), *Rosa26–CAG–loxp–stop–loxp–EGFP* (Ai47; ref. [59]), *Col1a1*2.3*–EGFP (ref. [60]), *Scf^GFP* (ref. [16]), *Adrb1* null (ref. [61]), *Adrb2* null (ref. [61]), *Adrb3* null (ref. [62]) and *Ngf* null mice (ref. [63]).

To generate *Ngf^mScarlet* mice, CleanCap Cas9 messenger RNA (TriLink) and single guide RNAs (transcribed using MEGAshortscript Kit (Ambion) and purified using the MEGAclear Kit (Ambion)), and recombineering plasmids were microinjected into C57BL/Ka zygotes. The coding sequence for the monomeric red fluorescent protein (*mScarlet*) was as described[64]. Chimeric mice were genotyped by restriction fragment length polymorphism analysis and insertion of the *mScarlet* sequence into the correct locus was confirmed by Southern blotting and sequencing of the targeted allele. Founders were mated with C57BL/Ka mice to obtain germline transmission then backcrossed with wild-type C57BL/Ka mice for at least three generations before analysis.

To generate the *Ngf* floxed allele, the targeting vector was obtained from The European Conditional Mouse Mutagenesis Program, linearized, and electroporated into C57BL-derived Bruce4 ES cells. Successfully targeted clones were expanded in culture then injected into C57BL/6-Tyrc-2J blastocysts. Chimeric mice were bred with C57BL/Ka mice to obtain germline transmission. The LacZ and neocassette was removed by mating with Flpe mice[65] and backcrossed for five generations onto a C57BL/Ka background before analysis.

### Genotyping primers

Primers for genotyping *Ngf^mScarlet* mice were 5′-GTG TTC TAC TTT GGG TAT TGA ATC C, 5′-CTC CAA CCC ACA CAC TGA CAC TGT C, 5′-GCT TAT AGT AGT CGG GGA TGT CGG C and 5′-CAC TGT GAA AAG ACA GAA GGC ACA ACT AGA G. Primers for genotyping *Ngf^flox* mice were 5′-CTT GTT TTC CAT CAT AGA GTT GGC TTG TT, 5′-CTT ACC TCA CTG CGG CCA GTA TA. Primers for genotyping *Adrb2^flox* mice were 5′-ACT GCT CCA AGA AGC AGA CTC TG, 5′-GTC GTT GTC ATC ATC ATC ACT GTG. Primers for genotyping *Adrb3^flox* mice were 5′-AAG ATG TAG ATG GGG GTG CGG TG, 5′-AAA CTA GAG GCG ACC AGA GAG GTC AG.

### Flow cytometry

Bone marrow haematopoietic cells were isolated by flushing the long bones using Ca²⁺- and Mg²⁺-free Hanks' Balanced Salt Solution (HBSS) (HBSS-free) with 2% bovine serum. Spleen cells were obtained by crushing the spleen between two glass slides. The cells were dissociated into a single cell suspension by gently passing them through a 25-gauge needle and then filtering through 70 μm nylon mesh. HSCs were isolated using anti-CD150 (TC15-12F12.2), anti-CD48 (HM48-1), anti-Sca1(E13-161.7) and anti-c-kit (2B8) along with the following antibodies against lineage markers: anti-Ter119, anti-B220 (6B2), anti-Gr1 (8C5), anti-CD2 (RM2-5), anti-CD3 (17A2), anti-CD5 (53-7.3) and anti-CD8 (53-6.7). Haematopoietic progenitors were isolated with the lineage markers anti-Ter119, anti-B220, anti-Gr1, anti-CD2, anti-CD3, anti-CD5 and anti-CD8 as well as additional antibodies against CD34 (RAM34), CD135 (FLT3) (A2F10), CD16/32 (FcγR) (clone 93), CD127 (IL7Ra) (A7R34), CD43 (1B11), CD24 (M1/69), IgM (II/41), CD44 (IM7), CD41 (MWReg30), CD105 (MJ7/18), CD11b (M1/70), CD71 (R17217) and CD25 (PC61.5). All antibodies were used at 1:200 for flow cytometric analyses, unless otherwise specified. PE–Cyanine7 streptavidin was used at 1:500. DAPI (5 μg ml⁻¹) was used to exclude dead cells.

For flow cytometric analysis of stromal cells, WBM was flushed using HBSS-free with 2% bovine serum then enzymatically dissociated with type I collagenase (3 mg ml⁻¹), dispase (4 mg ml⁻¹) and DNase I (1 U ml⁻¹) at 37 °C for 30 min as described previously[23]. Samples were then stained with antibodies and analysed by flow cytometry. Goat-anti-LepR-biotin (AF497), BV421 streptavidin (used at 1:500), anti-CD45 (30F-11), anti-CD31 (clone 390) and anti-TER119 antibodies were used to isolate LepR⁺ stromal cells that were negative for haematopoietic and endothelial markers. For analysis of bone marrow endothelial cells, mice were intravenously injected with 10 μg per mouse of eFluor660-conjugated anti-VE-cadherin antibody (BV13, eBiosciences). Ten minutes later, the long bones were removed and bone marrow was flushed, digested and stained as above. Samples were analysed using FACSAria Fusion or FACSCanto II flow cytometers and FACSDiva (BD) or FlowJo v10.6.1 (Tree Star) software. The flow cytometry gating strategy used for the isolation of haematopoietic stem and progenitor cell populations, LepR⁺ cells and endothelial cells is shown in Extended Data Fig. 2.

### Deep imaging of half bones

Femurs were longitudinally cut in half, then stained, and imaged as described[40]. The staining solution contained 10% dimethyl sulfoxide, 0.5% IgePal630 (Sigma) and 5% donkey serum (Jackson ImmunoResearch) in phosphate-buffered saline (PBS). Half bones were stained for 3 days at room temperature with primary antibodies. Then specimens were washed three times in PBS at room temperature for 1 day and put into staining solution containing secondary antibodies for 3 days followed by a 1-day wash. Antibodies used for whole mount staining included goat-anti-LepR-biotin (AF497, used at 1:200), rabbit-anti-peripherin (Abcam, used at 1:250), goat-anti-tdTomato (LSBio, used at 1:250), rabbit-anti-mCherry (Takara, used at 1:200), goat-anti-endomucin (R&D Systems, used at 1:250), rabbit-anti-laminin (Abcam, used at 1:250), rabbit-anti-S100B (Abcam, used at 1:250), rabbit-anti-perilipin (Sigma, used at 1:1,000), chicken anti-GFP (Aves Labs, used at 1:250), anti-SMA-FITC (Sigma, used at 1:250), Cy3-conjugated AffiniPure Fab fragment donkey anti-rabbit IgG (used at 1:250), Alexa Fluor-488-conjugated AffiniPure F(ab')2 Fragment Donkey Anti-chicken IgG (Jackson ImmunoResearch, used at 1:250), Alexa Fluor 488-AffiniPure F(ab')2 Fragment Donkey Anti-Rabbit IgG (Jackson ImmunoResearch, used at 1:250) and 555-conjugated donkey anti-goat antibody (Life Technologies, used at 1:250). Images were acquired using Leica SP8 or Leica Stellaris confocal microscopes.

### qRT–PCR

For qRT–PCR, cells were flow cytometrically sorted from enzymatically dissociated bone marrow into Trizol (Invitrogen). RNA was extracted and reverse transcribed into complementary DNA using SuperScript III (Invitrogen) and random primers. Quantitative PCR was performed using a Roche LightCycler 480. The primers used for quantitative PCR analysis included mouse *Ngf*: 5′-CCA AGG ACG CAG CTT TCT ATA C-3′ and 5′-CTG CCT GTA CGC CGA TCA AAA-3′; *Actb*: 5′-GCT CTT TTC CAG CCT TCC TT-3′ and 5′-CTT CTG CAT CCT GTC AGC AA-3′.

### Irradiation and competitive reconstitution assays

Adult recipient mice were irradiated using an XRAD 320 X-ray irradiator (Precision X-Ray) or Cesium-137 Gammacell 1000 irradiator (Best Theratronics) with two doses of 540 rad at least 4 h apart (1,080 rads total). C57BL/Ka (CD45.1/CD45.2 heterozygous) mice were used as recipients. A total of 500,000 unfractionated bone marrow cells from donor (CD45.2) and competitor (CD45.1) mice were mixed and injected intravenously through the retro-orbital venous sinus. Recipient mice were bled from 4 to 16 weeks after transplantation to examine the levels of donor-derived myeloid, B and T cells in their blood. RBCs were lysed with ammonium chloride potassium buffer before antibody staining. The antibodies used to analyse donor chimerism in the blood were

anti-CD45.1 (A20), anti-CD45.2 (104), anti-Gr1 (8C5), anti-Mac1 (M1/70), anti-B220 (6B2) and anti-CD3 (KT31.1). For sublethal irradiation, mice were irradiated using a Cesium-137 Gammacell 1000 irradiator (Best Theratronics Ltd.) with one dose of 650 rads.

## Bone marrow adipocyte isolation

Bone marrow from mice at 14 days after irradiation and bone marrow transplantation was enzymatically dissociated with DNase I (200 U ml$^{-1}$), Collagenase type I (3 mg ml$^{-1}$) and Dispase (2 mg ml$^{-1}$) at 37 °C for 30 min. Centrifugation was performed at 104$g$ for 5 min at 4 °C, pelleting most cells, including haematopoietic cells and most stromal cells. The floating cells containing mostly adipocytes were transferred to a new tube, washed twice with HBSS, then lysed with Buffer RLT plus before RNA extraction using the Qiagen RNeasy Plus Micro Kit.

## Evans blue extravasation assay

As previously described[25], mice were retro-orbitally injected with 200 µl of 0.5% Evans blue in PBS and sacrificed 15 min later. Femurs and tibias were collected, crushed, and then Evans blue was eluted in 200 µl of PBS. After a brief centrifugation to pellet cells and debris, the concentration of Evans blue in the supernatant was measured using a Nanodrop spectrophotometer (Thermo Scientific) at a wavelength of 610 nm. Femurs and tibias from mice without Evans blue injection were used as negative controls.

## Statistics and reproducibility

In each type of experiment, the number of mice analysed and the number of independent experiments is indicated in the figure legend. Mice were allocated to experiments randomly and samples processed in an arbitrary order, but formal randomization techniques were not used. Data collection and analysis were not performed blind to the conditions of the experiments. Samples sizes were not pre-determined on the basis of statistical power calculations but were similar to those used in prior publications[35,36]. No data were excluded.

Before analysing the statistical significance of differences among treatments, we tested whether data were normally distributed and whether variance was similar among groups. To test for normality, we performed the Shapiro–Wilk tests when $3 \le n < 20$ or D'Agostino Omnibus tests when $n \ge 20$. To test whether variability significantly differed among groups we performed $F$-tests (for experiments with two groups) or Levene's median tests (for experiments with more than two groups). When the data significantly deviated from normality or variability significantly differed among groups, we log$_2$-transformed the data and tested again for normality and variability. If the transformed data no longer significantly deviated from normality and equal variability, we performed parametric tests on the transformed data. If log$_2$ transformation was not possible or the transformed data still significantly deviated from normality or equal variability, we performed non-parametric tests on the non-transformed data.

When data or log$_2$-transformed data were normal and equally variable, statistical analyses were performed using Student's $t$-tests (when there were two groups), one-way analyses of variance (ANOVAs) (when there were more than two groups) or two-way ANOVAs/matched samples two-way ANOVAs (when there were two or more groups with multiple cell populations, tissues or timepoints). When the data or log$_2$-transformed data were normally distributed but unequally variable, statistical analysis was performed using Welch's $t$-tests (when there were two groups). When the data or log$_2$-transformed data were abnormally distributed, statistical analysis was performed using Mann–Whitney tests (when there were two groups), Kruskal–Wallis tests (when there were more than two groups) or Friedman tests (when there were more than two groups and samples were matched). After ANOVAs, $P$ values from multiple comparisons were adjusted using Dunnett's (when there were more than two groups and comparisons

were between a control group and other groups), Šidák's (when there were more than two groups and planned comparisons) or Tukey's method (when all the pairwise comparisons were performed). After Kruskal–Wallis tests or Friedman tests, multiple comparisons were adjusted using Dunn's method. Holm–Šidák's method was used to adjust comparisons involving multiple Student's $t$-tests, Welch's $t$-tests or Mann–Whitney tests. Log-rank tests were used to assess the statistical significance of survival differences. All statistical tests were two-sided. All data represent mean ± standard deviation. Statistical tests were performed using GraphPad Prism V10.0.0.

## Reporting summary

Further information on research design is available in the Nature Portfolio Reporting Summary linked to this article.

## Data availability

Microscopy and flow cytometry data reported in this paper will be shared by the lead contacts upon reasonable request. Published microarray data that were re-analysed in this study are available in the NCBI GEO database under accession code GSE33158. Published bulk and single-cell RNA sequencing data that were re-analysed in this study are available in the NCBI BioProject database under accession codes PRJNA914703 and PRJNA835050. Source data are provided with this paper. All other data supporting the findings of this study are available from the corresponding authors on reasonable request.

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

## Acknowledgements

S.J.M. is a Howard Hughes Medical Institute (HHMI) Investigator, the Mary McDermott Cook Chair in Pediatric Genetics, the Kathryn and Gene Bishop Distinguished Chair in Pediatric Research, the director of the Hamon Laboratory for Stem Cells and Cancer, and a Cancer Prevention and Research Institute of Texas Scholar. This study was supported partly by funding from the National Institutes of Health (DK118745 to S.J.M.), the Josephine Hughes Sterling Foundation (to S.J.M.), the Kleberg Foundation (to S.J.M.), and the Moody Medical Research Institute (to S.J.M.). B.S. was supported by startup grants from the National Institute of Biological Sciences, Beijing, and the National Natural Science Foundation of China (82272563). S.F. was supported by startup grants from China Pharmaceutical University (3150140001) and the National Natural Science Foundation of China (82203653). A.T. was supported by the Leopoldina Fellowship Program (LPDS 2016-16) of the German National Academy of Sciences and the Fritz Thyssen Foundation. We thank the BioHPC high performance computing cloud at UT Southwestern Medical Center for providing computational resources and the Moody Foundation Flow Cytometry Facility. This article is subject to HHMI's Open Access to Publications policy. HHMI lab heads have previously granted a non-exclusive CC BY 4.0 licence to the public and a sublicensable licence to HHMI in their research articles. Pursuant to those licences, the author-accepted manuscript of this article can be made freely available under a CC BY 4.0 licence immediately upon publication.

## Author contributions

B.S. and S.J.M. conceived the project, designed and interpreted the experiments. B.S. performed most of the experiments, with technical assistance and discussions from C.E., A.T., J.M.U., G.M.C. and S.F. J.G.P. and M.M.M. discovered that NGF deletion in LepR⁺ cells led to bone marrow denervation. M.M.M. developed a deep imaging protocol that enables immunofluorescence analysis of nerve fibres during bone marrow regeneration. X.G. generated and characterized the *Ngf*–mScarlet allele and the β adrenergic receptor floxed mice with assistance from Y.N., M.Y., Y.Z. and J.G. N.K. performed single-cell RNA sequencing. Z.Z. performed bioinformatic and statistical analyses. B.S. and S.J.M. wrote the manuscript.

## Competing interests

The authors declare no competing interests.

## Additional information

**Extended data** is available for this paper at https://doi.org/10.1038/s41556-023-01284-9.

**Correspondence and requests for materials** should be addressed to Bo Shen or Sean J. Morrison.

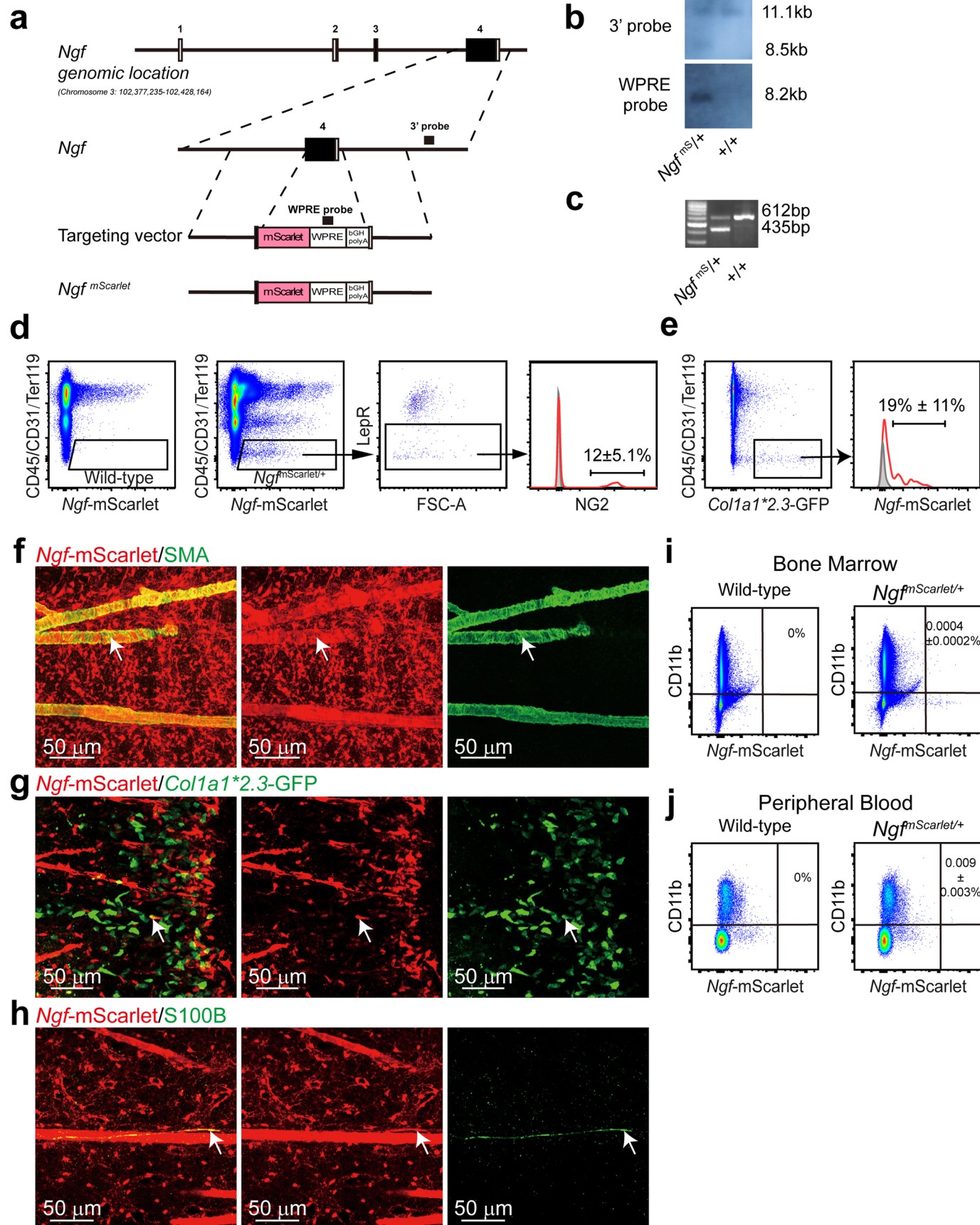

**Extended Data Fig. 1 | See next page for caption.**

**Extended Data Fig. 1 | Generation and characterization of the *Ngf*<sup>mScarlet</sup> mouse reporter allele. (a-c)** The mouse *Ngf* gene was modified by inserting an *mScarlet-WPRE-pA* cassette after an alternative ATG start codon in exon 4, replacing most of the coding sequence in exon 4. Open boxes indicate untranslated regions and black boxes indicate translated regions of *Ngf*. The correctly targeted founder mouse (F0) was identified by southern blotting (**b**) using 3′ and WPRE probes (black bars in a) (representative of three independent experiments). (**c**) PCR genotyping of genomic DNA confirmed germline transmission of the *Ngf*<sup>mScarlet</sup> allele. Mice were backcrossed at least three times onto a C57BL/Ka background before analysis (representative of three independent experiments). (**d**) Flow cytometric analysis showed that 12% of *Ngf*-mScarlet⁺ bone marrow stromal cells were NG2⁺ smooth muscle cells in enzymatically dissociated bone marrow cells. (**e**) When bones were crushed and enzymatically dissociated, 19% of osteoblasts were *Ngf*-mScarlet⁺ (4 mice from 4 independent experiments). (**f-h**). Deep imaging of femur bone marrow from 2-8 month-old *Ngf*<sup>mScarlet/+</sup> mice. The *Ngf*-mScarlet⁺ cells included SMA⁺ periarteriolar smooth muscle cells (arrow, **f**), a subset of *Col1a1*-GFP⁺ osteoblasts associated with trabecular bone in the metaphysis (arrow, g), and S100⁺ Schwann cells associated with nerve fibers in the bone marrow (arrow, **h**) (each panel reflects data from three mice from three independent experiments). As in Fig. 1, most of the *Ngf*-mScarlet⁺ cells in these images were LepR⁺ perisinusoidal stromal cells. (**i, j**) Flow cytometric analysis showed that only rare macrophages in the bone marrow (i) or blood (j) were *Ngf*-mScarlet⁺ (3 mice from 3 independent experiments).

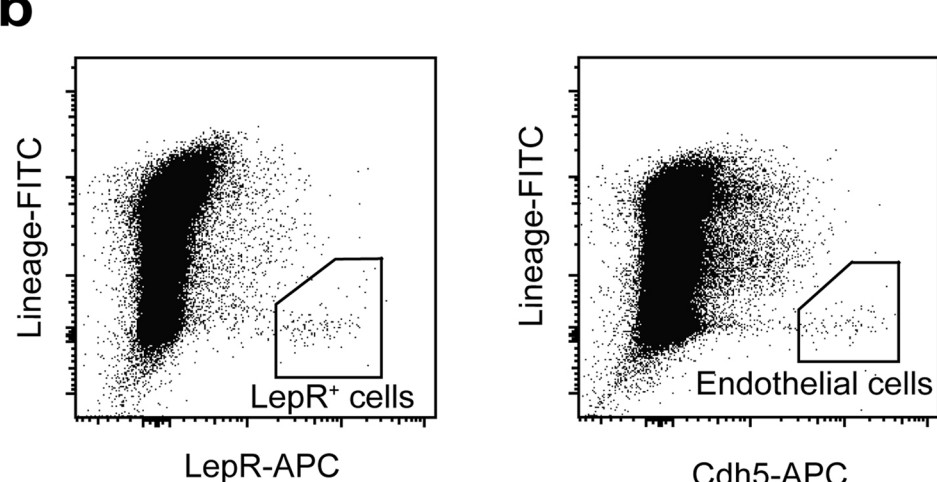

**Extended Data Fig. 2 | Flow cytometry gating strategy for the isolation of hematopoietic stem and progenitor cell populations, LepR⁺ cells and endothelial cells.** (**a**) Representative flow cytometry gates used to isolate hematopoietic stem and progenitor cell populations from bone marrow. (**b**) Representative flow cytometry gates used to isolate LepR⁺ cells and endothelial cells from bone marrow.

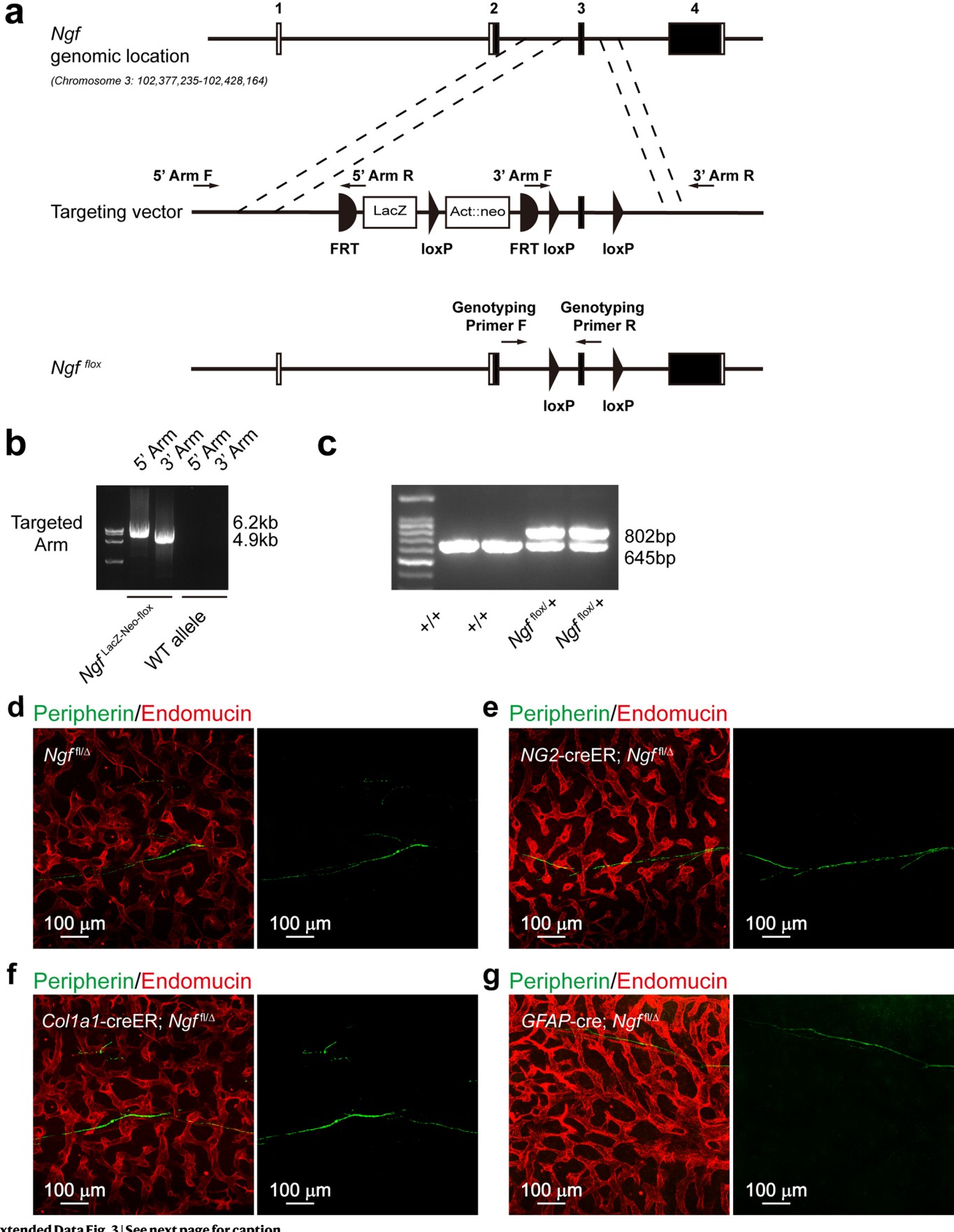

**Extended Data Fig. 3 | See next page for caption.**

**Extended Data Fig. 3 | Generation of the *Ngf^flox^* mouse allele.** (**a**) LoxP elements were inserted on either side of exon 3 of *Ngf* such that Cre-mediated recombination eliminates exon 3 and introduces a frameshift. Open boxes indicate untranslated exon sequences and black boxes indicate translated sequences. The insertion sites were selected to avoid disrupting conserved intron sequences. The *Ngf^LacZ-Neo-flox^* targeting vector was obtained from the European Conditional Mouse Mutagenesis Program (EUCOMM), linearized, and electroporated into C57BL-derived Bruce4 ES cells. Chimeric mice were generated by injecting ES clones into blastomeres and were bred with C57BL/Ka mice to obtain germline transmission of the *Ngf^LacZ-Neo-flox^* allele. (**b**) *Ngf^LacZ-Neo-flox^* mice were bred with *Flpe* mice[55] to remove the *LacZ-Neo* cassette. Successful removal of the *LacZ-Neo* cassette was confirmed by PCR primers spanning the *FRT* sites, as shown by arrows on the targeting vector in panel **a**. (**c**) PCR genotyping of genomic DNA confirmed germline transmission of the *Ngf^flox^* allele using the genotyping primers shown in panel **a**. Mice were backcrossed at least five times onto a C57BL/Ka background before analysis (images in panels **b** and **c** are representative of three independent experiments). (**d**-**g**) Deep imaging of peripherin[+] nerve fibers in the bone marrow of 6-8 month-old *Ngf*^fl/Δ^ control (**d**), *NG2*-CreER; *Ngf*^fl/Δ^ (**e**), *Col1a1*-CreER; *Ngf*^fl/Δ^ (**f**), and *GFAP*-Cre; *Ngf*^fl/Δ^ (**g**) mice. Images are representative of a total of 4 mice per genotype from 4 experiments.

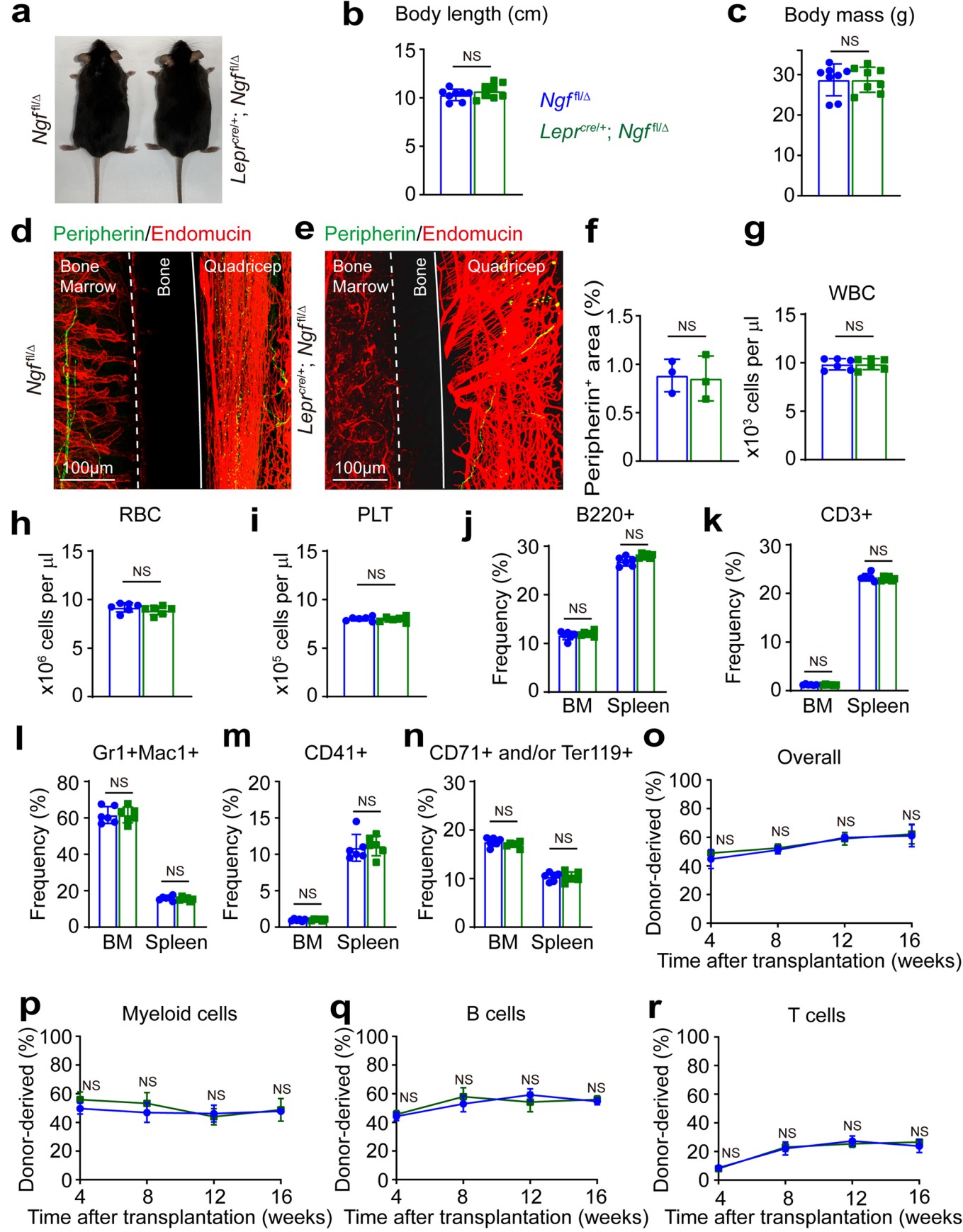

**Extended Data Fig. 4 | See next page for caption.**

**Extended Data Fig. 4 | *Lepr*^cre/+*;Ngf*^fl/Δ mice developed normally, had normal numbers of peripheral nerves outside of bones, and bone marrow innervation was dispensable for hematopoiesis in adult bone marrow.** (a-c) Six month-old *Lepr*^cre/+*;Ngf*^fl/Δ mice were grossly normal in size and appearance as compared to *Ngf*^fl/Δ littermate controls (**a**), with similar body length (**b**) and body mass (**c**) (a total of 8 mice per genotype from 8 independent experiments). (**d-f**) Immunofluorescence analysis of nerve fibers in longitudinal femur sections from 6 month-old *Lepr*^cre/+*;Ngf*^fl/Δ (**d**) and *Ngf*^fl/Δ littermate control mice (**e**), showing the presence of nerve fibers outside bone marrow in both *Lepr*^cre/+*;Ngf*^fl/Δ and control mice but nerve fibers were only present inside the bone marrow of control mice (**d**). (**f**) Nerves in the quadriceps of 6-8 month-old *Lepr*^cre/+*;Ngf*^fl/Δ and littermate control mice appeared to be comparable in numbers (3 mice per genotype from 3 independent experiments). (**g-i**) Blood cell counts in 6 month-old *Lepr*^cre/+*;Ngf*^fl/Δ mice and littermate controls. (**j-n**) 6 month-old *Lepr*^cre/+*;Ngf*^fl/Δ and littermate controls exhibited no significant differences in the frequencies of B220+ B cells (**j**), CD3+ T cells (**k**), Gr-1+Mac-1+ myeloid cells (**l**), CD41+ megakaryocyte lineage cells (**m**), or CD71+/Ter119+ erythroid lineage cells (**n**) in the bone marrow and spleen (panels **g-n** reflect a total of 6 mice per genotype in 6 independent experiments). (**o-r**) Bone marrow cells from 6-8 month-old *Lepr*^cre/+*;Ngf*^fl/Δ and littermate control mice gave similar levels of donor cell reconstitution upon competitive transplantation into irradiated mice (bone marrow cells from 5 donor mice per genotype were transplanted into a total of 5 recipients per donor in 5 independent experiments). All data represent mean ± standard deviation. Statistical significance was assessed using Student's t-tests (**b**, **c**, and **f**), t-tests followed by the Holm-Sidak's multiple comparisons adjustment (**g-n**), or matched samples two-way ANOVAs followed by the Sidak's multiple comparisons adjustment (**o-r**). All statistical tests were two-sided. Not significant (NS): P > 0.05.

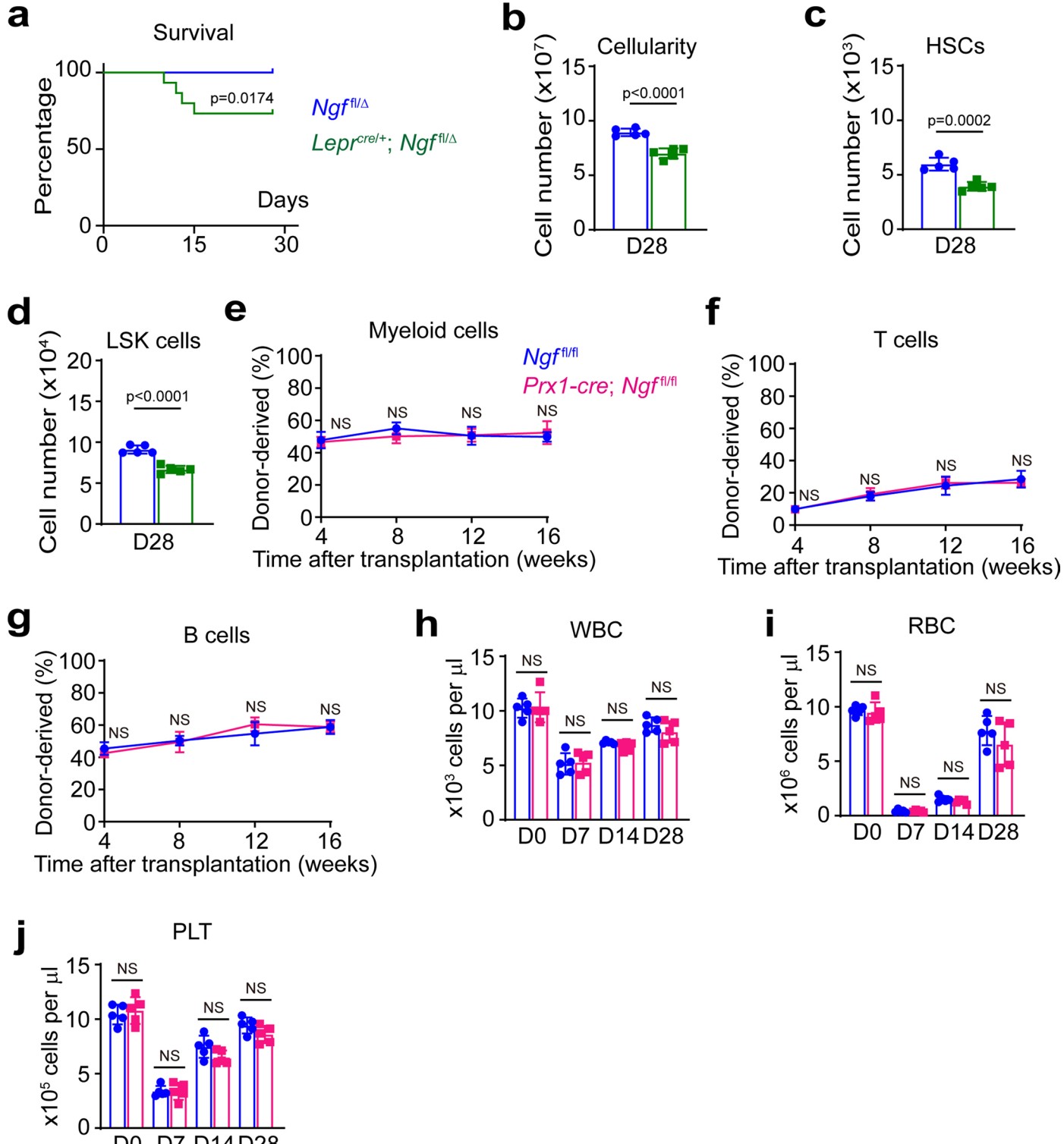

**Extended Data Fig. 5 | Nerve fibers promote hematopoietic and vascular regeneration after lethal and sublethal irradiation but are not required under steady-state conditions.** (**a**) Survival of 6-8 month-old *Lepr*[cre/+]; *Ngf*[fl/Δ] and *Ngf*[fl/Δ] littermate control mice after sublethal irradiation. (**b-d**) Cellularity (**b**), numbers of HSCs (**c**) and LSK cells (**d**) in bone marrow (one tibia and one femur) from of 6 month-old *Lepr*[cre/+]; *Ngf*[fl/Δ] and *Ngf*[fl/Δ] littermate control mice at 28 days after sublethal irradiation (5 mice per genotype from 5 independent experiments per time point). (**e-g**) Bone marrow cells from femurs of non-irradiated 2 month-old *Prx1-cre; Ngf*[fl/fl] mice and *Ngf*[fl/fl] littermate controls gave similar levels of myeloid, T cell and B cell reconstitution upon competitive transplantation into irradiated recipients (bone marrow cells from 5 donor mice were transplanted into a total of 5 recipients per donor in 5 independent experiments). (**h-j**) White blood cell (**h**), red blood cell (**i**), and platelet (**j**) counts from 2 month-old *Prx1-cre; Ngf*[fl/fl] and *Ngf*[fl/fl] littermate control mice before (D0) and 7, 14, and 28 days after irradiation and transplantation (5 mice per genotype from 5 independent experiments per time point). All data represent mean ± standard deviation. The statistical significance of differences among treatments was assessed using long-rank test (**a**), Student's t-test (**b**), t-tests followed by the Holm-Sidak's multiple comparisons adjustment (**c**, **d**, and **h-j**), or matched samples two-way ANOVAs followed by the Sidak's multiple comparisons adjustment (**e-g**). All statistical tests were two-sided. Not significant (NS): P > 0.05.

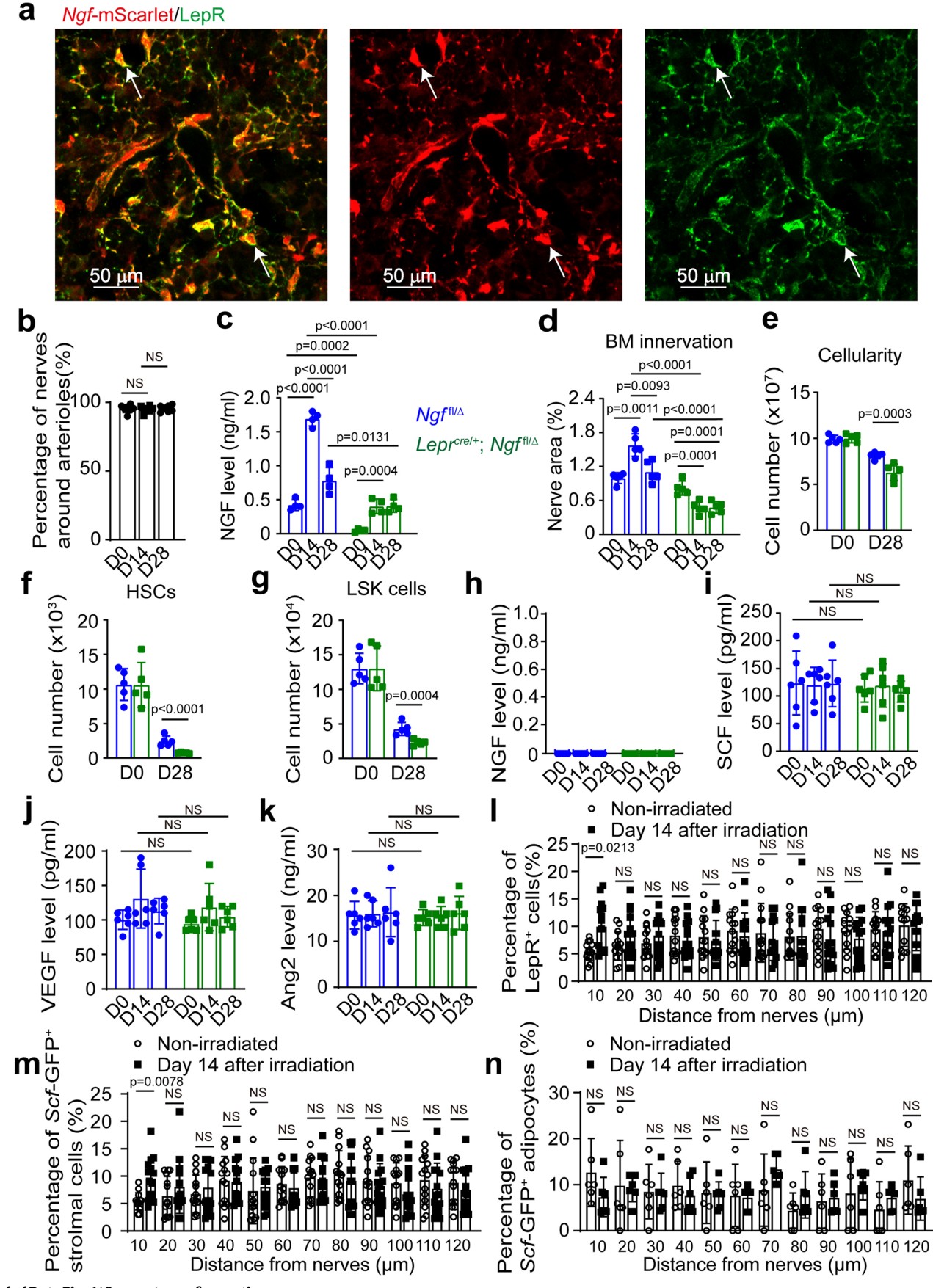

**Extended Data Fig. 6 | See next page for caption.**

**Extended Data Fig. 6 | Bone marrow nerve sprouting after myeloablation increases hematopoietic regeneration and growth factor production.**
(**a**) LepR$^+$ cells were positive for *Ngf*-mScarlet in sections from *Ngf*$^{mScarlet/+}$ femur bone marrow at 14 days after irradiation and bone marrow transplantation (representative of 3 experiments). (**b**) The percentage of nerve fibers in the bone marrow that were within 10 μm of arterioles in *Ngf*$^{fl/Δ}$ control mice before (D0), or 14 or 28 days after irradiation and transplantation (7 mice from 7 independent experiments per time point). (**c**) NGF levels in bone marrow serum from 4-5 month-old *Lepr*$^{cre/+}$;*Ngf*$^{fl/Δ}$ and *Ngf*$^{fl/Δ}$ littermate control mice before, or 14 or 28 days after irradiation and transplantation (4 mice per genotype from 4 independent experiments). (**d**) The area occupied by peripherin$^+$ nerve fibers in bone marrow sections from 4-5 month-old *Lepr*$^{cre/+}$;*Ngf*$^{fl/Δ}$ and littermate controls before, or 14 or 28 days after irradiation and transplantation. (**e-g**) Cellularity (**e**), number of HSCs (**f**) and LSK cells (**g**) in bone marrow (one tibia and one femur) from 4-5 month-old *Lepr*$^{cre/+}$;*Ngf*$^{fl/Δ}$ mice and littermate controls 28 days after irradiation and transplantation (a total of 5 mice per genotype from 5 independent experiments in panels **d-g**). (**h-k**) Blood serum levels of NGF (n = 5) (**h**), SCF (n = 6) (**i**), VEGF (n = 6) (**j**), and Ang2 (n = 6) (**k**) from 6 month-old *Lepr*$^{cre/+}$;*Ngf*$^{fl/Δ}$ mice and littermate controls before, or 14 or 28 days after irradiation and transplantation (5-6 independent experiments). (**l-n**) The distance from LepR$^+$ cells (n = 14 mice per treatment) (**l**), *Scf*-GFP$^+$ stromal cells (n = 14) (**m**), and *Scf*-GFP$^+$ adipocytes (n = 6) (**n**) to the nearest nerve fiber before or 14 days after irradiation and transplantation (6-14 independent experiments). All data represent mean ± standard deviation. The statistical significance of differences among treatments was assessed using a one-way ANOVA followed by the Sidak's multiple comparisons adjustment (**b** and **c**), Student's t-tests (**c**, **i-m**) or Mann-Whitney tests (**n**) followed by the Holm-Sidak's multiple comparisons adjustment, or two-way ANOVAs followed by the Tukey's (**d**) or Sidak's (**e-g**) multiple comparisons adjustment. All the statistical tests were two-sided. Not significant (NS): P > 0.05.

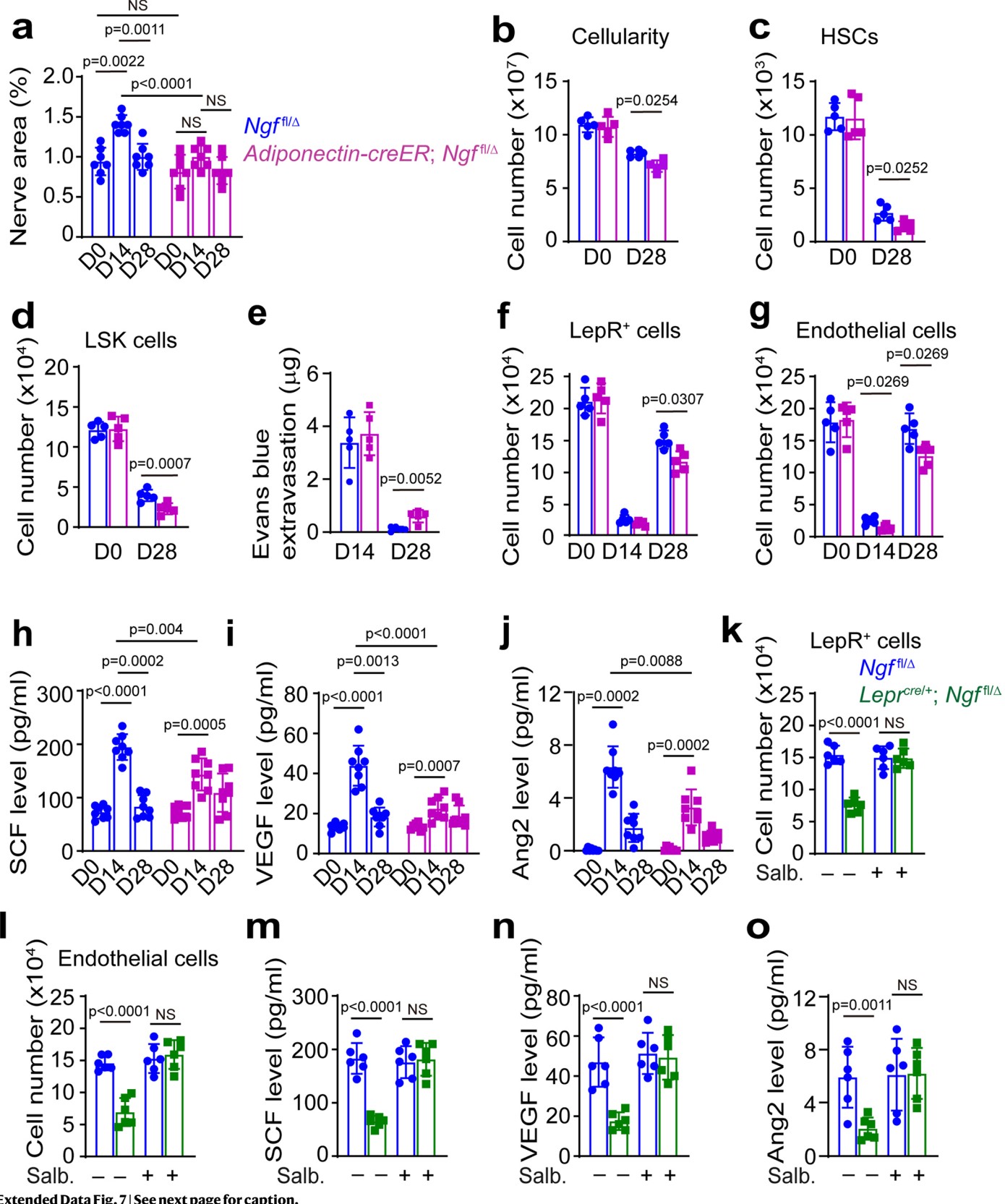

**Extended Data Fig. 7 | See next page for caption.**

**Extended Data Fig. 7 | *Adiponectin*-creER; *Ngf*<sup>fl/Δ</sup> mice exhibit defects in nerve fiber sprouting and hematopoietic and vascular regeneration after irradiation.** (**a**) The area occupied by peripherin⁺ nerve fibers in bone marrow sections from 2-3 month-old *Adiponectin*-creER; *Ngf*<sup>fl/Δ</sup> and *Ngf*<sup>fl/Δ</sup> littermate controls before, or 28 days after irradiation and transplantation (7 mice per genotype per time point from 7 independent experiments). (**b-d**) Cellularity (**b**), numbers of HSCs (**c**) and LSK cells (**d**) in bone marrow (one tibia and one femur) from *Adiponectin*-creER; *Ngf*<sup>fl/Δ</sup> and littermate control mice before and 28 days after irradiation and transplantation. (**e**) Leakage of intravenously-injected Evans blue dye into femur bone marrow at the indicated time points after irradiation and bone marrow transplantation. (**f, g**) Numbers of LepR⁺ cells (**f**) and endothelial cells (**g**) in the bone marrow before, or 14 or 28 days after irradiation and bone marrow transplantation (5 mice per genotype from 5 independent experiments in panels **b-g**). (**h - j**) SCF (**h**), VEGF (**i**) and Ang2 (**j**) in bone marrow serum from 2-3 month-old *Adiponectin*-creER; *Ngf*<sup>fl/Δ</sup> and littermate

control mice before, or 14 or 28 days after irradiation and transplantation (a total of 8 mice per genotype per time point from 8 independent experiments). (**k-o**) When administered to *Lepr*<sup>cre/+</sup>; *Ngf*<sup>fl/Δ</sup> mice after irradiation, the b2 agonist salbutamol rescued the regeneration of bone marrow LepR⁺ cells (**k**) and endothelial cells (**l**) at 28 days after irradiation, as well as bone marrow SCF (**m**), VEGF (**n**), and Ang2 (**o**) levels 14 days after irradiation (6 mice per genotype per treatment from 6 independent experiments). All data represent mean ± standard deviation. Statistical significance was assessed using matched samples two-way ANOVAs followed by the Tukey's (**a**, **h**, and **i**) or Sidak's (**b**, **d**, **f**, and **k-o**) multiple comparisons adjustment, Student's t-tests (**c**, **e**, and **g**) or Mann-Whitney tests (for genotype comparisons of **j**) followed by the Holm-Sidak's multiple comparisons adjustment, or Friedman tests followed by the Dunn's multiple comparisons adjustment (for time-point comparisons of **j**). All statistical tests were two-sided. Not significant (NS): P > 0.05.

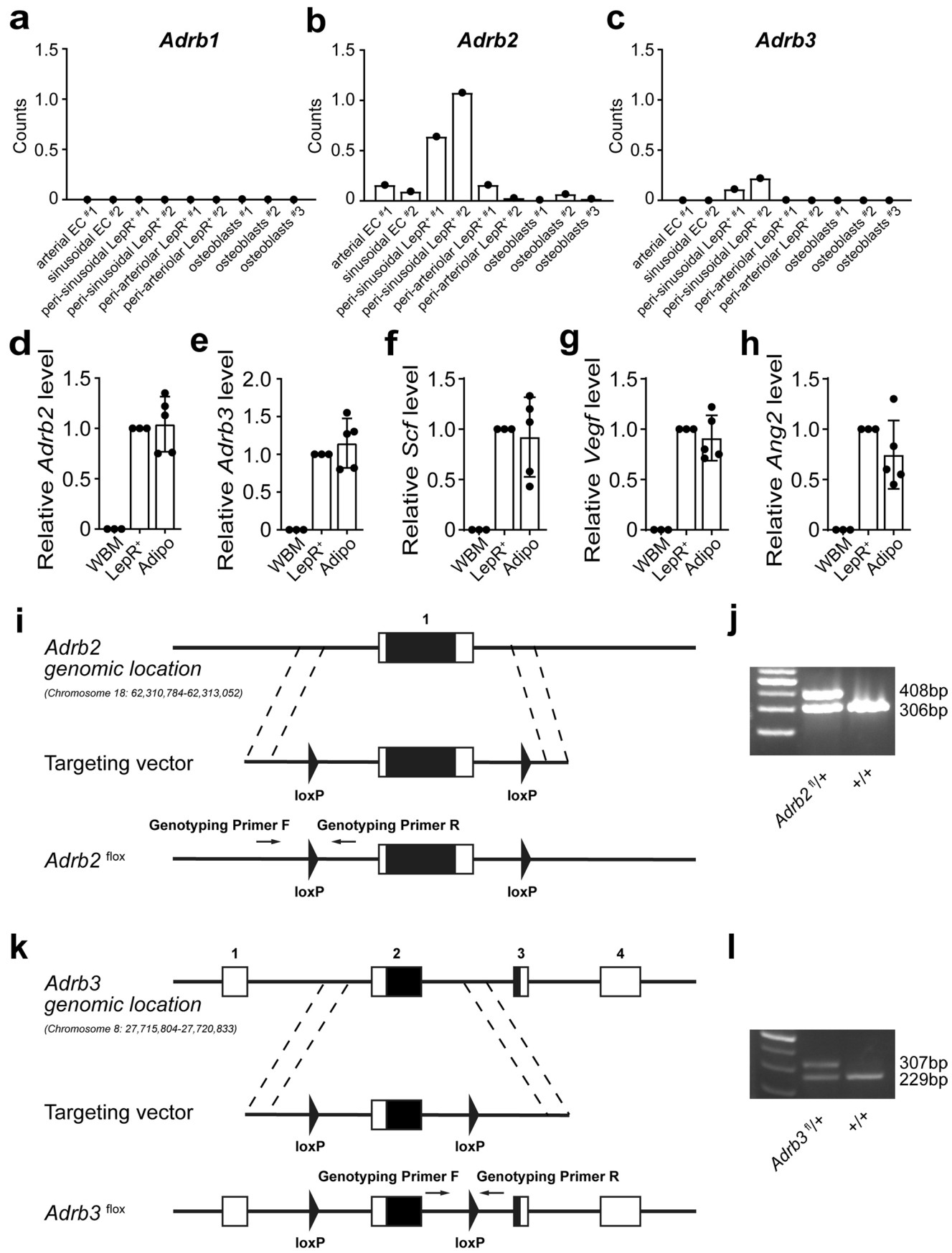

**Extended Data Fig. 8 | See next page for caption.**

**Extended Data Fig. 8 | β-adrenergic receptors are expressed by LepR⁺ stromal cells and adipocytes in the bone marrow. (a-c)** The expression patterns of *Adrb1* (**a**), *Adrb2* (**b**), and *Adrb3* (**c**) (which encode the β1, β2, and β3 adrenergic receptors, respectively) among non-hematopoietic bone marrow cells based on single cell RNA sequencing from ref. [29]. (**d-h**) *Adrb2* (**d**), *Adrb3* (**e**), *Scf* (**f**), *Vegf* (**g**), and *Ang2* (**h**) transcript levels by qRT-PCR in unfractionated cells, LepR⁺ cells, and adipocytes from adult mouse bone marrow 14 days after irradiation and transplantation of a radioprotective dose of bone marrow cells (a total of 3 mice (for unfractionated cells and LepR⁺ cells) or 5 mice (for adipocytes) from 3 independent experiments). All data represent mean ± standard deviation. (**i-l**) Generation of *Adrb2ᶠˡᵒˣ* and *Adrb3ᶠˡᵒˣ* mouse alleles. (**i**) The *Adrb2ᶠˡᵒˣ* allele was generated by inserting loxp elements on either side of exon 1. The insertion sites were chosen to avoid disrupting intron sequences that are conserved among

species. Using the *Adrb2ᶠˡᵒˣ* allele, Cre recombination removed exon 1, which contains the entire *Adrb2* coding sequence. (**j**) PCR genotyping of genomic DNA with the primers shown in panel i confirmed germline transmission of the *Adrb2ᶠˡᵒˣ* allele (representative of three independent experiments). Mice were backcrossed at least five times onto a C57BL/Ka background before analysis. (**k**) The *Adrb2ᶠˡᵒˣ* allele was generated by inserting loxp elements on either side of exon 2. The insertion sites were chosen to avoid disrupting intron sequences conserved among species. Cre recombination removed exon 2, which contains the start codon, generating a frameshift mutation. (**l**) PCR genotyping of genomic DNA with the primers shown in panel k confirmed germline transmission of the *Adrb3ᶠˡᵒˣ* allele (representative of three independent experiments for **j-l**). Mice were backcrossed at least five times onto a C57BL/Ka background before analysis. All data represent mean ± standard deviation.

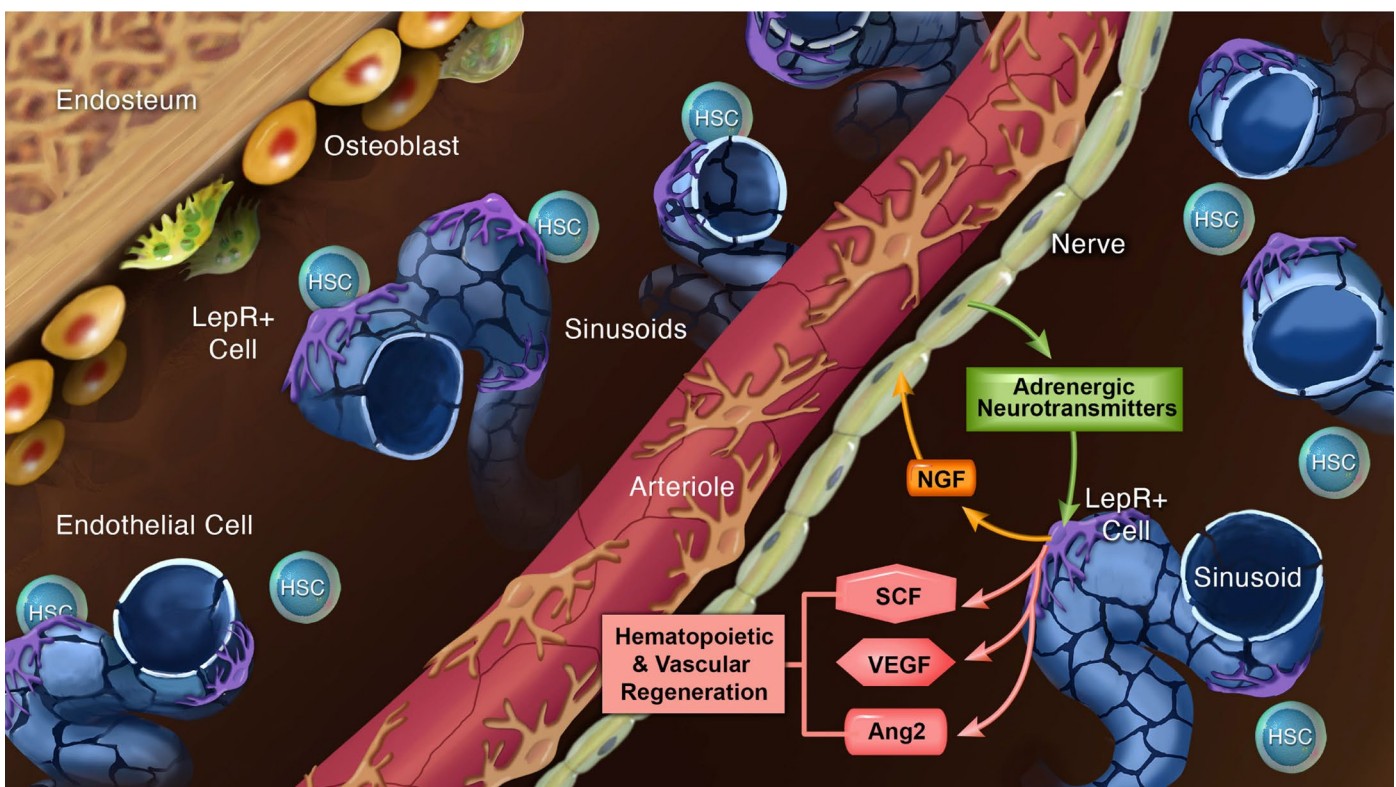

**Extended Data Fig. 9 | Schematic illustrating a reciprocal relationship between LepR⁺ stromal cells and nerve fibers in the bone marrow.** The nerve fibers are maintained by NGF produced by LepR⁺ cells and, in turn, promote hematopoietic and vascular regeneration by secreting adrenergic neurotransmitters that activate β2/β3 adrenergic receptors in LepR⁺ cells and the adipocytes they give rise to after myeloablation.

# Reporting Summary

## Statistics

For all statistical analyses, confirm that the following items are present in the figure legend, table legend, main text, or Methods section.

| n/a | Confirmed | |
|---|---|---|
| ☐ | ☒ | The exact sample size (*n*) for each experimental group/condition, given as a discrete number and unit of measurement |
| ☐ | ☒ | A statement on whether measurements were taken from distinct samples or whether the same sample was measured repeatedly |
| ☐ | ☒ | The statistical test(s) used AND whether they are one- or two-sided <br> *Only common tests should be described solely by name; describe more complex techniques in the Methods section.* |
| ☒ | ☐ | A description of all covariates tested |
| ☐ | ☒ | A description of any assumptions or corrections, such as tests of normality and adjustment for multiple comparisons |
| ☐ | ☒ | A full description of the statistical parameters including central tendency (e.g. means) or other basic estimates (e.g. regression coefficient) AND variation (e.g. standard deviation) or associated estimates of uncertainty (e.g. confidence intervals) |
| ☐ | ☒ | For null hypothesis testing, the test statistic (e.g. *F*, *t*, *r*) with confidence intervals, effect sizes, degrees of freedom and *P* value noted <br> *Give P values as exact values whenever suitable.* |
| ☒ | ☐ | For Bayesian analysis, information on the choice of priors and Markov chain Monte Carlo settings |
| ☒ | ☐ | For hierarchical and complex designs, identification of the appropriate level for tests and full reporting of outcomes |
| ☒ | ☐ | Estimates of effect sizes (e.g. Cohen's *d*, Pearson's *r*), indicating how they were calculated |

*Our web collection on statistics for biologists contains articles on many of the points above.*

## Software and code

Policy information about availability of computer code

| Data collection | Flow cytometry data were collected using BD FACSDiva 8.0. Confocal images were acquired using Zeiss Zen-2, or Leica Application Suite X 4.6.1.27508. Bulk and single-cell RNA-seq data were obtained from previous published studies. |
|---|---|
| Data analysis | GraphPad Prism V10.0 and R 4.0.2 with the stats, fBasics 4021.93, car 3.0-10, and Seurat 4.3.0 packages for bioinformatic and statistical analyses. Flow cytometry data analysis using BD FACSDiva 8.0, and FlowJo V10 (Treestar). Confocal images were processed and analysed using Zeiss Zen-2, or rendered in 3D and analysed using either Bitplane Imaris v7.7.1 or Aivia 10.5.1 software, |

For manuscripts utilizing custom algorithms or software that are central to the research but not yet described in published literature, software must be made available to editors and reviewers. We strongly encourage code deposition in a community repository (e.g. GitHub). See the Nature Portfolio guidelines for submitting code & software for further information.

## Data

Policy information about availability of data

All manuscripts must include a data availability statement. This statement should provide the following information, where applicable:
- Accession codes, unique identifiers, or web links for publicly available datasets
- A description of any restrictions on data availability
- For clinical datasets or third party data, please ensure that the statement adheres to our policy

Microscopy and flow cytometry data reported in this paper will be shared by the lead contacts upon reasonable request. Published microarray data that were re-

## Human research participants

Policy information about studies involving human research participants and Sex and Gender in Research.

| | |
|---|---|
| Reporting on sex and gender | N/A |
| Population characteristics | N/A |
| Recruitment | N/A |
| Ethics oversight | N/A |

Note that full information on the approval of the study protocol must also be provided in the manuscript.

# Field-specific reporting

Please select the one below that is the best fit for your research. If you are not sure, read the appropriate sections before making your selection.

☒ Life sciences          ☐ Behavioural & social sciences          ☐ Ecological, evolutionary & environmental sciences

For a reference copy of the document with all sections, see nature.com/documents/nr-reporting-summary-flat.pdf

# Life sciences study design

All studies must disclose on these points even when the disclosure is negative.

| | |
|---|---|
| Sample size | Samples sizes were not pre-determined based on statistical power calculations but our sample sizes are similar to those reported in previous publications (Nature 591:438, Nat Cell Biol 19:891, Nat Med 19:695). |
| Data exclusions | No data were excluded. |
| Replication | To ensure reproducibility, multiple independent biological replicates were performed with numbers of independent experiments specified in each figure legend. We did not exclude data from unsuccessful replication attempts. |
| Randomization | Mice were allocated to experiments randomly and samples processed in an arbitrary order, but formal randomization techniques were not used. |
| Blinding | No formal blinding was applied when performing the experiments or analyzing the data. Blinding was not performed during data collection or analysis. |

# Reporting for specific materials, systems and methods

We require information from authors about some types of materials, experimental systems and methods used in many studies. Here, indicate whether each material, system or method listed is relevant to your study. If you are not sure if a list item applies to your research, read the appropriate section before selecting a response.

### Materials & experimental systems

| n/a | Involved in the study |
|---|---|
| ☐ | ☒ Antibodies |
| ☒ | ☐ Eukaryotic cell lines |
| ☒ | ☐ Palaeontology and archaeology |
| ☐ | ☒ Animals and other organisms |
| ☒ | ☐ Clinical data |
| ☒ | ☐ Dual use research of concern |

### Methods

| n/a | Involved in the study |
|---|---|
| ☒ | ☐ ChIP-seq |
| ☐ | ☒ Flow cytometry |
| ☒ | ☐ MRI-based neuroimaging |

# Antibodies

**Antibodies used**

The following antibodies have been used in this study:

Anti-mouse CD2, FITC, clone RM2-5, Cat. #35-0021, LOT: C0021121118354, Tonbo, 1:200, Flow cytometry and Immunofluorescence analysis

Anti-mouse CD3, FITC, clone 17A2, Cat. #100204, LOT: B304392, BioLegend, 1:200, Flow cytometry and Immunofluorescence analysis

Anti-mouse CD5, FITC, clone 53-7.3, Cat. #100606, LOT: B210716, BioLegend, 1:200, Flow cytometry and Immunofluorescence analysis

Anti-mouse CD8a, FITC, clone 53-6.7, Cat. #35-0081, LOT: C0081100219354, Tonbo, 1:200, Flow cytometry and Immunofluorescence analysis

Anti-mouse Ter119, FITC, clone TER-119, Cat. #116206, LOT: B272256, BioLegend, 1:200, Flow cytometry and Immunofluorescence analysis

Anti-mouse B220, FITC, clone RA3-6B2, Cat. #11-0452-86, LOT: 436128, eBiosciences, 1:200, Flow cytometry and Immunofluorescence analysis

Anti-mouse Gr-1, FITC, clone RB6-8C5, Cat. #35-5931, LOT: C5931080318354, Tonbo, 1:200, Flow cytometry and Immunofluorescence analysis

Anti-mouse c-Kit, APC-eFluor780, clone 2B8, Cat. #47-1171-82, LOT: 2018834, eBiosciences, 1:200, Flow cytometry

Anti-mouse Sca-1, PerCP-Cyanine5.5, clone D7, Cat. # 45-5981-82, LOT: 2162718, EBiosciences, 1:200, Flow cytometry

Anti-mouse CD150, PE, clone TC15-12F12.1, Cat. #115904, LOT: B270365, BioLegend, 1:200, Flow cytometry

Anti-mouse CD48, Alexa Fluor-700, clone HM48-1, Cat. #103426, LOT: B279067, BioLegend, 1:200, Flow cytometry

Anti-mouse CD127, PE-Cyanine7, clone A7R34, Cat. #60-1271, LOT: C1271121219603, Tonbo, 1:200, Flow cytometry

Anti-mouse CD135, APC, clone A2F10, Cat. #135310, LOT: B269023, BioLegend, 1:200, Flow cytometry

Anti-mouse CD34, Biotin, clone RAM34, Cat. #13-0341-85, LOT: 2075112, eBiosciences, 1:200, Flow cytometry

Anti-mouse CD16/32, BV510, clone 93, Cat. #101333, LOT: B303788, BioLegend, 1:200, Flow cytometry

Anti-mouse CD41, Alexa Fluor-700, clone /MWReg30, Cat. #133926, LOT: B270215, BioLegend, 1:200, Flow cytometry

Anti-mouse CD105, APC, clone MJ7/18, Cat. #120414, LOT: B266785, BioLegend, 1:200, Flow cytometry

Anti-mouse Gr-1, PE-Cyanine7, clone RB6-8C5, Cat. #108416, LOT: B284962, BioLegend, 1:200, Flow cytometry

Anti-mouse CD11b, APC-eFluor780, clone M1/70, Cat. #47-0112-82, LOT: 2011193, EBiosciences, 1:200, Flow cytometry

Anti-mouse B220, PerCP-Cyanine5.5, clone RA3-6B2, Cat. #65-0452, LOT: C0452060619653, Tonbo, 1:200, Flow cytometry

Anti-mouse IgM, APC, clone 11/41, Cat. #17-5790-82, LOT: 2167008, eBiosciences, 1:200, Flow cytometry

Anti-mouse CD43, PE, clone S7, Cat. #553271, LOT: 9336727, Fisher Scientific, 1:200, Flow cytometry

Anti-mouse CD3, redFluor-710, clone 17A2, Cat. #80-0032, LOT: C0032010319803, Tonbo, 1:200, Flow cytometry

Anti-mouse Ter119, BV510, clone TER-119, Cat. #116237, LOT: B270148, BioLegend, 1:200, Flow cytometry

Anti-mouse CD71, FITC, clone R17217, Cat. #11-0711-82, LOT: 2159109, eBiosciences, 1:200, Flow cytometry

Anti-mouse CD31, FITC, clone 390, Cat. #11-0311-82, LOT: 2086274, eBiosciences, 1:200, Flow cytometry

Anti-mouse CD144, eFluor-660, clone eBioBV13, Cat. #50-1441-82, LOT: 2007696, eBiosciences, 10 ug/mouse, Flow cytometry

Anti-mouse Peripherin, primary antibody, Cat. #ab4666, LOT: GR3383934-9, Abcam,1:250, Immunofluorescence analysis

Anti-mouse Leptin receptor, Biotin, Cat. #BAF497, LOT: BFV0719071, R&D Systems, 1:200, Flow; 1:200, Immunofluorescence analysis

Anti-mCherry, Cat. #632496, LOT: 1904182, Takara, 1:200, Immunofluorescence analysis

Cy3-conjugated AffiniPure Fab fragment donkey anti-rabbit IgG, Cat. #711-167-003, LOT: 145173, Jackson ImmunoResearch, 1:250, Immunofluorescence analysis

Alexa Fluor-488-conjugated AffiniPure Fab fragment donkey anti-chicken IgG, Cat. #703-546-155, LOT: 144594, Jackson ImmunoResearch, 1:250, Immunofluorescence analysis

PE-Cyanine7 streptavidin, Cat. #557598, LOT: 9011715, BD Biosciences, 1:500, Flow cytometry

BV421 streptavidin, Cat. #405226, LOT: B286541, BioLegend, 1:500, Flow cytometry

Anti-Green Fluorescent Protein Antibody, Cat. #GFP-1020, LOT: GFP3717982, Aves Labs, 1:250, Immunofluorescence analysis

Donkey anti-Goat IgG (H+L) Cross-Adsorbed Secondary Antibody, Alexa Fluor-555, Cat. #A-21432, LOT:2400919, Life Technologies, 1:500, Immunofluorescence analysis

Anti-tdTomato, Cat. #LS-C340696, LOT: 200314, LSBio, 1:500, Immunofluorescence analysis

Anti-Endomucin, Cat. #AF4666, LOT: CAAS0222041, R&D Systems, 1:250, Immunofluorescence analysis

Anti-Laminin, Cat. #ab7463, LOT: GR3408983-4, 1:250, Immunofluorescence analysis

Anti-S100 beta, Cat. #ab52642, LOT: GR3215095-26, 1:250, Immunofluorescence analysis

Anti-Perilipin, Cat.#P1873, LOT: 0000149699, 1:1000, Immunofluorescence analysis

Anti-Actin, α-Smooth Muscle-FITC antibody, Cat: F3777, LOT: 0000122364, 1:250, Immunofluorescence analysis

Alexa Fluor-488 AffiniPure F(ab')2 Fragment Donkey Anti-Rabbit IgG (H+L). Cat.#711-546-152. LOT: 161669, 1:250, Immunofluorescence analysis

Validation

All antibodies are commercially available and have been validated in previously published studies (e.g. Nature 495:231, Nature 591:438). We have independently validated antibodies that were central to our conclusions. For instance, the anti-LepR antibody was validated using mouse bone marrow cells deficient for Lepr (Cell Stem Cell 15:154).

Anti-mouse CD2. This monoclonal antibody recognizes mouse CD2. https://tonbobio.com/products/fitc-anti-mouse-cd2-rm2-5

Anti-mouse CD3. This monoclonal antibody recognizes mouse CD3. https://www.biolegend.com/en-us/products/fitc-anti-mouse-cd3-antibody-45

Anti-mouse CD5. This monoclonal antibody recognizes mouse CD5. https://www.biolegend.com/en-us/products/fitc-anti-mouse-cd5-antibody-159

Anti-mouse CD8a. This monoclonal antibody recognizes mouse CD8. https://tonbobio.com/products/fitc-anti-mouse-cd8a-53-6-7

Anti-mouse Ter119. This monoclonal antibody recognizes mouse Ter119. https://www.biolegend.com/en-us/products/fitc-anti-mouse-ter-119-erythroid-cells-antibody-1865

Anti-mouse B220. This monoclonal antibody recognizes mouse B220. https://www.thermofisher.com/antibody/product/CD45R-B220-Monoclonal-Antibody-RA3-6B2-FITC-eBioscience/11-0452-86

Anti-mouse Gr-1. This monoclonal antibody recognizes mouse Gr-1. https://tonbobio.com/products/fitc-anti-mouse-ly-6g-gr-1-rb6-8c5

Anti-mouse c-kit. This monoclonal antibody recognizes mouse c-kit. https://www.thermofisher.com/antibody/product/CD117-c-Kit-Antibody-clone-2B8-Monoclonal/47-1171-82

Anti-mouse Sca-1. This monoclonal antibody recognizes mouse Sca-1. https://www.thermofisher.com/antibody/product/Ly-6A-E-Sca-1-Antibody-clone-D7-Monoclonal/45-5981-82

Anti-mouse CD150. This monoclonal antibody recognizes mouse CD150. https://www.biolegend.com/en-us/products/pe-anti-mouse-cd150-slam-antibody-1369

Anti-mouse CD48. This monoclonal antibody recognizes mouse CD48. https://www.biolegend.com/en-us/products/alexa-fluor-700-anti-mouse-cd48-antibody-6670

Anti-mouse CD127. This monoclonal antibody recognizes mouse CD127. https://tonbobio.com/products/pe-cyanine7-anti-mouse-cd127-il-7ra-a7r34

Anti-mouse CD135. This monoclonal antibody recognizes mouse CD135. https://www.biolegend.com/en-us/products/apc-anti-mouse-cd135-antibody-6284

Anti-mouse CD34. This monoclonal antibody recognizes mouse CD34. https://www.thermofisher.com/antibody/product/CD34-Antibody-clone-RAM34-Monoclonal/13-0341-85

Anti-mouse CD16/32. This monoclonal antibody recognizes mouse CD16/32. https://www.biolegend.com/en-us/products/brilliant-violet-510-anti-mouse-cd16-32-antibody-9917

Anti-mouse CD41. This monoclonal antibody recognizes mouse CD41. https://www.biolegend.com/en-us/products/alexa-fluor-700-anti-mouse-cd41-antibody-13058

Anti-mouse CD105. This monoclonal antibody recognizes mouse CD105. https://www.biolegend.com/en-us/products/apc-anti-mouse-cd105-antibody-6519

Anti-mouse Gr-1. This monoclonal antibody recognizes mouse Gr-1. https://www.biolegend.com/en-us/products/pe-cy7-anti-mouse-ly-6g-ly-6c-gr-1-antibody-1931

Anti-mouse CD11b. This monoclonal antibody recognizes mouse CD11b. https://www.thermofisher.com/antibody/product/CD11b-Antibody-clone-M1-70-Monoclonal/47-0112-82

Anti-mouse B220. This monoclonal antibody recognizes mouse B220. https://tonbobio.com/products/percp-cyanine5-5-anti-human-mouse-cd45r-b220-ra3-6b2

Anti-mouse IgM. This monoclonal antibody recognizes mouse IgM. https://www.thermofisher.com/antibody/product/IgM-Antibody-clone-II-41-Monoclonal/17-5790-82

Anti-mouse CD43. This monoclonal antibody recognizes mouse CD43. https://www.fishersci.com/shop/products/cd43-rat-anti-mouse-pe-clone-s7-bd/bdb553271?matchedCatNo=BDB553271&searchHijack=true&searchTerm=BDB553271&searchType=RAPID

Anti-mouse CD3. This monoclonal antibody recognizes mouse CD3. https://tonbobio.com/products/redfluor-710-anti-mouse-cd3-17a2

Anti-mouse CD71. This monoclonal antibody recognizes mouse CD71. https://www.thermofisher.com/antibody/product/CD71-Transferrin-Receptor-Antibody-clone-R17217-RI7-217-1-4-Monoclonal/11-0711-82

Anti-mouse CD31. This monoclonal antibody recognizes mouse CD31. https://www.thermofisher.com/antibody/product/CD31-PECAM-1-Antibody-clone-390-Monoclonal/11-0311-82

Anti-mouse CD144. This monoclonal antibody recognizes mouse CD144. https://www.thermofisher.com/antibody/product/CD144-VE-cadherin-Antibody-clone-eBioBV13-BV13-Monoclonal/50-1441-82

Anti-mouse Peripherin. This polyclonal antibody recognizes mouse Peripherin. https://www.abcam.com/peripherin-antibody-ab4666.html

Anti-mouse Leptin Receptor. This monoclonal antibody recognizes mouse Leptin Receptor. https://www.rndsystems.com/products/mouse-leptin-r-biotinylated-antibody_baf497

Anti-mCherry. This polyclonal antibody recognizes tdTomato. https://www.takarabio.com/products/antibodies-and-elisa/fluorescent-protein-antibodies/red-fluorescent-protein-antibodies?catalog=632496

Cy3-conjugated AffiniPure Fab fragment donkey anti-rabbit IgG. This Fab fragment antibody was generated by papain digestion of whole IgG antibodies to remove the entire Fc portion, including the hinge region. This antibody is monovalent, containing only a single antigen binding site. Based on antigen-binding assay and/or ELISA, the antibody reacts with rabbit IgG. It also reacts with the light chains of other rabbit immunoglobulins. No binding was detected against non-immunoglobulin serum proteins. https://www.jacksonimmuno.com/catalog/products/711-167-003

Alexa Fluor-488-conjugated AffiniPure Fab fragment donkey anti-chicken IgG. This F(ab')2 fragment antibody was generated by pepsin digestion of whole IgG antibodies to remove most of the Fc region while leaving some of the hinge region. F(ab')2 fragments have two antigen-binding Fab portions linked together by disulfide bonds and therefore they are divalent. It is used for specific applications, such as to avoid binding of secondary antibodies to live cells with Fc receptors or to Protein A or Protein G. Based on immunoelectrophoresis and/or ELISA, the antibody reacts with whole molecule chicken IgY. It also reacts with the light chains of other chicken immunoglobulins. No binding was detected against non-immunoglobulin serum proteins. https://www.jacksonimmuno.com/catalog/products/703-546-155

PE-Cyanine7 streptavidin. This is a second-step reagent for the indirect immunofluorescent staining of cells in combination with biotinylated primary antibodies for flow cytometric analysis. https://www.bdbiosciences.com/us/reagents/research/antibodies-buffers/second-step-reagents/avidinstreptavidin/pe-cy7-streptavidin/p/557598

BV421 streptavidin. This is a second-step reagent for the indirect immunofluorescent staining of cells in combination with biotinylated primary antibodies for flow cytometric analysis. https://www.biolegend.com/en-us/products/brilliant-violet-421-streptavidin-7297

Anti-Green Fluorescent Protein. This polyclonal antibody recognizes mouse Peripherin. https://www.aveslabs.com/products/anti-green-fluorescent-protein-antibody-gfp

Donkey anti-Goat IgG (H+L) Cross-Adsorbed Secondary Antibody, Alexa Fluor-555. To minimize cross-reactivity, these donkey anti-goat IgG (H+L) whole secondary antibodies have been affinity purified and cross-adsorbed against rabbit, rat, mouse, and human IgG. Cross-adsorption or pre-adsorption is a purification step to increase specificity of the antibody resulting in higher sensitivity and less background staining. The secondary antibody solution is passed through a column matrix containing immobilized serum proteins from potentially cross-reactive species. Only the nonspecific-binding secondary antibodies are captured in the column, and the highly specific secondaries flow through. The benefits of this extra step are apparent in multiplexing/multicolor-staining experiments (e.g., flow cytometry) where there is potential cross-reactivity with other primary antibodies or in tissue/cell fluorescent staining experiments where there are may be the presence of endogenous immunoglobulins.
https://www.thermofisher.com/antibody/product/Donkey-anti-Goat-IgG-H-L-Cross-Adsorbed-Secondary-Antibody-Polyclonal/A-21432

Anti-tdTomato. This polyclonal antibody recognizes mouse tdtomato.
https://www.lsbio.com/antibodies/tdtomato-antibody-if-immunofluorescence-ihc-wb-western-ls-c340696/351334

Anti-Endomucin. This polyclonal antibody recognizes mouse Endomucin.
https://www.rndsystems.com/cn/products/mouse-endomucin-antibody_af4666#product-details

Anti-Laminin. This antibody is pan-specific and reacts well with all laminin isoforms tested: laminin-1 (alpha-1, beta-1, and gamma-1) and laminin-2 (alpha-2, beta-1, and gamma-1).
https://www.abcam.cn/products/primary-antibodies/laminin-12-antibody-ab7463.html

Anti-S100 beta. This polyclonal antibody recognizes mouse S100 beta.
https://www.abcam.cn/products/primary-antibodies/s100-beta-antibody-ep1576y-astrocyte-marker-ab52642.html

Anti-Perilipin. This polyclonal antibody recognizes mouse Perilipin.
https://www.sigmaaldrich.com/HK/en/product/sigma/p1873

Anti-Actin, α-Smooth Muscle-FITC. Monoclonal Anti-Actin, α-Smooth Muscle specifically recognizes the α-smooth muscle isoform of actin (42 kDa) by ELISA and immunoblotting.[2] It does not react with the other major actin isoforms present in fibroblasts or epithelial cells (β and γ-cytoplasmic), striated muscle (α-sarcomeric), myocardium (α-myocardial), or γ-smooth muscle isoform.
https://www.sigmaaldrich.com/HK/en/product/sigma/f3777

Alexa Fluor-488 AffiniPure F(ab')2 Fragment Donkey Anti-Rabbit IgG (H+L). F(ab')2 fragment antibodies are generated by pepsin digestion of whole IgG antibodies to remove most of the Fc region while leaving some of the hinge region. F(ab')2 fragments have two antigen-binding Fab portions linked together by disulfide bonds and therefore they are divalent. The average molecular weight is about 110 kDa. They are used for specific applications, such as to avoid binding of secondary antibodies to live cells with Fc receptors or to Protein A or Protein G.
https://www.jacksonimmuno.com/catalog/products/711-546-152

Anti-mouse Ter119. This monoclonal antibody recognizes mouse Ter119. https://www.biolegend.com/en-us/products/brilliant-violet-510-anti-mouse-ter-119-erythroid-cells-antibody-8243

Anti-Green Fluorescent Protein Antibody. This polyclonal antibody recognizes mouse Green Fluorescent Protein. https://www.aveslabs.com/products/anti-green-fluorescent-protein-antibody-gfp

# Animals and other research organisms

Policy information about studies involving animals; ARRIVE guidelines recommended for reporting animal research, and Sex and Gender in Research

| Laboratory animals | Lepr-cre (JAX Strain #:008320), Adiponectin-CreER (JAX Strain #:024671), NG2-DsRed (JAX Strain #:008241), NG2-CreER (JAX Strain #:008538), Col1a1-CreER (JAX Strain #:016241), GFAP-Cre (JAX Strain #:024098), Rosa26-CAG-loxp-stop-loxp-tdTomato (Ai14; JAX Strain #:007914), Rosa26-CAG-loxp-stop-loxp-EGFP (Ai47; previously published in Cell 174:465), Col1a1*2.3-EGFP (JAX Strain #:013134), ScfGFP (JAX Strain #:017860), Ngf null (previously published in Cell 76:1001), Adrb1 null (derived from JAX Strain #:003810), Adrb2 null (derived from JAX Strain #:003810), and Adrb3 null mice (JAX Strain #:006402) were previously characterized and used in this study. Ngf-mScarlet, Ngf floxed, Adrb2 floxed, and Adrb3 floxed mice were generated in this study. All mice were maintained on a C57BL/6J background, and two-month to 8-month-old mice were used. |
|---|---|
| Wild animals | No wild animals were used. |
| Reporting on sex | Random sex assignment was used in each experiment and findings apply to both sexes. |
| Field-collected samples | No field-collected samples were used. |
| Ethics oversight | All mouse experiments complied with all relevant ethical regulations and were performed according to protocols approved by the Institutional Animal Care and Use Committee at UT Southwestern Medical Center (UTSW; protocol 2017-101896) and the National Institute of Biological Sciences, Beijing (NIBS; protocol NIBS2022M0024). Mice were maintained under pathogen-free conditions of a 12 h light/dark cycle at controlled temperature (20-25°C) and humidity (50-70%), and were provided with food and water ad libitum. |

Note that full information on the approval of the study protocol must also be provided in the manuscript.

# Flow Cytometry

## Plots

Confirm that:

☒ The axis labels state the marker and fluorochrome used (e.g. CD4-FITC).

☒ The axis scales are clearly visible. Include numbers along axes only for bottom left plot of group (a 'group' is an analysis of identical markers).

☒ All plots are contour plots with outliers or pseudocolor plots.

☒ A numerical value for number of cells or percentage (with statistics) is provided.

# Methodology

**Sample preparation**

Bone marrow hematopoietic cells were isolated by flushing the long bones with Ca2+- and Mg2+- free HBSS (HBSS-free) with 2% heat-inactivated bovine serum. Spleen cells were obtained by crushing the spleen between two glass slides. The cells were dissociated into a single cell suspension by gently passing them through a 25-gauge needle and then filtering through 70-um nylon mesh.

For flow cytometric analysis of stromal cells, bone marrow was flushed using HBSS-free with 2% bovine serum. Then whole bone marrow was digested with type I collagenase (3mg/ml), dispase (4mg/ml) and DNase I (1U/ml ) at 37°C for 30 min. Samples were then stained with antibodies and analyzed by flow cytometry.

**Instrument**

BD FACS Aria Fusion (for cell sorting or analysis), BD Canto (for analysis).

**Software**

BD FACSDiva 8.0, FlowJo V10

**Cell population abundance**

The abundance of the relevant cell populations within post-sort fractions was 90-100% in experiments.

**Gating strategy**

Flow cytometry gating strategy for the isolation of hematopoietic stem and progenitor cell populations, LepR+ cells and endothelial cells were shown in Extended Data Fig. 2. To eliminate dead cells from sorts and analyses, cells were stained with 4',6-diamidino-2-phenylindole (DAPI).

☒ Tick this box to confirm that a figure exemplifying the gating strategy is provided in the Supplementary Information.

