## [Peer Review File · Nature Cell Biology]

Peer Review Information

Journal: Nature Cell Biology

Manuscript Title: Leptin Receptor+ cells promote bone marrow innervation and regeneration by synthesizing nerve growth factor

Corresponding author name(s): Professor Sean Morrison

Editorial Notes:

Transferred manuscripts This document only contains reviewer comments, rebuttal and decision letters for versions considered at Nature Cell Biology.

Reviewer Comments & Decisions:

Decision Letter, initial version:

Dear Sean,

Your manuscript, "Leptin Receptor+ cells promote bone marrow innervation and regeneration by synthesizing nerve growth factor", has now been seen by 3 referees, who are experts in bone marrow stromal cells and niche, vascular and hematopoietic regeneration (referee 1); vascular and hematopoietic regeneration, bone marrow innervation, HSC niche (referee 2); and stromal niche, HSCs, regeneration, scRNAseq (referee 3). As you will see from their comments (attached below) they find this work of potential interest, but have raised substantial concerns, which in our view would need to be addressed with considerable revisions before we can consider publication in Nature Cell Biology.

Nature Cell Biology editors discuss the referee reports in detail within the editorial team, including the chief editor, to identify key referee points that should be addressed with priority, and requests that are overruled as being beyond the scope of the current study. To guide the scope of the revisions, I have listed these points below. I should stress that the referees' concerns point to a premature dataset and these points would need to be addressed with experiments and data, and reconsideration of the study for this journal and re-engagement of referees would depend on strength of these revisions.

In particular, it would be essential to:

(A) Experimentally address issues raised about the proposed interplay between LepR+ cells/adipocytes and nerve fibers in the context of regeneration, as indicated by:

Referee #2:

"The images showing NGF distribution, in the steady-state, in the Ngf-mScarlet reporter are spectacular and very convincing. However, the data on regeneration (Fig. 4b) is less clear. It appears that Ngf-expression is widespread and not restricted to Peripelin+ adipocytes. The authors should show a counterstain for LepR+ cells to validate their claims".

"The observation that new arteries and nerves sprout after myeloablation is striking (Fig. 4f). Since the distribution of these fibers is not homogenous, do the authors observe increased LepR+ cell and hematopoietic recovery near nerve fibers?"

Referee #3:

"The authors proposed that nerve fibers produce adrenergic neurotransmitters to activate LepR+ cells, increasing the expression of regeneration factors, such as stem cell factor (SCF), Angiopoietin 2 (Ang2), and vascular endothelial growth factor (VEGF), for hematopoietic and vascular regeneration. However, it is unclear whether LepR+ cells adjacent to nerve fibers produce more regeneration factors than other LepR+ cells away from nerve fibers, as adrenergic neurotransmitters only target cells in close proximity. In fact, nerve fibers are rare and adjacent to arteries, but LepR+ cells are more broadly distributed and adjacent to sinusoids in bone marrow. Therefore, the authors should provide more information on the spatial distribution of Ang2/VEGF-expressing cells in relation to nerve fibers in bone marrow to clarify how local nerve fibers contribute to hematopoietic and vascular regeneration through LepR+ cells. Otherwise, they should consider the possibility that adrenergic neurotransmitters are circulating rather than being produced locally by nerve fibers".

"It is possible that adrenergic neurotransmitters are from circulation upon irradiation stress, rather than being produced locally by nerve fibers. This could explain why LepR-expressing cells did not decrease the expression of SCF, Ang2, and affect angiogenesis or hematopoiesis in innervation-defective LepR-Cre;NGF mice during homeostasis. Additionally, LepR+ cells exist in other tissues besides bone marrow, which might affect the adrenergic neurotransmitters in circulation upon damage. The authors should investigate whether LepR+ cells exist and express NGF in other tissues, such as quadriceps, kidney, adrenal medulla, and heart, to determine the tissue-specific role of LepR+ cells derived NGF in innervation".

(B) Address the questions raised in terms of the effects on both the hematopoietic and stromal compartment, as specifically requested by:

Referee #1:

"Figure 3a: The survival results in this panel indeed are consistent with a difference in the recovery of radioprotective cells which are responsible for survival in the first weeks after lethal dose TBI. Yang et al. (Blood 2005;105:2717) previously showed that the Lin-Sca1+ckit+CD34+Flt3- short term HSC population is the primary radioprotective cell population following myeloablation. It would be helpful for the authors to measure the percentages and numbers of HSCs (short term and long term) and HSPCs (lin-Sca1+ckit+) at day +7-10 in both groups prior to the natural deaths of the mice in the LepRCre+;Ngf fl/+ mice. At the same time, PB counts would also be helpful to the analysis as to the

underlying hematologic cause of early deaths in these mice".

"Figure 3b-d: The results at day +28 clearly indicate a hematopoietic deficit in the LepRCre;Ngf fl/+ mice compared to the controls. The differences may be understating the severity of the hematopoietic defect in the LepRCre;Ngf fl/+ mice since the mice that died earlier in the study are censored. Measurement of these parameters at day +7-10 would be helpful as mentioned above".

"Did the authors compare the hematopoietic response of LepRCre;Ngf fl/+ mice vs. control mice to sublethal irradiation or 5FU chemotherapy? Such comparisons would also be helpful toward complete understanding of how the loss of Ngf production by LepR+ stromal cells alters hematopoietic regeneration".

"Figure 3e-h: It is interesting that the defects in Evans Blue dye extravasation and EC and LepR+ stromal cell numbers are not different at day +14, proximate to the time period when the hematopoietic defect is severe enough to cause mortality in the LepRCre;Ngf fl/+ mice. As noted above, this could be due to censoring of mice which died before the day +14 time point. The authors should perform imaging of the BM vasculature and EC and LepR+ cell counts and Evans Blue dye analysis within the first 5-10 days post transplant to determine if an early defect in the niche populations can be detected".

"Figure 4n: The rescue effect of Salbutamol on the Evans Blue Dye extravasation is convincing. What is the effect of Salbutamol on the numbers of BM ECs, and LepR+ stromal cells at the same time frame?"

Referee #2:

"For the regeneration experiments in Figures 4 and 5. What are the numbers of LepR+ and endothelial cells? This information is missing and it is necessary to understand how the stroma regenerates".

(C) Provide additional data to further clarify the mechanistic aspects of regeneration, as indicated by:

Referee #1:

"The data in Figure 5l-n suggest that the lack of SCF, VEGF and/or Ang2 production by LepR+ cells are responsible for the hematopoietic recovery that should occur during the first 28 days after TBI/BMT. Are all 3 growth factors required to rescue the LepRCre;Adrb2;Adrb3 fl/fl mice after TBI/BMT? Would any or all of these growth factors rescue the LepRCre;Ngf fl/+ mice from the hematopoietic defect shown in Figure 3?"

Referee #3:

"To provide further insight into the role of LepR+ cells in hematopoietic and vascular regeneration, the authors should investigate or explain the mechanism by which adrenergic receptor signaling regulates regeneration factors in MSCs".

(D) All other referee concerns pertaining to strengthening existing data, methodological details, clarifications and textual changes, should also be addressed.

(E) Finally, please pay close attention to our guidelines on statistical and methodological reporting (listed below) as failure to do so may delay the reconsideration of the revised manuscript. In particular please provide:

We would be happy to consider a revised manuscript that would satisfactorily address these points, unless a similar paper is published elsewhere, or is accepted for publication in Nature Cell Biology in the meantime.

- ensure that it conforms to our format instructions and publication policies (see below and <https://www.nature.com/nature/for-authors>).

- provide a point-by-point rebuttal to the full referee reports verbatim, as provided at the end of this letter.

- provide the completed Reporting Summary (found here <https://www.nature.com/documents/nr-reporting-summary.pdf>). This is essential for reconsideration of the manuscript will be available to editors and referees in the event of peer review. For more information see <http://www.nature.com/authors/policies/availability.html> or contact me.

When submitting the revised version of your manuscript, please pay close attention to our [href="https://www.nature.com/nature-portfolio/editorial-policies/image-integrity">Digital Image Integrity Guidelines](https://www.nature.com/nature-portfolio/editorial-policies/image-integrity). and to the following points below:

Nature Cell Biology is committed to improving transparency in authorship. As part of our efforts in this direction, we are now requesting that all authors identified as 'corresponding author' on published papers create and link their Open Researcher and Contributor Identifier (ORCID) with their account on the Manuscript Tracking System (MTS), prior to acceptance. ORCID helps the scientific community achieve unambiguous attribution of all scholarly contributions. You can create and link your ORCID from the home page of the MTS by clicking on 'Modify my Springer Nature account'. For more information please visit www.springernature.com/orcid.

This journal strongly supports public availability of data. Please place the data used in your paper into a public data repository, or alternatively, present the data as Supplementary Information. If data can only be shared on request, please explain why in your Data Availability Statement, and also in the correspondence with your editor. Please note that for some data types, deposition in a public repository is mandatory - more information on our data deposition policies and available repositories appears below.

[redacted]

We would like to receive a revised submission within six months.

We hope that you will find our referees' comments, and editorial guidance helpful. Please do not hesitate to contact me if there is anything you would like to discuss.

Best wishes,

Stelios

Stylianos Lefkopoulos, PhD
He/him/his
Associate Editor
Nature Cell Biology
Springer Nature
Heidelberger Platz 3, 14197 Berlin, Germany

E-mail: stylianos.lefkopoulos@springernature.com
Twitter: @s_lefkopoulos

Reviewers' Comments:

Reviewer #1:

Remarks to the Author:

In the manuscript by Shen et al., the authors describe the expression pattern of Ngf in the bone marrow, the hematopoietic effects of Ngf deletion in LepR+ stromal cells on hematopoietic regeneration and nerve fiber density, and the authors demonstrate molecular linkage between NGF production by LepR+ stromal cells, nerve fiber restoration, beta adrenergic signaling, and production of hematopoietic growth factors by adrenergic receptor expressing LepR+ cells. There are several interesting findings in this paper. My major comments and concerns are summarized below:

Figure 1. The data convincingly show the expression of Ngf on lepR+ cells in the BM. In panel 1G, is the staining overlapping the arterioles a function of lepR+ cells surrounding these structures?

Figure 2:

2c: Please show additional lower power microscopic views of the BM so that the difference in total density of nerve fibers in the LepR-Cre+;Ngf fl/+ mice can be appreciated versus the control mice.

Figure 3. The results in this Figure are fascinating and raise several interesting questions.

Figure 3a: The survival results in this panel indeed are consistent with a difference in the recovery of radioprotective cells which are responsible for survival in the first weeks after lethal dose TBI. Yang et al. (Blood 2005;105:2717) previously showed that the Lin-Sca1+ckit+CD34+Flt3- short term HSC population is the primary radioprotective cell population following myeloablation. It would be helpful for the authors to measure the percentages and numbers of HSCs (short term and long term) and HSPCs (lin-Sca1+ckit+) at day +7-10 in both groups prior to the natural deaths of the mice in the LepRCre+;Ngf fl/+ mice. At the same time, PB counts would also be helpful to the analysis as to the underlying hematologic cause of early deaths in these mice.

Figure 3b-d: The results at day +28 clearly indicate a hematopoietic deficit in the LepRCre;Ngf fl/+ mice compared to the controls. The differences may be understating the severity of the hematopoietic defect in the LepRCre;Ngf fl/+ mice since the mice that died earlier in the study are censored. Measurement of these parameters at day +7-10 would be helpful as mentioned above.

Did the authors compare the hematopoietic response of LepRCre;Ngf fl/+ mice vs. control mice to sublethal irradiation or 5FU chemotherapy? Such comparisons would also be helpful toward complete understanding of how the loss of Ngf production by LepR+ stromal cells alters hematopoietic regeneration.

Figure 3e-h: It is interesting that the defects in Evans Blue dye extravasation and EC and LepR+ stromal cell numbers are not different at day +14, proximate to the time period when the hematopoietic defect is severe enough to cause mortality in the LepRCre;Ngf fl/+ mice. As noted above, this could be due to censoring of mice which died before the day +14 time point. The authors should perform imaging of the BM vasculature and EC and LepR+ cell counts and Evans Blue dye analysis within the first 5-10 days post transplant to determine if an early defect in the niche populations can be detected.

Please specify the TBI dose in Figure 3 BMT studies.

Please specify the TBI dose in the "irradiation" studies in Figure 4.

Figure 4c: Please show the isotype gates for this panel to confirm the gating for the lepR+ cells.

Figure 4n: The rescue effect of Salbutamol on the Evans Blue Dye extravasation is convincing. What is the effect of Salbutamol on the numbers of BM ECs, and LepR+ stromal cells at the same time frame?

Figure 5i and j: Do the LepRCre;Adrb2 fl/fl;Adrb3 fl/fl mice display a survival defect compared to the control mice after TBI and BMT as the LepRCre;Ngf fl/fl mice displayed in Figure 3a?

Figure 5l-n: Do adipocytes express the Adrb2 and Adrb3 receptors? If not, are adipocytes the source of SCF, VEGF and Ang2 that continues to be produced following TBI/BMT in the LepRCre;Adrb2;Adrb3 fl/fl mice, as shown in these panels?

The data in Figure 5l-n suggest that the lack of SCF, VEGF and/or Ang2 production by LepR+ cells are responsible for the hematopoietic recovery that should occur during the first 28 days after TBI/BMT. Are all 3 growth factors required to rescue the LepRCre;Adrb2;Adrb3 fl/fl mice after TBI/BMT? Would any or all of these growth factors rescue the LepRCre;Ngf fl/+ mice from the hematopoietic defect shown in Figure 3?

Reviewer #2:

Remarks to the Author:

In this manuscript Shen, Murphy et al., investigate the crosstalk between sympathetic nerves and bone marrow stromal cells during regeneration after transplantation. The authors first demonstrate that expression of the neurotrophic factor NGF is largely restricted to LepR+ stromal cells by generating Ngf-mScarlet reporter mice. Then they generate Ngfflox mice to conditionally delete Ngf in LepR cells. They show that this results in loss of NGF and progressive loss of bone marrow nerves by the age of six months but that this does not impair normal hematopoiesis. Using this model they confirm prior observations demonstrating that sympathetic nerves are required for bone marrow regeneration. Then they show that NGF levels are upregulated by myeloablation and that this correlates with increase innervation. By using 5-month old LepR-cre:Ngffl/Δ mice (when the BM still is innervated but NGF levels are reduced) they demonstrate that NGF produced from LepR+ cells and adipocytes is required for production of growth factors contributing to hematopoietic regeneration (e.g. VEGF and SCF). Finally they generate Adrb2 and Adrb3 floxed mice and use them to demonstrate that sympathetic nerves activate Lep-Cre+ cells through beta2 and beta 3 adrenergic receptors. This is an outstanding and impactful manuscript that elegantly uses novel mouse models to uncover bidirectional crosstalk between stromal cells and sympathetic nerves. The manuscript is a pleasure to read and it will be of interest to a broad readership. I just have some comments and suggestions on how to improve the manuscript.

-The manuscript relies on the use of LepR-cre:Ngffl/Δ mice at different ages since the effects of the deletion are progressive (first loss of NGF by age 5 months and denervation by age 6 months). This was a little bit confusing and it would be helpful if this progressive loss was explained more clearly

when the mouse model is first described (Fig. 2).

-The images showing NGF distribution, in the steady-state, in the Ngf-mScarlet reporter are spectacular and very convincing. However, the data on regeneration (Fig. 4b) is less clear. It appears that Ngf-expression is widespread and not restricted to Periplin+ adipocytes. The authors should show a counterstain for LepR+ cells to validate their claims.

-The observation that new arteries and nerves sprout after myeloablation is striking (Fig. 4f). Since the distribution of these fibers is not homogenous, do the authors observe increased LepR+ cell and hematopoietic recovery near nerve fibers?

-For the regeneration experiments in Figures 4 and 5. What are the numbers of LepR+ and endothelial cells? This information is missing and it is necessary to understand how the stroma regenerates.

- Figure 5. The fact that nerves mediate regeneration by targeting beta2 and beta 3 adrenergic receptors was already well established (Reference 11). This should be cited.

Minor comments

Fig. 1 d. It would be helpful to show in the main figure (not only in the Figure Legend) that the scale bar shows levels of Ngf.

What is the number of cells transplanted after myeloablation? The authors only mention that they use a radioprotective dose. This is important for interpretation and reproducibility as the number of donor cells transplanted also affects regeneration of the microenvironment.

Daniel Lucas
Associate Professor,
Cincinnati Children's Hospital Medical Center.

Reviewer #3:

Remarks to the Author:

The study by Shen et al. investigated the role of LepR+ cells derived nerve growth factor (NGF) in innervation and hematopoietic and vascular regeneration in the bone marrow. The authors used

genetic models and found that deleting NGF from Lepr⁺ cells impaired innervation and hematopoietic and vascular regeneration. Moreover, deleting adrenergic receptors $\beta 2$ and $\beta 3$ from Lepr⁺ cells impaired hematopoietic regeneration. The study is very well-organized and controlled. While the study provides potentially important information, the rationale and mechanism behind these observations are unclear, precluding a more favorable review.

Major comments:

1. The authors proposed that nerve fibers produce adrenergic neurotransmitters to activate Lepr⁺ cells, increasing the expression of regeneration factors, such as stem cell factor (SCF), Angiopoietin 2 (Ang2), and vascular endothelial growth factor (VEGF), for hematopoietic and vascular regeneration. However, it is unclear whether Lepr⁺ cells adjacent to nerve fibers produce more regeneration factors than other Lepr⁺ cells away from nerve fibers, as adrenergic neurotransmitters only target cells in close proximity. In fact, nerve fibers are rare and adjacent to arteries, but Lepr⁺ cells are more broadly distributed and adjacent to sinusoids in bone marrow. Therefore, the authors should provide more information on the spatial distribution of Ang2/VEGF-expressing cells in relation to nerve fibers in bone marrow to clarify how local nerve fibers contribute to hematopoietic and vascular regeneration through Lepr⁺ cells. Otherwise, they should consider the possibility that adrenergic neurotransmitters are circulating rather than being produced locally by nerve fibers.
2. It is possible that adrenergic neurotransmitters are from circulation upon irradiation stress, rather than being produced locally by nerve fibers. This could explain why Lepr-expressing cells did not decrease the expression of SCF, Ang2, and affect angiogenesis or hematopoiesis in innervation-defective Lepr-Cre;NGF mice during homeostasis. Additionally, Lepr⁺ cells exist in other tissues besides bone marrow, which might affect the adrenergic neurotransmitters in circulation upon damage. The authors should investigate whether Lepr⁺ cells exist and express NGF in other tissues, such as quadriceps, kidney, adrenal medulla, and heart, to determine the tissue-specific role of Lepr⁺ cells derived NGF in innervation.
3. The authors should investigate whether adipocytes or Lepr⁺ cells contribute to innervation and hematopoietic and vascular regeneration after irradiation. As adipocytes are abundant in bone marrow after irradiation, it is unclear whether Adipoq-Cre;NGF achieves a similar phenotype as Lepr-Cre;NGF mice. Additionally, it remains unclear whether the Lepr⁺ cells that remain as mesenchymal stem cells (MSCs) also contribute to the NGF production for regeneration.
4. To provide further insight into the role of Lepr⁺ cells in hematopoietic and vascular regeneration, the authors should investigate or explain the mechanism by which adrenergic receptor signaling regulates regeneration factors in MSCs.
5. The authors should consider the potential direct autocrine effect of NGF on MSCs, as the NGF receptor P75 is present in these cells for bone repair (James et al., Sci Adv 2022).

Minor comments:

1. The authors should consider the possibility that adrenergic neurotransmitters regulate hematopoietic and vascular regeneration directly, as adrenergic receptor $\beta 3$ is expressed and functions in HSCs (Lapidot et al., Nat Immunol 2007), and adrenergic receptor $\beta 2$ is expressed and functions in endothelial cells (Frenette et al., Science 2017).
2. NGF is also produced by macrophages (James et al., Sci Adv 2022). The authors need to compare the expression and function of NGF in macrophages and Lepr⁺ cells.
3. The authors should provide an explanation and discussion for the discrepancy between their findings and those of previous studies that show denervation of sympathetic nerves results in loss of HSCs (Hiromitsu Nakauch Cell 2011).

FINANCIAL AND NON-FINANCIAL COMPETING INTERESTS – the authors must include one of three declarations: (1) that they have no financial and non-financial competing interests; (2) that they have financial and non-financial competing interests; or (3) that they decline to respond, after the Author Contributions section. This statement will be published with the article, and in cases where financial and non-financial competing interests are declared, these will be itemized in a web supplement to the

article. For further details please see <https://www.nature.com/licenceforms/nrg/competing-interests.pdf>.

Methods should be written concisely, but should contain all elements necessary to allow interpretation and replication of the results. As a guideline, Methods sections typically do not exceed 3,000 words. The Methods should be divided into subsections listing reagents and techniques. When citing previous methods, accurate references should be provided and any alterations should be noted. Information must be provided about: antibody dilutions, company names, catalogue numbers and clone numbers for monoclonal antibodies; sequences of RNAi and cDNA probes/primers or company names and catalogue numbers if reagents are commercial; cell line names, sources and information on cell line identity and authentication. Animal studies and experiments involving human subjects must be reported in detail, identifying the committees approving the protocols. For studies involving human subjects/samples, a statement must be included confirming that informed consent was obtained. Statistical analyses and information on the reproducibility of experimental results should be provided in a section titled "Statistics and Reproducibility".

All Nature Cell Biology manuscripts submitted on or after March 21 2016 must include a Data availability statement as a separate section after Methods but before references, under the heading "Data Availability". For Springer Nature policies on data availability see <http://www.nature.com/authors/policies/availability.html>; for more information on this particular policy see <http://www.nature.com/authors/policies/data/data-availability-statements-data-citations.pdf>. The Data availability statement should include:

- Accession codes for primary datasets (generated during the study under consideration and designated as "primary accessions") and secondary datasets (published datasets reanalysed during the study under consideration, designated as "referenced accessions"). For primary accessions data should be made public to coincide with publication of the manuscript. A list of data types for which submission to community-endorsed public repositories is mandated (including sequence, structure, microarray, deep sequencing data) can be found here <http://www.nature.com/authors/policies/availability.html#data>.
- Unique identifiers (accession codes, DOIs or other unique persistent identifier) and hyperlinks for datasets deposited in an approved repository, but for which data deposition is not mandated (see here for details <http://www.nature.com/sdata/data-policies/repositories>).

- At a minimum, please include a statement confirming that all relevant data are available from the authors, and/or are included with the manuscript (e.g. as source data or supplementary information), listing which data are included (e.g. by figure panels and data types) and mentioning any restrictions on availability.
- If a dataset has a Digital Object Identifier (DOI) as its unique identifier, we strongly encourage including this in the Reference list and citing the dataset in the Methods.

We recommend that you upload the step-by-step protocols used in this manuscript to the Protocol Exchange. More details can found at www.nature.com/protocolexchange/about.

All imaging data should be accompanied by scale bars, which should be defined in the legend. Cropped images of gels/blots are acceptable, but need to be accompanied by size markers, and to retain visible background signal within the linear range (i.e. should not be saturated). The boundaries of panels with low background have to be demarked with black lines. Splicing of panels should only be considered if unavoidable, and must be clearly marked on the figure, and noted in the legend with a statement on whether the samples were obtained and processed simultaneously. Quantitative comparisons between samples on different gels/blots are discouraged; if this is unavoidable, it should only be performed for samples derived from the same experiment with gels/blots were processed in parallel, which needs to be stated in the legend.

The total number of Supplementary Figures (not including the “unprocessed scans” Supplementary Figure) should not exceed the number of main display items (figures and/or tables (see our Guide to Authors and March 2012 editorial <http://www.nature.com/ncb/authors/submit/index.html#supinfo>; <http://www.nature.com/ncb/journal/v14/n3/index.html#ed>). No restrictions apply to Supplementary Tables or Videos, but we advise authors to be selective in including supplemental data.

GUIDELINES FOR EXPERIMENTAL AND STATISTICAL REPORTING

REPORTING REQUIREMENTS – We are trying to improve the quality of methods and statistics reporting in our papers. To that end, we are now asking authors to complete a reporting summary that collects information on experimental design and reagents. The Reporting Summary can be found here <https://www.nature.com/documents/nr-reporting-summary.pdf>) If you would like to reference the guidance text as you complete the template, please access these flattened versions at <http://www.nature.com/authors/policies/availability.html>.

We strongly recommend the presentation of source data for graphical and statistical analyses as a

separate Supplementary Table, and request that source data for all independent repeats are provided when representative experiments of multiple independent repeats, or averages of two independent experiments are presented. This supplementary table should be in Excel format, with data for different figures provided as different sheets within a single Excel file. It should be labelled and numbered as one of the supplementary tables, titled "Statistics Source Data", and mentioned in all relevant figure legends.

Author Rebuttal to Initial comments

Referee #1:

*In the manuscript by Shen et al., the authors describe the expression pattern of *Ngf* in the bone marrow, the hematopoietic effects of *Ngf* deletion in *LepR*⁺ stromal cells on hematopoietic regeneration and nerve fiber density, and the authors demonstrate molecular linkage between NGF production by *LepR*⁺ stromal cells, nerve fiber restoration, beta adrenergic signaling, and production of hematopoietic growth factors by adrenergic receptor expressing *LepR*⁺ cells. There are several interesting findings in this paper. My major comments and concerns are summarized below:*

*Figure 1. The data convincingly show the expression of *Ngf* on *lepR*⁺ cells in the BM. In panel 1G, is the staining overlapping the arterioles a function of *lepR*⁺ cells surrounding these structures?*

RESPONSE: *Ngf* is expressed by *LepR*⁺ cells around both arterioles and sinusoids. The staining surrounding arterioles in Fig. 1g looks denser than the staining around sinusoids because the *Ngf*-expressing periaarteriolar cells include both *LepR*⁺*Oln*⁺ stromal cells and *SMA*⁺*NG2*⁺ smooth muscle cells (Fig. 1e and ED Fig. 1f). Nonetheless, the *Ngf*-expressing smooth muscle cells are small in number when compared to the *Ngf*-expressing perisinusoidal *LepR*⁺ cells throughout the bone marrow. Consequently, 90% of all *Ngf*-expressing cells in the bone marrow are *LepR*⁺ cells (Fig. 1h) and deletion of *Ngf* from *LepR*⁺ cells eliminates more than 90% of NGF protein from the bone marrow (Fig. 2a) as well as all nerve fibers (Fig. 2b). Conversely, deletion of *Ngf* with *NG2*-creER did not significantly affect bone marrow NGF levels (Fig. 2a) or nerve fibers in the bone marrow (Fig. 2b).

*Figure 2: 2c: Please show additional lower power microscopic views of the BM so that the difference in total density of nerve fibers in the *LepR*-Cre⁺; *Ngf* fl/+ mice can be appreciated versus the control mice.*

RESPONSE: We have replaced the images in the original Fig. 2c with new lower magnification images of bone marrow from *LepR*^{Cre/+}; *Ngf*^{fl/Δ} and littermate controls. These images show nerve

fibers (green) in control bone marrow but not in the bone marrow of 6 month old *Lep^{Cre/+}; Ngf^{fl/Δ}* mice. Images of nerve fibers at various magnifications are also present in several other figures (e.g. ED Fig. 3d-g; Figure 4a and 4b; Figure 5f).

Figure 3. The results in this Figure are fascinating and raise several interesting questions.

*Figure 3a: The survival results in this panel indeed are consistent with a difference in the recovery of radioprotective cells which are responsible for survival in the first weeks after lethal dose TBI. Yang et al. (Blood 2005;105:2717) previously showed that the Lin-Sca1+ckit+CD34+Flt3- short term HSC population is the primary radioprotective cell population following myeloablation. It would be helpful for the authors to measure the percentages and numbers of HSCs (short term and long term) and HSPCs (lin-Sca1+ckit+) at day +7-10 in both groups prior to the natural deaths of the mice in the *Lep^{Cre/+}; Ngf^{fl/+}* mice. At the same time, PB counts would also be helpful to the analysis as to the underlying hematologic cause of early deaths in these mice.*

RESPONSE: We have added considerable new data to address this but were not able to assess the frequencies and numbers of HSCs within the first 7-10 days after myeloablation as the surface markers used to identify short-term and long-term HSCs change after myeloablation in a manner that makes them impossible to accurately quantify until approximately 14 days after myeloablation (e.g. Blood 89:3596). We have added new data showing peripheral blood counts (Fig. 3b-d), bone marrow cellularity (Fig. 3e), and numbers of LSK cells in bone marrow (Fig. 3f) before irradiation and at 7, 14, and 28 days after irradiation. White blood cells, red blood cells, platelets, and LSK cells were all significantly depleted at 14 and 28 days after irradiation in *Lep^{Cre/+}; Ngf^{fl/Δ}* as compared to littermate control mice. We also added new data showing significant reductions in bone marrow cellularity (Fig. 3h), numbers of LSK cells (Fig. 3i), LepR⁺ cells (Fig. 3j), and endothelial cells (Fig. 3k) in *Lep^{Cre/+}; Ngf^{fl/Δ}* mice as compared to littermate controls at 10 days after irradiation. These new data are consistent with the data in the original manuscript, which showed that bone marrow cellularity (Fig. 3e), HSC numbers (Fig. 3g), and LSK cell numbers (Fig. 3f) were all depleted at 28 days after irradiation in *Lep^{Cre/+}; Ngf^{fl/Δ}* as compared to littermate control mice. *Lep^{Cre/+}; Ngf^{fl/Δ}* mice were also less likely to survive a sublethal dose of irradiation as compared to littermate control mice (new ED Fig. 5a). *Lep^{Cre/+}; Ngf^{fl/Δ}* mice were thus less able to regenerate HSCs, LSK cells, hematopoiesis, and bone marrow stromal cells from 10 to 28 days after irradiation.

*Figure 3b-d: The results at day +28 clearly indicate a hematopoietic deficit in the *Lep^{Cre/+}; Ngf^{fl/+}* mice compared to the controls. The differences may be understating the severity of the hematopoietic defect in the *Lep^{Cre/+}; Ngf^{fl/+}* mice since the mice that died earlier in the study are censored. Measurement of these parameters at day +7-10 would be helpful as mentioned above.*

RESPONSE: As noted above, we added new data at 10 days after irradiation showing significant reductions in bone marrow cellularity (Fig. 3h), LSK cells (Fig. 3i), LepR⁺ cells (Fig. 3j), and endothelial cells (Fig. 3k) in *Lep^{Cre/+}; Ngf^{fl/Δ}* mice as compared to controls.

Did the authors compare the hematopoietic response of LepRCre;Ngf fl/+ mice vs. control mice to sublethal irradiation or 5FU chemotherapy? Such comparisons would also be helpful toward complete understanding of how the loss of Ngf production by LepR+ stromal cells alters hematopoietic regeneration.

RESPONSE: We have added new data showing that *Lep^{Cre/+}; Ngf^{fl/Δ}* mice exhibit reduced survival and regeneration of bone marrow cellularity, HSC numbers, and LSK numbers after sublethal irradiation (ED Fig. 5a-d) and after 5-fluorouracil treatment (Fig. 3s-v). Thus, NGF production by LepR⁺ cells is important to promote hematopoietic regeneration irrespective of whether myeloablation is by irradiation or chemotherapy.

Figure 3e-h: It is interesting that the defects in Evans Blue dye extravasation and EC and LepR+ stromal cell numbers are not different at day +14, proximate to the time period when the hematopoietic defect is severe enough to cause mortality in the LepRCre;Ngf fl/+mice. As noted above, this could be due to censoring of mice which died before the day +14 time point. The authors should perform imaging of the BM vasculature and EC and LepR+ cell counts and Evans Blue dye analysis within the first 5-10 days post transplant to determine if an early defect in the niche populations can be detected.

RESPONSE: We added new data to Fig. 3j and 3k showing that LepR⁺ cells and endothelial cells were both depleted in *Lep^{Cre/+}; Ngf^{fl/Δ}* mice as compared to littermate controls at 10 days after sublethal irradiation. Bone marrow vasculature was also leakier in *Lep^{Cre/+}; Ngf^{fl/Δ}* as compared to control mice at this time point (new Fig. 3l). Figure 3o-r shows a defect in the proliferation of LepR⁺ cells and endothelial cells in *Lep^{Cre/+}; Ngf^{fl/Δ}* as compared to littermate control mice at 14 days after lethal irradiation. In this experiment there was a trend toward reduced numbers of LepR⁺ cells and endothelial cells in the *Lep^{Cre/+}; Ngf^{fl/Δ}* mice at 14 days after irradiation but the differences were not statistically significant until 28 days after irradiation (the next time point examined). All of the data thus indicate early defects in the regeneration of LepR⁺ cells and endothelial cells in *Lep^{Cre/+}; Ngf^{fl/Δ}* mice that were evident 10 to 14 days after irradiation.

Please specify the TBI dose in Figure 3 BMT studies.

Please specify the TBI dose in the “irradiation” studies in Figure 4.

RESPONSE: All experiments that involved bone marrow transplantation, including those in Figures 3 and 5 (previous Figures 3 and 4), were performed with lethal irradiation (a total of 1080 rads given over 4 hours). Experiments that involved sublethal irradiation without bone marrow transplantation were performed with one dose of 650 rads (ED Fig. 5a-d). We have added this information to the text and the methods.

Figure 4c: Please show the isotype gates for this panel to confirm the gating for the lepR+ cells.

RESPONSE: Figure 4c in the original manuscript is now Figure 5c in the revised version. We have added isotype control staining to this figure to confirm the LepR⁺ cell gating.

Figure 4n: The rescue effect of Salbutamol on the Evans Blue Dye extravasation is convincing. What is the effect of Salbutamol on the numbers of BM ECs, and LepR+ stromal cells at the same time frame?

RESPONSE: We have added new data showing that salbutamol treatment also rescued the regeneration of LepR⁺ cells and endothelial cells in *LepR^{Cre/+}; Ngf^{fl/Δ}* mice (ED Fig. 7k and 7l) at the same time point as we had shown the complete rescue of bone marrow cellularity, HSC numbers, LSK numbers, and blood vessel leakiness by salbutamol (Fig. 5k-n; all at 28 days after irradiation).

Figure 5i and j: Do the LepRCre;Adrb2 fl/fl;Adrb3 fl/fl mice display a survival defect compared to the control mice after TBI and BMT as the LepRCre;Ngf fl/fl mice displayed in Figure 3a?

RESPONSE: Yes. We have added a new Fig. 6h showing that *LepR^{Cre/+}; Adrb2^{fl/fl}; Adrb3^{fl/fl}* mice exhibited reduced survival after irradiation as compared to littermate controls, similar to the reduced survival observed in *LepR^{Cre/+}; Ngf^{fl/Δ}* mice (Fig. 3a).

Figure 5l-n: Do adipocytes express the Adrb2 and Adrb3 receptors? If not, are adipocytes the source of SCF, VEGF and Ang2 that continues to be produced following TBI/BMT in the LepRCre;Adrb2;Adrb3 fl/fl mice, as shown in these panels?

RESPONSE: Yes, we showed in ED Fig. 8a-e that both LepR⁺ cells and adipocytes express *adrb2* and *adrb3*. We also showed that both LepR⁺ cells and adipocytes express *Scf*, *Vegf* and *Ang2* (new ED Fig. 8f-h).

The data in Figure 5l-n suggest that the lack of SCF, VEGF and/or Ang2 production by LepR+ cells are responsible for the hematopoietic recovery that should occur during the first 28 days after TBI/BMT. Are all 3 growth factors required to rescue the LepRCre;Adrb2;Adrb3 fl/fl mice after TBI/BMT? Would any or all of these growth factors rescue the LepRCre;Ngf fl/+ mice from the hematopoietic defect shown in Figure 3?

RESPONSE: Published data indicate that SCF, VEGF and Ang2 all contribute individually to the regeneration of hematopoiesis and/or vasculature after irradiation. We showed that SCF production by LepR⁺ cells and adipocytes promotes the regeneration of HSCs and hematopoiesis in the bone marrow after irradiation (Nature Cell Biology 19:891). One of our collaborators, Shentong Fang, showed that VEGF produced by LepR⁺ cells promotes the regeneration of bone marrow sinusoids after irradiation (Blood 136:1871). Others have shown that Ang2 also promotes sinusoid regeneration (Science 343:416). The Ang2 work was performed in the liver but *Ang2* is also expressed by LepR⁺ cells in the bone marrow (ED Fig. 8h), suggesting it is likely to participate in sinusoid regeneration in the bone marrow as well. Multiple factors that are known to promote hematopoietic or vascular regeneration are thus upregulated after myeloablation in LepR⁺ cells and adipocytes as a consequence of β adrenergic receptor signaling. Since each of these factors is necessary for hematopoietic and/or vascular regeneration, it would be necessary to provide all three factors to rescue regeneration in *LepR^{Cre/+}; Adrb2^{fl/fl}; Adrb3^{fl/fl}* or *LepR^{Cre/+}; Ngf^{fl/ Δ}* mice. Moreover, it is likely that these are not the only regeneration factors that are produced by LepR⁺ cells and their progeny; therefore, even over-expression of all three factors in *LepR^{Cre/+}; Ngf^{fl/ Δ}* mice would likely not be sufficient to fully rescue bone marrow regeneration. However, testing this would require the generation of new mice that over-express multiple factors and since all three factors have already been shown to be necessary for hematopoietic/vascular regeneration, such experiments would add little beyond what has already been published.

Reviewer #2:

In this manuscript Shen, Murphy et al., investigate the crosstalk between sympathetic nerves and bone marrow stromal cells during regeneration after transplantation. The authors first demonstrate that expression of the neurotrophic factor NGF is largely restricted to LepR+ stromal cells by generating Ngf-mScarlet reporter mice. Then they generate Ngfflox mice to conditionally delete Ngf in LepR cells. They show that this results in loss of NGF and progressive loss of bone marrow nerves by the age of six months but that this does not impair normal hematopoiesis. Using this model they confirm prior observations demonstrating that sympathetic

nerves are required for bone marrow regeneration. Then they show that NGF levels are upregulated by myeloablation and that this correlates with increase innervation. By using 5-month old LepR-cre:Ngffl/Δ mice (when the BM still is innervated but NGF levels are reduced) they demonstrate that NGF produced from LepR+ cells and adipocytes is required for production of growth factors contributing to hematopoietic regeneration (e.g. VEGF and SCF). Finally they generate Adrb2 and Adrb3 floxed mice and use them to demonstrate that sympathetic nerves activate Lep-Cre+ cells through beta2 and beta 3 adrenergic receptors. This is an outstanding and impactful manuscript that elegantly uses novel mouse models to uncover bidirectional crosstalk between stromal cells and sympathetic nerves. The manuscript is a pleasure to read and it will be of interest to a broad readership. I just have some comments and suggestions on how to improve the manuscript.

The manuscript relies on the use of LepR-cre:Ngffl/Δ mice at different ages since the effects of the deletion are progressive (first loss of NGF by age 5 months and denervation by age 6 months). This was a little bit confusing and it would be helpful if this progressive loss was explained more clearly when the mouse model is first described (Fig. 2).

RESPONSE: Thanks for this suggestion. We revised the text and added new data to clarify this point. The new data show that *Ngf* transcript levels in LepR⁺ cells decline between 2 and 6 months of age in *LepR^{Cre/+}; Ngf^{fl/Δ}* mice: *Ngf* transcript levels were reduced by 65% in LepR⁺ cells from *LepR^{Cre/+}; Ngf^{fl/Δ}* as compared to control mice at 2 months of age and by more than 90% at 6 months of age (new Fig. 2e). This is consistent with data showing that *LepR^{Cre}* recombines inefficiently before 2 months of age (Dev Cell 58:348). Nearly complete deletion of *Ngf* from LepR⁺ cells is necessary to eliminate nerve fibers from the bone marrow.

The images showing NGF distribution, in the steady-state, in the Ngf-mScarlet reporter are spectacular and very convincing. However, the data on regeneration (Fig. 4b) is less clear. It appears that Ngf-expression is widespread and not restricted to Perilipin+ adipocytes. The authors should show a counterstain for LepR+ cells to validate their claims.

RESPONSE: We added new images of *Ngf*-Scarlet and anti-LepR antibody staining in bone marrow sections from *Ngf^{mScarlet/+}* mice at 14 days after irradiation (ED Fig. 6a). Consistent with the flow cytometry data that demonstrated that most LepR⁺ cells express *Ngf*-mScarlet after irradiation (Fig. 5c), the new images in ED Fig. 6a show widespread co-staining of LepR and *Ngf*-mScarlet in perisinusoidal cells throughout the bone marrow.

The original Figure 4b showing NGF expression by perilipin⁺ adipocytes is now Figure 5b in the revised manuscript. We showed by single cell RNA sequencing, flow cytometric analysis of *Ngf*-Scarlet mice, and quantitative RT-PCR that NGF is expressed by LepR⁺ cells in both normal

(e.g. Figure 1) and regenerating (e.g. Figure 5c and 5d) bone marrow. Deletion of *Ngf* from LepR⁺ cells and their progeny eliminates more than 90% of the NGF protein from the bone marrow under steady-state conditions (Fig. 2a) and impairs the regeneration of hematopoiesis and vasculature after myeloablation (Fig. 3). Thus, the reason why NGF expression appears widespread in Figure 5b is that in addition to being expressed by adipocytes, NGF is also expressed by LepR⁺ cells. LepR⁺ cells represent fewer than 1% of cells in the bone marrow, but have long processes that make these cells appear abundant in bone marrow images. Moreover, the image in Figure 5b was from a relatively thick section (30 μm) to facilitate the imaging of large adipocytes. Consequently, the *Ngf*-Scarlet in this image reflects the aggregate expression of *Ngf* by many LepR⁺ cells and adipocytes throughout the section. We have revised the figure legend to clarify this.

The observation that new arteries and nerves sprout after myeloablation is striking (Fig. 4f). Since the distribution of these fibers is not homogenous, do the authors observe increased LepR+ cell and hematopoietic recovery near nerve fibers?

RESPONSE: Figure 4f in the original manuscript is now Figure 5f in the revised manuscript. We have added new data quantifying the distances from LepR⁺ cells, *Scf*-GFP⁺ stromal cells, and *Scf*-GFP⁺ adipocytes to nerve fibers in the bone marrow of non-irradiated and irradiated mice (ED Fig. 6l-n). LepR⁺ cells (ED Fig. 6l) and *Scf*-GFP⁺ stromal cells (ED Fig. 6m) were both significantly more likely to localize within 10 μm of nerve fibers at 14 days after irradiation as compared to in non-irradiated mice. This is consistent with the idea that regeneration is enhanced immediately adjacent to nerve fibers. On the other hand, the vast majority of LepR⁺ cells, *Scf*-GFP⁺ stromal cells, and *Scf*-GFP⁺ adipocytes were at least 20 μm from nerve fibers and there was no significant difference in the percentages of these cells that were 20 to 120 μm from nerve fibers before and after irradiation. This suggests that nerve fibers promote the regeneration of LepR⁺ cells throughout the bone marrow, but that regeneration may be enhanced immediately adjacent to nerve fibers.

How could the nerve fibers promote the regeneration of niche cells throughout the bone marrow? Peripheral nerves release adrenergic neurotransmitters through volume transmission, not through spatially restricted synaptic transmission (e.g. *Neuroscience* 155:997; *Pharmacological Reviews* 52:595). Non-synaptic volume transmission involves the broad release of neurotransmitters that can diffuse considerable distances away from nerve fibers. Volume transmission from some types of neurons allows the neurotransmitters to act on receiving cells that are hundreds of microns away (*Progress in Neurobiology* 90:82). LepR⁺ cells also have long processes that allow them to interact with cells that are not adjacent to the LepR⁺ cell body. Volume transmission, long LepR⁺ cell processes, and perhaps other mechanisms that

propagate signals among LepR⁺ cells may enable nerve fibers around arterioles to promote regeneration throughout the bone marrow.

For the regeneration experiments in Figures 4 and 5. What are the numbers of LepR⁺ and endothelial cells? This information is missing and it is necessary to understand how the stroma regenerates.

RESPONSE: We added new data showing the numbers of LepR⁺ cells and endothelial cells in the regeneration experiments in Figures 5 and 6 (these are the new versions of Figures 4 and 5 in the original manuscript). Salbutamol treatment rescued not only the regeneration of bone marrow cellularity, HSC numbers, LSK cell numbers, and vascular patency in the bone marrow (Fig. 5k-n) but also the regeneration of LepR⁺ cells and endothelial cells (new ED Fig. 7k and 7l) in *Lep^{Cre/+}; Ngf^{fl/Δ}* mice. Consistent with these data, *Lep^{Cre/+}; Adrb2^{fl/fl}*; *Adrb3^{fl/fl}* mice exhibited impaired regeneration of LepR⁺ cells and endothelial cells (new Fig. 6m and 6n) in addition to hematopoietic cells (Fig. 6i-k). Thus, the activation of β adrenergic receptors in LepR⁺ cells and their progeny by nerve fibers promotes the regeneration of LepR⁺ stromal cells, endothelial cells, and hematopoietic cells.

Figure 5. The fact that nerves mediate regeneration by targeting beta2 and beta 3 adrenergic receptors was already well established (Reference 11). This should be cited.

RESPONSE: In the original manuscript, we described this study as having shown that “Sympathetic nerve fibers in the bone marrow release adrenergic neurotransmitters that promote hematopoietic regeneration by activating β adrenergic receptors”. We have edited this sentence in the revised manuscript to read “Sympathetic nerve fibers in the bone marrow release adrenergic neurotransmitters that promote hematopoietic regeneration by activating β2 and β3 adrenergic receptors”. The main advance in this part of our manuscript is to show that β adrenergic receptors act within LepR⁺ cells and their progeny to promote hematopoietic and vascular regeneration. It was not clear from prior studies what cells the β adrenergic receptors act in to promote hematopoietic regeneration.

Minor comments

Fig. 1 d. It would be helpful to show in the main figure (not only in the Figure Legend) that the scale bar shows levels of Ngf.

RESPONSE: We have revised Fig. 1d to indicate that the scale bar shows *Ngf* levels.

What is the number of cells transplanted after myeloablation? The authors only mention that they use a radioprotective dose. This is important for interpretation and reproducibility as the number of donor cells transplanted also affects regeneration of the microenvironment.

RESPONSE: We transplanted 1,000,000 whole bone marrow cells to radioprotect mice after lethal irradiation. We have added this information to the methods section.

Reviewer #3:

Remarks to the Author:

*The study by Shen et al. investigated the role of *Lepr*⁺ cells derived nerve growth factor (NGF) in innervation and hematopoietic and vascular regeneration in the bone marrow. The authors used genetic models and found that deleting NGF from *Lepr*⁺ cells impaired innervation and hematopoietic and vascular regeneration. Moreover, deleting adrenergic receptors $\beta 2$ and $\beta 3$ from *Lepr*⁺ cells impaired hematopoietic regeneration. The study is very well-organized and controlled. While the study provides potentially important information, the rationale and mechanism behind these observations are unclear, precluding a more favorable review.*

Major comments:

*The authors proposed that nerve fibers produce adrenergic neurotransmitters to activate *Lepr*⁺ cells, increasing the expression of regeneration factors, such as stem cell factor (SCF), Angiopoietin 2 (Ang2), and vascular endothelial growth factor (VEGF), for hematopoietic and vascular regeneration. However, it is unclear whether *Lepr*⁺ cells adjacent to nerve fibers produce more regeneration factors than other *Lepr*⁺ cells away from nerve fibers, as adrenergic neurotransmitters only target cells in close proximity. In fact, nerve fibers are rare and adjacent to arteries, but *Lepr*⁺ cells are more broadly distributed and adjacent to sinusoids in bone marrow. Therefore, the authors should provide more information on the spatial distribution of Ang2/VEGF-expressing cells in relation to nerve fibers in bone marrow to clarify how local nerve fibers contribute to hematopoietic and vascular regeneration through *Lepr*⁺ cells. Otherwise, they should consider the possibility that adrenergic neurotransmitters are circulating rather than being produced locally by nerve fibers.*

RESPONSE: We have added new data quantifying the distances from *LepR*⁺ cells, *Scf*-GFP⁺ stromal cells, and *Scf*-GFP⁺ adipocytes to nerve fibers in the bone marrow of non-irradiated and irradiated mice (ED Fig. 6l-n). *LepR*⁺ cells (ED Fig. 6l) and *Scf*-GFP⁺ stromal cells (ED Fig. 6m) were both significantly more likely to localize within 10 μ m of nerve fibers at 14 days after irradiation as compared to in non-irradiated mice. This is consistent with the idea that regeneration is enhanced immediately adjacent to nerve fibers. On the other hand, the vast

majority of LepR⁺ cells, Scf-GFP⁺ stromal cells, and Scf-GFP⁺ adipocytes were at least 20 μm from nerve fibers and there were no significant differences in the percentages of these cells that were 20 to 120 μm from nerve fibers before and after irradiation. This suggests that nerve fibers promote the regeneration of LepR⁺ cells throughout the bone marrow, but regeneration may be enhanced immediately adjacent to nerve fibers.

How could the nerve fibers promote the regeneration of niche cells throughout the bone marrow? Peripheral nerves release adrenergic neurotransmitters through volume transmission, not through spatially restricted synaptic transmission (e.g. Neuroscience 155:997; Pharmacological Reviews 52:595). While synaptic transmission can only occur immediately adjacent to nerve fibers, non-synaptic volume transmission involves the broad release of adrenergic neurotransmitters that can diffuse considerable distances away from nerve fibers. Volume transmission sometimes allows neurotransmitters to act on receiving cells that are hundreds of microns away (Progress in Neurobiology 90:82). This would enable nerve fibers associated with arterioles to promote regeneration throughout the bone marrow.

It is possible that adrenergic neurotransmitters are from circulation upon irradiation stress, rather than being produced locally by nerve fibers. This could explain why Lepr-expressing cells did not decrease the expression of SCF, Ang2, and affect angiogenesis or hematopoiesis in innervation-defective Lepr-Cre;NGF mice during homeostasis. Additionally, Lepr+ cells exist in other tissues besides bone marrow, which might affect the adrenergic neurotransmitters in circulation upon damage. The authors should investigate whether Lepr+ cells exist and express NGF in other tissues, such as quadriceps, kidney, adrenal medulla, and heart, to determine the tissue-specific role of Lepr+ cells derived NGF in innervation.

RESPONSE: We have added new data in which we used *Prx1-cre* to conditionally delete *Ngf* from mesenchymal stromal cells in limbs. This leads to the loss of NGF from LepR⁺ cells within the bone marrow of limb bones, but there is no recombination in the axial skeleton (new Fig. 4). Consistent with this, *Prx1-cre; Ngf^{fl/fl}* mice exhibited a loss of nerve fibers from the bone marrow in femurs but not in vertebrae (new Fig. 4a-c). This shows that NGF from LepR⁺ cells regulates bone marrow nerve fibers by acting locally within the bone marrow compartment in which it is produced, not as a result of systemic effects. After irradiation, hematopoietic and vascular regeneration were impaired in femur, but not vertebral, bone marrow from *Prx1-cre; Ngf^{fl/fl}* mice as compared to littermate controls (new Fig. 4g-l). Thus, LepR⁺ cells act locally within the bone marrow to promote innervation and the nerve fibers act locally, not systemically, to promote hematopoietic and vascular regeneration.

Consistent with this, we were unable to detect changes in the levels of circulating growth factors in the blood after irradiation. As compared to non-irradiated mice, bone marrow NGF, SCF, VEGF, and Ang2 levels increased and nerve fibers sprouted in the bone marrow 14 days after irradiation, and these effects were not observed in *LepR^{Cre/+}; Ngf^{fl/Δ}* mice (Fig. 5a, and Fig. 5f-j). However, none of these growth factors significantly changed in the blood before versus after myeloablation, or in the blood of *LepR^{Cre/+}; Ngf^{fl/Δ}* as compared to littermate control mice after irradiation (new ED Fig. 6h-k). Therefore, we were unable to detect any evidence of systemic changes in the levels of these growth factors

It's also important to note that there are large bodies of literature that demonstrate NGF acts locally to promote the maintenance of nerve fibers within target tissues (e.g. PNAS 74:4516; Annual Review of Neuroscience 24:1217). There is no evidence that we are aware of for the existence of LepR⁺ cells outside of the bone marrow that express NGF or HSC niche factors. Given the genetic evidence described above, even if such cells are identified someday, they could not explain the local effects observed in our experiments. Moreover, the existence of LepR⁺ cells in the bone marrow that express high levels of NGF and HSC niche factors offers a simple explanation for all of the effects we observed on bone marrow regeneration.

The authors should investigate whether adipocytes or LepR+ cells contribute to innervation and hematopoietic and vascular regeneration after irradiation. As adipocytes are abundant in bone marrow after irradiation, it is unclear whether Adipoq-Cre;NGF achieves a similar phenotype as LepR-Cre;NGF mice. Additionally, it remains unclear whether the LepR+ cells that remain as mesenchymal stem cells (MSCs) also contribute to the NGF production for regeneration.

RESPONSE: Both LepR⁺ cells and adipocytes express NGF (Fig. 5b and 5c, new ED Fig. 6a) as well as SCF, VEGF, and Ang2 (ED Fig. 8f-h) such that both cell populations likely contribute to nerve sprouting, vascular, and hematopoietic regeneration during bone marrow regeneration. We have added new data in which we deleted *Ngf* using *Adiponectin-CreER*. We found that 2-3 month old *Adiponectin-CreER; Ngf^{fl/Δ}* mice (2 weeks after tamoxifen treatment) phenocopied 4-5 month old *LepR^{Cre/+}; Ngf^{fl/Δ}* mice (new ED Fig. 7a-j), with a loss of nerve fiber sprouting in the bone marrow after irradiation (new ED Fig. 7a), impaired hematopoietic and vascular regeneration (new ED Fig. 7b-g), and reduced growth factor levels (new ED Fig. 7h-j) after irradiation. However, this experiment does not distinguish between effects of LepR⁺ cells versus adipocytes on nerve maintenance because there is a nearly complete overlap between *LepR^{Cre}* and *Adiponectin-CreER* expression in the bone marrow: virtually all LepR⁺ cells express adiponectin and vice versa, including all mesenchymal stem cells in the bone marrow (Cell Stem Cell 29:1547). There is no known Cre allele that recombines in bone marrow adipocytes

but not in LepR⁺ cells. Thus, it is currently not technically possible to functionally distinguish the contributions of LepR⁺ cells from the adipocytes they give rise to after irradiation.

To provide further insight into the role of LepR⁺ cells in hematopoietic and vascular regeneration, the authors should investigate or explain the mechanism by which adrenergic receptor signaling regulates regeneration factors in MSCs.

RESPONSE: We have added new data suggesting that β -adrenergic receptors signal through protein kinase A (PKA) in LepR⁺ cells. In other cell types, β -adrenergic receptors are known to signal through PKA to increase the expression of factors like VEGF (e.g. Nature Medicine 12:939; Breast Cancer Res Treat 130:747). Consistent with this, we found that LepR⁺ cells from *LepR^{Cre/+}; Ngf^{fl/Δ}* mice had lower levels of phosphorylated PKA as compared to LepR⁺ cells from control mice at 14 days after irradiation and that treatment with salbutamol, which rescued hematopoietic and vascular regeneration in these mice, also rescued PKA phosphorylation in the LepR⁺ cells (new Fig. 5o) as well as levels of SCF, VEGF and Ang2 in bone marrow (new ED Fig. 7m-o). LepR⁺ cells from *LepR^{Cre/+}; Adrb2^{fl/fl}; Adrb3^{fl/fl}* mice also had lower levels of phosphorylated PKA as compared to LepR⁺ cells from control mice at 14 days after irradiation (new Fig. 6r). Thus, β -adrenergic receptors appear to signal through PKA to promote growth factor expression in LepR⁺ cells as they have already been shown to do in other cell types.

The authors should consider the potential direct autocrine effect of NGF on MSCs, as the NGF receptor P75 is present in these cells for bone repair (James et al., Sci Adv 2022).

RESPONSE: We have not made any claims related to osteogenesis or bone repair in our manuscript so the question of whether or not NGF influences bone repair by acting on MSCs would not affect any of our conclusions. Nonetheless, we looked into this possibility. As shown below, our single-cell RNA-seq analysis of bone marrow stromal cells indicates that *p75* is expressed by Schwann cells but there is little or no expression by LepR⁺ cells (all MSCs in adult bone marrow are LepR⁺; Cell Stem Cell 29:1547) or other stromal cells within the bone marrow. These data are consistent with the James et al. study which reported *p75* expression in MSCs from the periosteum but not the bone marrow (Sci Adv 8:eabl5716). All of our conclusions relate to effects of NGF on nerve fibers and hematopoietic/vascular regeneration within the bone marrow, not effects on bone regeneration in the periosteum on the outer surface of bones.

Consistent with our conclusions, peripheral neurons express NGF receptor (J Neurosci 8:3481, PNAS 84:3060) and NGF is known to act directly on these cells to promote neurite maintenance (Annu Rev Neurosci 24: 677). Therefore, there is no reason to posit indirect mechanisms involving other cell types.

Minor comments:

1. The authors should consider the possibility that adrenergic neurotransmitters regulate hematopoietic and vascular regeneration directly, as adrenergic receptor $\beta 3$ is expressed and functions in HSCs (Lapidot et al., Nat Immunol 2007), and adrenergic receptor $\beta 2$ is expressed and functions in endothelial cells (Frenette et al., Science 2017).

RESPONSE: The data are not consistent with this possibility. We have shown that beta adrenergic receptors act within LepR⁺ cells to promote hematopoietic and vascular regeneration because *LepR^{Cre/+}; Adrb2^{fl/fl}; Adrb3^{fl/fl}* mice phenocopy all aspects of the hematopoietic and vascular regeneration defects observed in *LepR^{Cre/+}; Ngf^{fl/ Δ}* mice. This would not happen if hematopoietic and vascular regeneration were regulated by β adrenergic receptors expressed by HSCs or endothelial cells as these cells do not recombine with *Lepr-cre* (e.g. Nature 481:457).

2. NGF is also produced by macrophages (James et al., Sci Adv 2022). The authors need to compare the expression and function of NGF in macrophages and LepR⁺ cells.

RESPONSE: We added new data in which we performed flow cytometric analysis of *Ngf*-mScarlet expression by CD11b expressing myeloid cells, which include macrophages, in the

blood and bone marrow. We found virtually no NGF expression by these cells - less than 0.01% of these cells expressed *Ngf*-mScarlet (new ED Fig. 1i and 1j). This is consistent with data in the original manuscript that showed by single cell RNA sequencing (Fig. 1d) and *Ngf*-mScarlet expression (Fig. 1h) that 90% of the *Ngf*-expressing cells in the bone marrow were LepR⁺ stromal cells. Finally, we showed that deletion of *Ngf* from LepR⁺ cells was sufficient to eliminate nearly all of the NGF protein (Fig. 2a) and nerve fibers from the bone marrow (Fig. 2b and 2c) and to impair hematopoietic and vascular regeneration after myeloablation (Figure 3). Thus, the data are not consistent with the possibility that macrophages are the important source of NGF for the phenotypes reported in our study.

3. The authors should provide an explanation and discussion for the discrepancy between their findings and those of previous studies that show denervation of sympathetic nerves results in loss of HSCs (Hiromitsu Nakauch Cell 2011).

RESPONSE: We found that the loss of nerve fibers from the bone marrow of adult *Lep^{Cre/+}; Ngf^{fl/Δ}* mice did not affect HSC frequency or hematopoiesis under steady-state conditions but that HSC and hematopoietic regeneration after myeloablation were impaired. These findings are consistent with multiple studies published by Daniel Lucas and Paul Frenette that showed ablation of sympathetic nerves from the bone marrow using 6-hydroxydopamine did not affect HSC frequency or hematopoiesis under steady state conditions but impaired the regeneration of HSCs and hematopoiesis after myeloablation (Nature 452:442, Nat Med 19:695, J Immunol 198:156; Cell 124:2). In contrast to all of these studies, Hiro Nakauchi's study (Cell 147:1146) showed that bone marrow HSCs were depleted at steady state when the lumbar sympathetic trunk was surgically cut to denervate femur muscle and bone marrow. We don't know why the Nakauchi study observed effects that were different from most of the other studies published in this area but our results are consistent with most of the studies of nerve fiber function in the bone marrow.

One difficulty in comparing the results from studies in this area is that every study took a different approach to eliminate nerve fibers. Most of the studies used neurotoxins that would be expected to have systemic effects on the peripheral nervous system. The Nakauchi study used a surgical approach, which might promote inflammation in addition to eliminating nerve fibers. Our study has the advantage of taking a genetic approach that did not involve neurotoxins or surgery.

We hope these new data and clarifications will render our manuscript acceptable for publication.

Sincerely,

Sean J. Morrison
Investigator, Howard Hughes Medical Institute
Director, Children's Research Institute
University of Texas Southwestern Medical Center

Decision Letter, first revision:

6th September 2023

Dear Sean,

Thank you for submitting your revised manuscript "Leptin Receptor+ cells promote bone marrow innervation and regeneration by synthesizing nerve growth factor" (NCB-A50456A). It has now been seen by the original referees and their comments are below. The reviewers find that the paper has improved in revision, and therefore we'll be happy in principle to publish it in Nature Cell Biology, pending minor revisions to comply with our editorial and formatting guidelines.

If the current version of your manuscript is in a PDF format, please email us a copy of the file in an editable format (Microsoft Word or LaTeX)-- we cannot proceed with PDFs at this stage.

Thank you again for your interest in Nature Cell Biology. Please do not hesitate to contact me if you have any questions.

Best regards,
Stelios

Stylianos Lefkopoulos, PhD
He/him/his
Associate Editor
Nature Cell Biology
Springer Nature
Heidelberger Platz 3, 14197 Berlin, Germany

E-mail: stylianos.lefkopoulos@springernature.com
Twitter: @s_lefkopoulos

Reviewer #1 (Remarks to the Author):

The authors have satisfactorily addressed all of my concerns.

Reviewer #2 (Remarks to the Author):

The authors have addressed all my concerns. Congratulations on an exciting manuscript.

Reviewer #3 (Remarks to the Author):

The authors have addressed and alleviated all of my concerns, and thus I fully recommend acceptance of the manuscript.

Final Decision Letter:

Dear Sean,

I am pleased to inform you that your manuscript, "Leptin Receptor+ cells promote bone marrow innervation and regeneration by synthesizing nerve growth factor", has now been accepted for publication in Nature Cell Biology.

Please note that *Nature Cell Biology* is a Transformative Journal (TJ). Authors may publish their research with us through the traditional subscription access route or make their paper immediately open access through payment of an article-processing charge (APC). Authors will not be required to make a final decision about access to their article until it has been accepted. Find out more about Transformative Journals

If you have not already done so, we strongly recommend that you upload the step-by-step protocols used in this manuscript to the Protocol Exchange (www.nature.com/protocolexchange), an open online resource established by Nature Protocols that allows researchers to share their detailed experimental know-how. All uploaded protocols are made freely available, assigned DOIs for ease of citation and are fully searchable through nature.com. Protocols and Nature Portfolio journal papers in which they are used can be linked to one another, and this link is clearly and prominently visible in the online versions of both papers. Authors who performed the specific experiments can act as primary authors for the Protocol as they will be best placed to share the methodology details, but the Corresponding Author of the present research paper should be included as one of the authors. By uploading your Protocols to Protocol Exchange, you are enabling researchers to more readily reproduce or adapt the methodology you use, as well as increasing the visibility of your protocols and papers. You can also establish a dedicated page to collect your lab Protocols. Further information can be found at www.nature.com/protocolexchange/about

With kind regards,
Stelios

Stylianos Lefkopoulos, PhD

He/him/his
Senior Editor, Nature Cell Biology
Springer Nature
Heidelberger Platz 3, 14197 Berlin, Germany

E-mail: stylianos.lefkopoulos@springernature.com
Twitter: [@s_lefkopoulos](https://twitter.com/s_lefkopoulos)
LinkedIn: [linkedin.com/in/stylianos-lefkopoulos-81b007a0](https://www.linkedin.com/in/stylianos-lefkopoulos-81b007a0)